# Leaf Area Index identified as a major source of variability in modelled CO₂ fertilization

Qianyu Li[1,4], Xingjie Lu[2], Yingping Wang[3], Xin Huang[2], Peter M. Cox[4], Yiqi Luo[1,2]

[1]Ministry of Education Key Laboratory for Earth System Modeling, Department of Earth System Science, Tsinghua University, Beijing 100084, China

[2] Center for Ecosystem Science and Society (Ecoss), Northern Arizona University, Flagstaff, AZ 86011, USA

[3] CSIRO Oceans and Atmosphere, PMB #1, Aspendale, Victoria 3195, Australia

[4]College of Engineering, Mathematics and Physical Sciences, University of Exeter, Exeter, EX4 4QF, UK

*Correspondence to*: Yiqi Luo (luoyiqi@mail.tsinghua.edu.cn)

**Abstract.** The concentration-carbon feedback ($\beta$), also called the $CO_2$ fertilization effect, is a key unknown in climate-carbon cycle projections. A better understanding of model mechanisms that govern terrestrial ecosystem responses to elevated $CO_2$ is urgently needed to enable a more accurate prediction of future terrestrial carbon sink. We conducted C-only, carbon-nitrogen (C-N) and carbon-nitrogen-phosphorus (C-N-P) simulations of the Community Atmosphere Biosphere Land Exchange model (CABLE) from 1901 to 2100 with fixed climate to identify the most critical model process that causes divergence in $\beta$. We calculated $CO_2$ fertilization effects at various hierarchical levels from leaf biochemical reaction and leaf photosynthesis to canopy gross primary production (GPP), net primary production (NPP), and ecosystem carbon storage (*cpool*) for seven $C_3$ plant functional types (PFTs) in response to increasing $CO_2$ under RCP 8.5 scenario. Our results show that $\beta$ values at biochemical and leaf photosynthesis levels vary little across the seven PFTs, but greatly diverge at canopy and ecosystem levels in all simulations. The low variation of the leaf-level $\beta$ is consistent with a theoretical analysis that leaf photosynthetic sensitivity to increasing $CO_2$ concentration is almost an invariant function. In the CABLE model, the major jump in variation of $\beta$ values from leaf- to canopy- and ecosystem-levels results from divergence in modelled leaf area index (LAI) within and among PFTs. The correlation of $\beta_{GPP}$, $\beta_{NPP}$, or $\beta_{cpool}$ each with $\beta_{LAI}$ is very high in all simulations. Overall, our results indicate that modelled LAI is a key factor causing the divergence in $\beta$ in the CABLE model. It is therefore urgent to constrain processes that regulate LAI dynamics in order to better represent the response of ecosystem productivity to increasing $CO_2$ in Earth System Models.

## 1. Introduction

Terrestrial ecosystems take up roughly 30% of anthropogenic $CO_2$ emissions, and is of great uncertainty and vulnerable to global climate change (Cox et al., 2000; Le Quéré et al., 2018). Persistent increase of atmospheric $CO_2$ concentration will stimulate plant growth and ecosystem carbon storage, forming a negative feedback to $CO_2$ concentration (Long et al., 2004; Friedlingstein et al., 2006; Canadell et al., 2007). This concentration-carbon feedback ($\beta$), also called the $CO_2$ fertilization effect, has been identified as a major uncertainty in modeling terrestrial carbon-cycle response to historical climate change (Huntzinger et al., 2017). In the Coupled Model Intercomparison Project (C4MIP) and the Coupled Model Intercomparison Project Phase 5 (CMIP5), all models agree that terrestrial carbon sink will gradually saturate in the future but disagree on the magnitude of $\beta$ (Friedlingstein et al., 2006; Arora et al., 2013; Friedlingstein et al., 2015). Some studies pointed out that the contribution of $\beta$ is 4 to 4.5 times larger, and more uncertain, than climate-climate feedback ($\gamma$) (Gregory et al., 2009; Bonan and Levis, 2010; Arora et al., 2013). Apart from the substantial uncertainty across different models, Smith et al. (2016) suggested that Earth System Models (ESMs) in CMIP5 overestimate global terrestrial $\beta$ values compared with remote sensing data and Free-Air $CO_2$ Enrichment (FACE) experimental results. Though satellite products they used may underestimate the effect of $CO_2$ fertilization on net primary productivity (De Kauwe et al., 2016), the large disparity between models and FACE experiments gives us little confidence in making policies to combat global warming.

The response of ecosystem carbon cycle to elevated $CO_2$ (e$CO_2$) is primarily driven by stimulation of leaf-level carboxylation rate in plants (Polglase and Wang, 1992; Long et al., 2004; Heimann et al., 2008). The $CO_2$ stimulation of carboxylation then translates into increasing gross primary production (GPP) and net primary production (NPP), possibly leading to increased biomass and soil carbon storage and slowing down anthropogenically driven increase in atmospheric $CO_2$ (Canadell et al., 2007; Iversen et al., 2012). The leaf-level $CO_2$ fertilization for $C_3$ plants is generally well characterized with models from Farquhar et al. (1980), and the basic biochemical mechanisms have been adopted by most land surface models although some models implement variants of Farquhar et al. (1980) (Rogers et al., 2017). Previous research with both theoretical analysis and data synthesis from a large number of experiments has revealed that normalized $CO_2$ sensitivity of leaf-level photosynthesis,

which represents kinetics sensitivity of photosynthetic enzymes, varies little among different $C_3$ species at a given $CO_2$ concentration (Luo and Mooney, 1996; Luo et al., 1996). However, the $CO_2$ fertilization effects are considerably more variable at canopy- and ecosystem-level than at the leaf-level, because a cascade of uncertain processes, such as soil moisture feedback (Fatichi et al., 2016), canopy scaling (Rogers et al., 2017), nutrient limitation (Zaehle et al., 2014), allocation (De Kauwe et al., 2014), and carbon turnover process (Friend et al., 2014) influence the responses of GPP, NPP and carbon storage. Therefore, understanding which processes in ecosystem models amplify the variability in $\beta$ from biochemical and leaf levels to canopy and ecosystem levels is quite important.

Leaf area index (LAI) largely affects canopy assimilation and plant growth under $eCO_2$. Many satellite products exhibit increasing trends of LAI over the past 30 years although marked disparity still exists among these products (Jiang et al., 2017). Zhu et al. (2016) has attributed global increases in satellite LAI primarily to increased $CO_2$ concentration. LAI plays a key role in scaling leaf-level biogeophysical and biogeochemical processes to global scale responses in ecosystem models, and the representation of LAI in models causes large uncertainty (Ewert, 2004; Hasegawa et al., 2017). Models generally predict that LAI dynamics will respond to $eCO_2$ positively due to enhanced NPP and leaf biomass (De Kauwe et al., 2014). But how the increasing LAI in turn feeds back to ecosystem carbon uptake as a result of more light interception has not been discussed in previous research. The relative contributions of the leaf-level photosynthesis and LAI to modelled $\beta$ have been rarely quantified and compared.

The $CO_2$ fertilization effects depend on locations, vegetation types and soil nutrient conditions. The strongest absolute $CO_2$ fertilization effect has been found in tropical and temperate forests where the larger biomass presents than other regions. In comparison, the weakest response to $eCO_2$ occurs in boreal forests (Joos et al., 2001; Peng et al., 2014). But with gradual $eCO_2$, relative response in tropical forests might not be very high owing to light limitation caused by canopy closure (Norby et al., 2005). In addition, $\beta$ might be overestimated by the neglect of nitrogen (N) limitations on plant growth (Luo et al., 2004; Thornton et al., 2009; Coskun et al., 2016). Several lines of evidence suggest that N availability also influences decomposition

of soil organic matter (Hunt et al., 1988; Neff et al., 2002; Averill et al., 2016). $\beta$ will be reduced by 50–78% in C-N coupled simulations compared with C-only simulations in land surface models (Thornton et al., 2007; Sokolov et al., 2008; Zaehle et al., 2010). Inadequate phosphorus (P) will also constrain terrestrial carbon uptake, especially in tropical area (Aerts and Chapin, 2000; Vitousek et al., 2010). It is reported that N limitation on carbon uptake is significant in boreal ecosystems, while P limitation has a profound influence in tropical ecosystems in CASA-CNP model (Wang et al., 2010). However, whether N and P limitations affect the variability of $\beta$ across different vegetation types at different hierarchical levels from biochemistry to ecosystem carbon storage, have not been carefully examined.

In this study, we tried to answer the following questions: how variability, as measured by coefficient of variation (CV) within and across different plant functional types (PFTs), in the $CO_2$ fertilization effects changes at different hierarchical levels from leaf to canopy GPP, ecosystem NPP and total carbon storage levels? What is the most important process causing the variability of $\beta$ for different geographical locations and PFTs? How nutrient limitations influence the variability of $\beta$ at different hierarchical levels? We used Community Atmosphere Biosphere Land Exchange model (CABLE) to identify key mechanisms driving diverse $\beta$ values under RCP 8.5 scenario.

## 2. Materials and methods

### 2.1 CABLE model description

CABLE (version 2.0) is the Australian community land surface model (Kowalczyk et al., 2006) and incorporates CASA-CNP to simulate global carbon (C), nitrogen (N) and phosphorus (P) cycles (Wang et al., 2010; Wang et al., 2011). Leaf photosynthesis, stomatal conductance, and heat and water transfer in CABLE are calculated using the two-leaf approach (Wang and Leuning, 1998) for both sunlit leaves and shaded leaves. The descriptions of photosynthesis module are in supplementary Text S1.

Leaf Area Index (LAI) is calculated as:

110 $\quad$ $\text{LAI} = C_{leaf} * \text{SLA}$ $\hspace{8cm}$ (1)

Where $C_{leaf}$ is leaf carbon pool, and SLA is specific leaf area.

In the CABLE model, leaf growth is divided into four phases. Phase 1 is from leaf budburst to the beginning of steady leaf growth, phase 2 is from the start of steady leaf growth to the start of leaf senescence, phase 3 is the period of leaf senescence, and phase 4 is from the end of leaf senescence to the start of leaf bud burst. During phase 1, allocation of available carbon to

115 $\quad$ leaf is fixed to 0.8, and allocation to wood and root are set to 0.1 for woody biomes, and 0 and 0.2 respectively for non-woody biomes. During steady leaf growth (phase 2), the allocation coefficients are constants but vary from biome to biome, taking their values from Fung et al. (2005). During phases 3 and 4, the leaf allocation is zero and available carbon is divided between wood and root in proportional to their allocation coefficients. For evergreen biomes, leaf phenology remains at phase 2 throughout the year (Wang et al., 2010). SLA is PFT-specific and does not change through time in this study.

120

GPP is the sum of canopy net photosynthesis rate ($A$) and day respiration ($R_d$). NPP is calculated as the difference between GPP and autotrophic respiration ($R_a$) (including maintenance and growth respiration), and acts as an input to the compartmental nine-pool carbon cycle model. The network for carbon transfer in the compartmental model is based on CASA' model (Fung et al., 2005), including three vegetation pools (leaf, wood and root), three litter pools (metabolic litter, structure

125 $\quad$ litter and coarse wood debris), three soil pools (fast soil pool, slow soil pool and passive soil pools). Heterotrophic soil respiration ($R_h$) is calculated as the sum of the respired $CO_2$ from the decomposition of all litter and soil organic carbon pools (Wang et al., 2010).

Wang et al. (2012) and Zhang et al. (2013) provided details explaining how nutrient limitations are incorporated into carbon

130 $\quad$ cycle in CASA-CNP module in the CABLE model. In brief, NPP is calculated as:

$\text{NPP} = \text{GPP}(L, V_{cmax}(N_l), J_{max}(N_l)) - \sum_i R_{mi}(N_i) - R_g(\frac{N_l}{P_l})$ $\hspace{4cm}$ (2)

Where L represents leaf area index, $V_{cmax}$ and $J_{max}$ are maximum carboxylation rate and maximum rate of electron transport of the top leaves, respectively, both are linearly dependent on leaf N (g N m$^{-2}$) according to the relationships developed by Kattge et al. (2009) for different plant functional types. $R_{mi}$ is maintenance respiration rates of plant tissue (i=leaf, wood and root), contingent on nitrogen amount in each part of plant. $R_g$ is growth respiration, which is described as a function of leaf nitrogen to phosphorus ratio. Heterotrophic respiration ($R_h$) is limited by the mineral N pool required for microbial soil carbon decomposition (Wang et al., 2010). Net ecosystem productivity (NEP = GPP $-$ $R_a$ $-$ $R_h$) is the amount of carbon that is either sequestered or lost from ecosystems, and is controlled by N and P availability via abovementioned C-N-P interactions.

## 2.2 Experimental design

CABLE was run from 1901 to 2100 for C-only, C-N and C-N-P modes. C-only simulation was designed to identify the key carbon cycle processes that influence the variability of the $CO_2$ fertilization effects. C-N and C-N-P simulations were run to explore how nutrients affect the patterns of and mechanisms underlying the variability of the $CO_2$ fertilization effects. The respective effects of N and P can be calculated through the difference in the carbon uptake between C-N and C-only or C-N-P and C-N simulations. CABLE was first spun up by using meteorological forcing from Community Climate System Model (CCSM) simulations (Hurrell et al., 2013) during 1901 to 1910 until steady states were achieved for the C-only, C-N and C-N-P cases separately. Hourly meteorological driving data include: temperature, specific humidity, air pressure, downward solar radiation, downward long-wave radiation, rainfall, snowfall, and wind. In order to separate the $CO_2$ fertilization effect from the effect of climate change, climate forcing was held as the average annual cycle of CCSM meteorological data from 1901 to 2100. Atmospheric $CO_2$ concentrations from 1901 to 2100 were taken from the CMIP5 dataset, representing global annual averages and the RCP8.5 scenario after 2010 (Etheridge et al., 1996; MacFarling Meure et al., 2006). The spatial resolution of CABLE used here is 1.9°×2.5° (latitude vs longitude). N deposition is prescribed from atmospheric transport models (Lamarque et al., 2010, 2011), spatially explicit but fixed as the average from 1901 to 2100 in time. N fixation is prescribed from a process-based model, spatially explicit but constant in time (Wang and Houlton, 2009). P enters ecosystems through constant rates of weathering and atmospheric deposition (from Mahowald et al. (2008)).

## 2.3 Calculation of $\beta$ values at five hierarchical levels

We aimed to analyze the $CO_2$ fertilization effects for biochemical reaction ($\mathcal{L}$), leaf photosynthesis rate ($p$), leaf-to-canopy scaling factor ($S$), leaf area index (LAI), sunlit leaf GPP ($GPP_{sun}$), shaded leaf GPP ($GPP_{sha}$), canopy GPP, NPP, and ecosystem carbon storage (*cpool*) from C-only, C-N and C-N-P simulations of CABLE. Canopy GPP is the sum of sunlit leaf GPP and shaded leaf GPP. Ecosystem carbon storage is the sum of plant, litter and soil carbon stock. Since $CO_2$ concentration increases at yearly basis, annual carbon fluxes and storages such as $GPP_{sun}$, $GPP_{sha}$, canopy GPP, NPP and ecosystem carbon storage were calculated. Leaf-to-canopy scaling factor and LAI were averaged within a year. $\beta$ values of these variables were calculated as the normalized sensitivities of those variables to atmospheric $CO_2$ concentration ($C_a$) as $\beta_V$:

$$\beta_V = \frac{1}{V} * \frac{dV}{dC_a} \tag{3}$$

Where V in the denominator represents average annual value of $S_{sun}$, $S_{sha}$, LAI, GPP, $GPP_{sun}$, $GPP_{sha}$, NPP and ecosystem carbon storage between two consecutive years. Subscripts "*sun*" and "*sha*" denote the sunlit and shaded components. dV is the difference of these variables between two consecutive years. $dC_a$ is the difference of corresponding $C_a$. The unit of $\beta_V$ is $ppm^{-1}$. It should be noted that $\beta_V$ is the relative response, which is similar to the traditional definition of $\beta$ factor by Bacastow and Keeling (1973), but different from the carbon-concentration feedback parameter in Friedlingstein et al. (2006). The relative response facilitates the comparison among PFTs with different initial biomass and the comparison across carbon fluxes and storages with different units.

Leaf biochemical response ($\mathcal{L}$) was first proposed by Luo et al. (1996). $\mathcal{L}$ function is the normalized response of leaf photosynthesis rate to a small change in intercellular $CO_2$ concentration ($C_i$) and has been suggested to be an invariant function for C3 plants grown in diverse environments. The rate of photosynthesis is typically RuBP-regeneration-limited under high $CO_2$ concentration. We found photosynthesis rates are increasingly limited by RuBP regeneration under RCP 8.5 scenario. Besides, theoretical analysis by Luo and Mooney (1996) showed that biochemical responses are similar for either Rubisco- or

RuBP-limited photosynthesis. In this study, $\mathcal{L}$ can be used to indicate leaf biochemical response to eCO$_2$. For sunlit leaf and shaded leaf, formulations of $\mathcal{L}$ under RuBP-regeneration-limitation are defined as:

$$\mathcal{L}_{sun} = \frac{3 * \Gamma_{*sun}}{(C_{isun}+2*\Gamma_{*sun})(C_{isun}-\Gamma_{*sun})} \tag{4}$$

$$\mathcal{L}_{sha} = \frac{3 * \Gamma_{*sha}}{(C_{isha}+2*\Gamma_{*sha})(C_{isha}-\Gamma_{*sha})} \tag{5}$$

In this study, $\Gamma_{*sun}$ and $\Gamma_{*sha}$ are yearly average CO$_2$ compensation points in the absence of day respiration for sunlit leaf and shaded leaf, respectively. $C_i$ varies significantly at sub-daily, intra-annual and inter-annual bases. We're interested in how $C_i$ responds to eCO$_2$ on an inter-annual basis. So, we first outputted hourly $C_i$ then calculated yearly GPP-weighted average $C_i$ for sunlit leaf ($C_{isun}$) and shaded leaf ($C_{isha}$).


Then leaf-level $\beta_p$ is defined as the product of $\mathcal{L}$ and $\frac{\mathrm{d}C_i}{\mathrm{d}C_a}$. For sunlit leaf and shaded leaf, the formulations are:

$$\beta_{p_{sun}} = \mathcal{L}_{sun} * \frac{\mathrm{d}C_{isun}}{\mathrm{d}C_a} \tag{6}$$

$$\beta_{p_{sha}} = \mathcal{L}_{sha} * \frac{\mathrm{d}C_{isha}}{\mathrm{d}C_a} \tag{7}$$

Leaf-to-canopy scaling factor ($S$) scales fluxes at the single top leaf of the canopy to whole canopy fluxes. The formulations of $S$ for sunlit leaves and shaded leaves are:

$$S_{sun} = \frac{1-\exp[-\mathrm{LAI}(k_n+k_b)]}{k_n+k_b} \tag{8}$$

$$S_{sha} = \frac{1-\exp(-k_n\mathrm{LAI})}{k_n} - \frac{1-\exp[-\mathrm{LAI}(k_n+k_b)]}{k_n+k_b} \tag{9}$$

Where $k_b$ is extinction coefficient of a canopy of black leaves for direct beam radiation. $k_n$ is an empirical parameter used

to describe the vertical distribution of leaf nitrogen in the canopy (Kowalczyk et al., 2006). In our simulation, $k_n$ is uniformly

assigned as 0.001 for different PFTs. The leaf-to-canopy scaling factor varies with time because $k_b$ is the function of sun angle, and LAI varies seasonally and inter-annually. The annual value of the leaf-to-canopy scaling factor was just calculated as the average of hourly leaf-to-canopy scaling factors in a year.

Big-leaf $\beta_{\text{GPP}_{sun}}$ (or $\beta_{\text{GPP}_{sha}}$) can be decomposed as the sum of normalized sensitivity of photosynthesis rate: $\beta_{p_{sun}}$ (or $\beta_{p_{sha}}$) and leaf-to-canopy scaling factor: $\beta_{S_{sun}}$ (or $\beta_{S_{sha}}$) as shown in Eq. (10) and Eq. (11). Detailed mathematical derivations are in supplementary Text S2.

$$\beta_{\text{GPP}_{sun}} = \beta_{p_{sun}} + \beta_{S_{sun}} \tag{10}$$

$$\beta_{\text{GPP}_{sha}} = \beta_{p_{sha}} + \beta_{S_{sha}} \tag{11}$$


There are ten patches in each model grid in CABLE. Each patch consists of a certain land use type with a specific fraction. To study the variation of $\beta$ across different $C_3$ PFTs, biome-level parameters such as $\Gamma_{*sun}$, $C_{isun}$, $S_{sun}$ and LAI were calculated as mean values based on PFTs, whereas biome-level GPP, $\text{GPP}_{sun}$, $\text{GPP}_{sha}$, NPP and ecosystem carbon storage were integrated sums based on PFTs. Then $\mathcal{L}_{sun}$, $\mathcal{L}_{sha}$, $\beta_{p_{sun}}$, $\beta_{p_{sha}}$, $\beta_{\text{GPP}}$, $\beta_{\text{NPP}}$ and $\beta_{cpool}$ at the year 2023 (relative to

2022) for different $C_3$ PFTs were calculated and compared. Coefficients of variation (CVs) of $\beta$ values were calculated across various $C_3$ PFTs for these hierarchical levels. The year 2023 was chosen because large oscillations of LAI occurred for shrub after 2025 in the C-N-P simulation (Fig. S1c). For C-N and C-N-P simulations, the time series of LAI, GPP, and NPP for shrub, $C_3$ grass and tundra underwent small short-term variability and therefore were smoothed using the "smooth" function in MATLAB software before the calculation of $\beta$. We also calculated $\beta$ values for each patch and CV of $\beta$ values across

different geographical locations within a specific PFT at different hierarchical levels at the year of 2023 to explore the variability of $\beta$ within the same PFTs. All abovementioned calculations were processed in MATLAB R2014b.

## 3. Results

### 3.1 Temporal trends of $\beta$ at ecosystem level for different PFTs

In C-only simulation, $\beta_{cpool}$ values for different $C_3$ PFTs all decline with time from 2011 to 2100 under RCP8.5 scenario (Fig. 1a). However, the magnitudes of $\beta_{cpool}$ differ among different PFTs, with the highest values occurring in deciduous broadleaf forest from 2011 to 2075 and in shrub after 2075, and lowest values occurring in deciduous needleleaf forest and tundra. $\beta_{cpool}$ values for deciduous needleleaf forest and tundra nearly overlap over time. As compared with C-only simulation, values of $\beta_{cpool}$ are reduced when N limitation is included as in C-N simulation for all $C_3$ PFTs except evergreen broadleaf forest (Fig. 1b). Deciduous broadleaf forest and evergreen broadleaf forest have the greatest $\beta_{cpool}$ values, while deciduous needleleaf forest and tundra still have the lowest $\beta_{cpool}$ values in C-N simulation. When both N and P limitations are taken into account as in C-N-P simulation, magnitudes and trends of $\beta_{cpool}$ are similar to those in C-N simulation (Fig. 1c) as P limitation is quite weak under present condition in the current version of CABLE (Zhang et al., 2011).

### 3.2 Variations of intercellular $CO_2$ concentration and $CO_2$ compensation point

To reveal which processes cause the large disparity of $\beta$ across PFTs as shown in Fig. 1, we first compared intercellular $CO_2$ concentration ($C_i$) and $CO_2$ compensation point in the absence of day respiration ($\Gamma_*$), which are critical parameters for leaf-level biochemical response. In C-only simulation, the ratio of $C_i$ to $C_a$ ($C_i/C_a$) is approximately constant with $eCO_2$ for each PFT (Fig. 2a, 2b). For sunlit leaf, $C_i/C_a$ values range from 0.64 to 0.70 with CV=0.03 across different $C_3$ PFTs (Table 1). $C_i/C_a$ values for shaded leaf are higher than those for sunlit leaf, and the range is 0.68 to 0.76 with CV=0.03 across different $C_3$ PFTs (Table 1). Evergreen broadleaf forest has the greatest $C_i/C_a$ value, while deciduous needleleaf forest has the lowest $C_i/C_a$ value. In C-N simulation, $C_i/C_a$ values for sunlit leaf are lower than those for the same PFT in C-only simulation, while $C_i/C_a$ values for shaded leaf change little as compared with those for the same PFT in C-only simulation (Table 1 and Fig. S2). $C_i/C_a$ values for both sunlit and shaded leaves in C-N-P simulation are very similar to those in C-N simulation (Table 1 and Fig. S3).

In all of the simulations, values of $CO_2$ compensation point in the absence of day respiration ($\Gamma_*$) for a specific PFT do not

change over time since air temperature as an input to the model is not affected by the biophysical feedback in the offline model

simulations (Fig. 2c, 2d, S2c, S2d, S3c, S3d). But there is a huge variance of $\Gamma_*$ across different $C_3$ PFTs because of different

leaf temperature which $\Gamma_*$ values depend on.

### 3.3 Comparison of $\beta$ at different hierarchical levels

To further trace the cause for the divergence of $\beta$ across PFTs as shown in Fig. 1 at a specific time, $\mathcal{L}_{sun}$, $\mathcal{L}_{sha}$, $\beta_{p_{sun}}$, $\beta_{p_{sha}}$,

$\beta_{GPP}$, $\beta_{NPP}$ and $\beta_{cpool}$ at the year 2023 for different $C_3$ PFTs in all simulations were plotted in Fig. 3. CV is marked above

data points for each variable to indicate degree of variation across different $C_3$ PFTs. In C-only simulation (Fig. 3a), results

show that at leaf biochemical level, $\mathcal{L}$ values for sunlit leaf and shaded leaf range from 0.00055 ppm$^{-1}$ to 0.00097 ppm$^{-1}$.

Variations of $\mathcal{L}_{sun}$ and $\mathcal{L}_{sha}$ among PFTs are small (CV=0.15 and 0.13). At leaf photosynthesis level, $\beta_{p_{sun}}$ and $\beta_{p_{sha}}$ for

the seven PFTs vary from 0.00041 ppm$^{-1}$ to 0.00072 ppm$^{-1}$, and the variations among different PFTs are not significant

(CV=0.18 and 0.12). But $\beta$ values are diverging when scaled up to GPP level with CV jumping to 0.49 among PFTs. $\beta$ values

of deciduous broadleaf forest and shrub greatly increase from leaf level to GPP level. However, canopy scaling effects do not

significantly amplify $\beta$ values at canopy levels ($\beta_{GPP}$) for deciduous needleleaf forest, tundra and evergreen broadleaf forest.

Magnitudes and variance of $\beta_{NPP}$ are similar to those of $\beta_{GPP}$ because NPP linearly correlates with GPP for all $C_3$ PFTs (Fig.

S4). Magnitudes of $\beta_{cpool}$ for all PFTs are decreased compared with those of $\beta_{NPP}$ and $\beta_{GPP}$. Deciduous broadleaf forest and

shrub have the highest $\beta_{GPP}$ and $\beta_{NPP}$ values (around 0.0026 ppm$^{-1}$). Deciduous broadleaf forest has the greatest $\beta_{cpool}$

value (around 0.0018 ppm$^{-1}$) among all. Deciduous needleleaf forest has the lowest $\beta_{GPP}$, $\beta_{NPP}$ and $\beta_{cpool}$ values. CV of

$\beta_{cpool}$ among different PFTs reaches the highest (0.58) compared with CV of $\beta$ values at other levels.

In C-N and C-N-P simulations, magnitudes and variations of $\beta$ at leaf biochemical and photosynthetic levels are comparable

to those in C-only simulation because $C_i$ and $\Gamma_*$ values only slightly change under nutrient limitations (Fig. 3b, 3c, S2, S3).

Nutrient-limited $\beta_{GPP}$ values are smaller than those in C-only simulation, except for evergreen broadleaf forest. There is a

large divergence of nutrient-limited $\beta_{\mathrm{GPP}}$ across different PFTs, which is similar to C-only simulation. However, unlike in C-only simulation, $\beta_{\mathrm{NPP}}$ values in nutrient-coupled simulations are reduced for most C$_3$ PFTs and diverge more compared with $\beta_{\mathrm{GPP}}$ values. Coefficients of variation (CVs) of $\beta_{cpool}$ in nutrient-coupled simulations exceed 0.8, larger than that in C-only simulation.

Within-PFT variations of $\beta$ in C-only simulation were listed in Table 2, including CVs for biochemical response $\mathcal{L}$, leaf-level $\beta_p$, $\beta_{\mathrm{GPP}}$, $\beta_{\mathrm{NPP}}$ and $\beta_{cpool}$ across different geographical locations within each PFT. Variations of biochemical and leaf-level responses are relatively smaller than those at canopy and ecosystem levels within all C$_3$ PFTs. $\beta_{\mathrm{GPP}}$ values greatly differentiate across different geographical locations. Variations of $\beta_{\mathrm{NPP}}$ are very similar to those of $\beta_{\mathrm{GPP}}$ within all PFTs except the evergreen needleleaf forest. CVs of $\beta_{cpool}$ are lower than those of $\beta_{\mathrm{NPP}}$ within most PFTs except evergreen broadleaf forest and tundra. Within-PFT variations of $\beta$ in C-N and C-N-P simulations are similar to those in C-only simulation (data not shown).

To further explore why $\beta$ values at canopy and ecosystem levels are diverging across different C$_3$ PFTs, the correlations between $\beta_{\mathrm{GPP}}$ and $\beta_{\mathrm{LAI}}$, $\beta_{\mathrm{NPP}}$ and $\beta_{\mathrm{LAI}}$, $\beta_{cpool}$ and $\beta_{\mathrm{LAI}}$ for C-only, C-N and C-N-P simulations were plotted at the year 2023. Results show that $\beta_{\mathrm{GPP}}$, $\beta_{\mathrm{NPP}}$ and $\beta_{cpool}$ all have significant linear correlations with $\beta_{\mathrm{LAI}}$ across different C$_3$ PFTs (Fig. 4). Results also show that $\beta_{\mathrm{LAI}}$ linearly correlates with $\beta_{\mathrm{GPP}}$, $\beta_{\mathrm{NPP}}$ and $\beta_{cpool}$ across patches within the same PFT, although there are some discontinuous points within evergreen broadleaf forest where canopy of many patches closes (Fig. S5-S7). Therefore variations of $\beta$ values from leaf to ecosystem scale can be well explained by $\beta_{\mathrm{LAI}}$, or the LAI response to increasing $CO_2$.

### 3.4 $\beta$ of sunlit and shaded leaves

To understand the in-depth mechanism for the influence of LAI on canopy GPP, we investigate the response of sunlit and shaded leaf GPP separately from C-only simulation. Temporal trends of sunlit leaf GPP ($GPP_{sun}$) and shaded leaf GPP ($GPP_{sha}$)

were plotted for each type of C$_3$ PFTs from 1901 to 2100 in Fig. 5. From the beginning of the simulation, GPP$_{sha}$ is higher than GPP$_{sun}$ for all C$_3$ PFTs. With significant increases of CO$_2$ concentration from 2011, GPP$_{sha}$ responds more drastically than GPP$_{sun}$. Shaded leaf GPP of deciduous broadleaf forest and shrub responds to eCO$_2$ more significantly than other PFTs. However, a single sunlit leaf has higher photosynthesis rate ($p_{sun}$) than a shaded leaf ($p_{sha}$) because of more radiation absorbed.

Thus, the LAI-dependent canopy scaling factor of shaded leaves ($S_{sha}$) contributes more to the magnitude and sensitivity of canopy GPP than photosynthesis rate.

Then temporal trends were plotted for $\beta_{\text{GPP}_{sun}}$ ($\beta_{\text{GPP}_{sha}}$) and decomposing factors $\beta_{p_{sun}}$ ($\beta_{p_{sha}}$) and $\beta_{S_{sun}}$ ($\beta_{S_{sha}}$) for each PFT as Eq. (10) and Eq. (11) to further evaluate the above inference. Results show that both of the sensitivities of GPP$_{sun}$ and

GPP$_{sha}$ tend to approach zero through time because the decomposing factors $\beta_{p_{sun}}$, $\beta_{p_{sha}}$, $\beta_{S_{sun}}$ and $\beta_{S_{sha}}$ all decline with time (Fig. 6). $\beta_{p_{sun}}$ and $\beta_{p_{sha}}$ overlap through time for each PFT. Magnitudes of $\beta_{\text{GPP}_{sha}}$ are higher than those of $\beta_{\text{GPP}_{sun}}$ for all C$_3$ PFTs. For deciduous needleleaf forest and tundra, both $\beta_{p_{sun}}$ ($\beta_{p_{sha}}$) and $\beta_{S_{sun}}$ ($\beta_{S_{sha}}$) contribute to the maginitudes and trends of $\beta_{\text{GPP}_{sun}}$ ($\beta_{\text{GPP}_{sha}}$). For evergreen needleleaf forest, deciduous broadleaf forest, shrub and C$_3$ grass, $\beta_{S_{sun}}$ ($\beta_{S_{sha}}$) dominates the magnitude and change of $\beta_{\text{GPP}_{sun}}$ ($\beta_{\text{GPP}_{sha}}$). For evergreen broadleaf forest, $\beta_{S_{sha}}$

predominates the magnitude and change of $\beta_{\text{GPP}_{sha}}$ before 2035.

## 4. Discussion

### 4.1 Variations of biochemical and photosynthetic responses to eCO$_2$

The direct CO$_2$ fertilization effect occurs at leaf level and is determined by kinetic sensitivity of Rubisco enzymes to internal leaf CO$_2$ concentration. In fact, the normalized short-term sensitivity of leaf-level photosynthesis to CO$_2$ is mainly regulated

by $C_i$ and slightly influenced by leaf temperature, regardless of light, nutrient availability, and species characteristics (Luo et al., 1996; Luo and Mooney, 1996). In our study, modelled $C_i/C_a$ ratio is approximately constant with eCO$_2$ for a specific PFT, and varies little within and across PFTs in all simulations. This is in line with FACE experimental results which show almost

constant $C_i/C_a$ values for different PFTs under $CO_2$ fertilization (Drake et al., 1997; Long et al., 2004). $\Gamma_*$ varies little for different species and only depends on leaf temperature (Luo and Mooney, 1996). Sensitivity analysis in a previous study has shown that a $\pm 5C°$ of leaf temperature changes caused approximately $\pm 7$ ppm changes in $\Gamma_*$, leading to variation of 0.12 to leaf-level $\beta$ (Luo and Mooney, 1996). The overall variation of leaf-level $\beta$ caused by variation in leaf temperature is still quite small compared with that of $\beta_{GPP}$. Therefore, biochemical and leaf-level $\beta$ values vary little within and among PFTs in this study. Our results also illustrate that nutrient effects do not significantly change $C_i$ and $\Gamma_*$, leading to similar biochemical and leaf-level $\beta$ values in all simulations, which is in accordance with Luo et al. (1996).

To identify the source of uncertainty of $\beta$ in CMIP5 models, Hajima et al. (2014) decomposed $\beta$ into several carbon cycle components. They used GPP divided by LAI (GPP/LAI) as a proxy to represent leaf-level photosynthesis for CMIP5 models, since there are no leaf-level process outputs of these models. They found the sensitivities of GPP/LAI to $eCO_2$ diverged a lot among models. One possible issue of this calculation is that it ignores different canopy structure used by each CMIP5 model such as big-leaf, two-leaf or multiple-layer. Our results just show that the sensitivities of GPP/LAI are different from our mechanistic calculation of leaf-level $\beta$ for different PFTs in a two-leaf model. $\beta$ values estimated from GPP/LAI formulation are greatly underestimated for woody trees and slightly overestimated for $C_3$ grass and tundra, but best match for shrub if compared with our calculation (Fig. S8). Therefore diagnostics such as $C_i$ and $\Gamma_*$ for leaf-level $\beta$ are more desirable for woody trees. Another advantage of our calculation of leaf-level $\beta$ is that the reason for the divergence of leaf-level $\beta$ across PFTs can be traced back to the difference from $C_i$ and leaf temperature as shown in Fig. 2.

### 4.2 Variations of $\beta$ at canopy and ecosystem levels

The two-leaf scaling scheme in CABLE is widely employed by many land surface models, such as Community Land Model version 4.5 (CLM4.5, Oleson et al., 2013) and the Joint UK Land Environment Simulator version 4.5 (JULES4.5, Best et al., 2011; Clark et al., 2011; Harper et al., 2016). We found the responses of ecosystem carbon cycle to $eCO_2$ diverge primarily because the responses of LAI diverge within and among PFTs in all simulations. Besides, GPP of shaded leaves responds to

eCO$_2$ stronger than GPP of sunlit leaves for all C$_3$ PFTs. This is because the portion of shaded leaves increase exponentially with increasing LAI (Fig. S9), leading to a rapid change of shaded leaf GPP. While for sunlit leaves, GPP shows a saturating response because of the decreasing portion of sunlit leaves with increasing LAI (Dai et al., 2004). Our results also indicate that saturation of GPP is not only regulated by the leaf-level photosynthetic response, but also by the response of the LAI-dependent

scaling factor to eCO$_2$. For shaded leaves, the sensitivity of the LAI-dependent scaling factor contributes more to the magnitude and trend of $\beta_{\mathrm{GPP}_{sha}}$ than that of photosynthesis rate. The evidence all suggests LAI is a key process in modeling the response of ecosystem carbon cycle to climate change.

It has been reported that different CMIP5 models have simulated diverse LAI during 1985-2006. And modelled LAI values in

most CMIP5 models have been overestimated according to satellite products (Anav et al., 2013). Many global vegetation models simulated increasing LAI trends globally in response to eCO$_2$ during historical period (Zhu et al., 2016). Our modelling study also shows that LAI responds positively to eCO$_2$ for all C$_3$ PFTs in all simulations. But experimental results are not consistent. In one review paper with 12 FACE experimental results, trees had a 21% increase in LAI, herbaceous C$_3$ grasses did not show a significant change in LAI (Ainsworth and Long, 2005). Some studies reported that LAI dynamics did not

significantly change in specific FACE experiments, such as in a closed-canopy deciduous broadleaf forest (ORNL FACE; Norby et al., 2003) and in a mature evergreen broadleaf forest (EucFACE; Duursma et al., 2016). The negligible change of LAI at the EucFACE probably leads to insignificant response of productivity at this site, even though leaf photosynthesis rate significantly increases under eCO$_2$ (Ellsworth et al., 2017). Besides the impact of LAI on global carbon cycle, the increasing trend of LAI exerts profound biophysical impacts to climate through altering the energy and water cycles on the Earth's surface

(Forzieri et al., 2017; Zeng et al., 2017). But there is a great uncertainty in the relationships between LAI and biophysical processes among land surface models (Forzieri et al., 2018).

In this study, modelled nutrient-unlimited $\beta_{\mathrm{GPP}}$ and $\beta_{\mathrm{NPP}}$ values are higher than leaf photosynthetic responses for all C$_3$ PFTs in C-only simulation (Fig. 3a). Nutrient-limited $\beta_{\mathrm{NPP}}$ are still higher than photosynthetic responses for many PFTs in

C-N and C-N-P simulations (Fig. 3b, 3c). However, it is generally observed in experiments that the leaf-level response is consistently larger than the whole plant response (Long et al., 2006; Leuzinger et al., 2011). One possible reason is that models overestimate the response of LAI to $eCO_2$, as this study has shown that LAI is an important factor in driving ecosystem response to $CO_2$ fertilization. And it is also likely the overestimation of the response of LAI to $eCO_2$ is responsible for the overestimation of $CO_2$ fertilization in ESMs reported by previous studies (Smith et al., 2015; Mystakidis et al., 2017).


The overall response of LAI to $eCO_2$ depends on several processes in this study: (1) NPP increase, (2) change in allocation of NPP to leaf, (3) change in specific leaf area (SLA) in response to $eCO_2$, (4) PFT-specific minimum and maximum LAI values prescribed in the model. First, the low responses of LAI to $eCO_2$ for deciduous needleleaf forest and tundra can be attributed to smaller NPP enhancements in cold areas. The large divergence of the response of LAI within PFTs is mainly due to the

large range of NPP increment across different geographical locations. The reduced magnitudes of $\beta_{LAI}$ under nutrient limitations is the direct outcome of reduced $\beta_{NPP}$. Accurate estimate of response of GPP and NPP is therefore fundamental to realistic LAI modeling. Second, diverse allocation schemes influence the responses of LAI for different PFTs. And, results from two FACE (Duke Forest and Oak Ridge) experiments indicate that the carbon allocated to leaves is decreased and more carbon is allocated to woods or roots at higher $CO_2$ concentration (De Kauwe et al., 2014). Unfortunately, CABLE has fixed

allocation coefficients and likely overestimates LAI response, leading to overestimated responses of GPP, NPP and total carbon storage. Third, we fixed SLA to calculate LAI in CABLE. But a reduction in SLA is a commonly observed response in $eCO_2$ experiments (Luo et al., 1994; Ainsworth and Long, 2005; De Kauwe et al., 2014). Tachiiri et al. (2012) also found SLA and $\beta$ values are most effectively constrained by observed LAI to smaller values in a model. Therefore, the fixed SLA may also lead to over-prediction of the response of canopy cover to $eCO_2$. Forth, in our results, LAI values for most $C_3$ PFTs are below

the maximum LAI limits with $eCO_2$ in C-only simulation. With only one exception, LAI values of many evergreen broadleaf forest patches saturate at the prescribed maximum value under high $CO_2$ concentration (Fig. S1a and Table. S1). That's why the sensitivity of LAI for evergreen broadleaf forest is low and thus leads to small relative GPP enhancements. If the preset LAI upper limits are narrowed, $\beta$ values are expected to be significantly reduced. Hence model parameters related to LAI

need to be better calibrated according to experiments and observations in order to better represent the response of ecosystem
productivity to eCO$_2$ (De Kauwe et al., 2014; Qu and Zhuang, 2018).

In this study, the almost identical values and variance of $\beta_{\text{NPP}}$ as those of $\beta_{\text{GPP}}$ within and across C$_3$ PFTs in C-only

simulation suggests carbon use efficiency (CUE) does not change with eCO$_2$, as autotrophic respiration is calculated from GPP

and plant carbon. In C-N and C-N-P simulations, magnitudes of $\beta_{\text{NPP}}$ for all C$_3$ PFTs except evergreen broadleaf forest all

decline compared with those of $\beta_{\text{GPP}}$, indicating CUE also decline with eCO$_2$ under nutrient limitations. However, FACE

experimental results indicate that CUE values under eCO$_2$ are not changed in N-limited Duke site (Hamilton et al., 2002;

Schäfer et al., 2003), increase in fertile POPFACE site (Gielen et al., 2005) or decrease in fertile ORNL site (DeLucia et al.,

2005). Thus, representations of nutrient effects on GPP and autotrophic respiration in land surface models should be carefully

calibrated with experimental data (DeLucia et al., 2007). Our results also show that $\beta_{\text{NPP}}$ values diverge more than $\beta_{\text{GPP}}$

values across different PFTs in nutrient-coupled simulations, because the different nutrient-limiting effects on autotrophic

respiration introduce additional variation across different PFTs. Although $\beta$ values at ecosystem levels are more variable

with nutrient effects, LAI responses are still linearly correlated well with $\beta_{\text{GPP}}$, $\beta_{\text{NPP}}$ and $\beta_{cpool}$ across C$_3$ PFTs in nutrient-

coupled simulations as in C-only simulation, confirming the dominant role of LAI in regulating carbon cycle response under

CO$_2$ fertilization.


The reduced magnitudes of $\beta_{cpool}$ compared with those of $\beta_{\text{GPP}}$ and $\beta_{\text{NPP}}$ in all simulations indicates carbon turnover

processes make ecosystems respond to eCO$_2$ less sensitively due to the slow allocation and carbon turnover processes. A

previous study using seven global vegetation models identified carbon residence time as the dominant cause for uncertainty in

terrestrial vegetation responses to future climate and atmospheric CO$_2$ change (Friend et al., 2014). The response of soil carbon

storage to eCO$_2$ also depends on soil carbon residence time (Harrison et al., 1993). In this study and many other models,

allocation coefficients are fixed over time (Walker et al., 2014). But allocation pattern to plant organs with different lifespan

has been reported to change in response to eCO$_2$ in experiments, thereby altering carbon residence time in plants and soil (De

Kauwe et al., 2014). Therefore, the fixed allocation scheme we adopted in this study might lead to some biases in simulating

the response of carbon residence time to eCO$_2$. In our study, soil decomposition rate is assumed not to be affected by CO$_2$

level, as in most other conventional soil carbon models (Friedlingstein et al., 2006; Luo et al., 2016). However, recent synthesis

of experimental data suggests replenishment of new carbon into soil due to eCO$_2$ increases turnover rate of soil carbon (Van

Groenigen et al., 2014; Van Groenigen et al., 2017). Within a certain PFT, the variation of $\beta_{cpool}$ across different geographical

locations is usually smaller than that of $\beta_{NPP}$. While the greater variation of $\beta_{cpool}$ than that of $\beta_{NPP}$ across different C$_3$

PFTs in C-only simulation suggests other processes such as different carbon allocation patterns, plant carbon turnover, and the

soil carbon dynamics of various PFTs, are responsible for the additional divergence. In nutrient-coupled simulations, the

variations of $\beta_{cpool}$ across different C$_3$ PFTs are only slightly larger than those of $\beta_{NPP}$, indicating that nutrients do not bring

much differential effects on carbon turnover processes for different PFTs.

### 4.3 Implication for understanding $\beta$ in other models

Although we analyze a single land-surface model in detail, the patterns of and mechanisms underlying the variability of $\beta$ we

found may be generally applicable to other models. The basic Farquhar photosynthesis model and two-leaf scaling scheme in

the CABLE model are shared by many land surface models. Some models use variants of Farquhar photosynthesis model such

as co-limitation approach described by Collatz et al. (1991). Inflection point from Rubisco- to RuBP- limited processes is an

important control of the absolute photosynthetic response to eCO$_2$ (Rogers et al., 2017). However, the relative photosynthetic

responses for different ecosystems will converge to a small range because the normalized photosynthetic response to eCO$_2$

only depends on estimates of intercellular CO$_2$ concentration ($C_i$), Michaelis-Menten constants ($K_c$, $K_o$) and CO$_2$

compensation point ($\Gamma_*$), and the relative photosynthetic responses are similar for either Rubisco- or RuBP-limited

photosynthesis (Luo et al., 1996; Luo and Mooney, 1996). Soil moisture availability is another key constraint on photosynthetic

response. Water stress on plants is generally alleviated under eCO$_2$ due to reduced stomatal conductance (Leuzinger and Körner,

2007; Fatichi et al., 2016). Different models simulate diverse levels of water stress on productivity (De Kauwe et al., 2017).

Water stress is simulated in many models to regulate stomatal conductance (Rogers et al., 2017; Wu et al., 2018). For example,

the CABLE model represents water stress by an empirical relationship based on soil texture and limits the slope of the coupled relationship between photosynthesis rate and stomatal conductance as Eq. (S11). The influence of water stress is reflected by $C_i$. Synthesis of many empirical study results and our results in this study all show that ratio of $C_i$ to $C_a$ is relatively constant, probably due to homeostatic regulations through photosynthetic rate and stomatal conductance (Pearcy and Ehleringer, 1984; Evans and Farquhar, 1991). Wong et al. (1979) showed plant stomata could maintain a constant $C_i/C_a$ ratio across wide range of environmental conditions, including water stress condition. Land surface models might simulate relatively constant $C_i/C_a$ ratios under water stress as well since photosynthesis and stomatal conductance are theoretically depicted based on experimental results. Moreover, Luo and Mooney (1996) found that changing $C_i/C_a$ ratio from 0.6 to 0.8 caused less than variation of 0.08 in sensitivity of leaf photosynthesis to a unit of increase in $C_a$. $K_c$, $K_o$ are variable among species, but only slightly affect leaf-level response (Luo and Mooney, 1996). Different leaf temperature will exert limited influence on the variability of leaf-level $\beta$ as we discussed above. Therefore, leaf-level $\beta$ values for different C$_3$ PFTs are more likely to converge in other land surface models.

A recent study used 16 crop models to simulate rice yield at two FACE sites (Hasegawa et al., 2017). These models have diverse representations of primary productivity. Their results showed that the variation of yield response across models was not much associated with model structure or magnitude of primary photosynthetic response to eCO$_2$, but was significantly related with the estimations of leaf area. This is consistent with our conclusion and highlights the great need to improve prognostic LAI modeling. Other land-surface modelling groups may benefit from a similar analysis to identify major causes of variability of $\beta$ across the hierarchical levels from biochemistry to land carbon storage. Candidate causes that can make substantial contributions to the variability include changes in changes in leaf area index, changes in carbon use efficiency and changes in land carbon residence times. If modelling groups can add leaf-level diagnostics in the next inter-model comparison project, it will greatly help disentangle the uncertainty of concentration-carbon feedback.

## 5. Conclusions

Exploring the variability of $\beta$ at different hierarchical levels within and across different $C_3$ PFTs helps unravel model mechanisms that govern terrestrial ecosystem responses to elevated $CO_2$. Our study shows that the sensitivities of biochemistry and leaf-level photosynthesis to $eCO_2$ are very similar within and across $C_3$ PFTs in C-only, C-N and C-N-P simulations of CABLE, in accordance with previous theoretical analysis. While $\beta$ values of GPP, NPP and ecosystem carbon storage diverge primarily because the sensitivities of LAI significantly differ within and across different PFTs in all simulations. After decomposing $\beta$ into photosynthetic and LAI components, we find LAI contributes more than photosynthesis to the magnitudes and trends of model responses. Our results indicate that processes related to LAI need to be better constrained with results from experiments and observations in order to better represent the responses of ecosystem carbon cycle processes to changes in $CO_2$ and climate.

## Acknowledgements

We acknowledge CSIRO supercomputing facility (pearcey) to run CABLE model. We thank Tsinghua University for providing Scholarship for Overseas Graduate Studies. This paper is financially supported by the National Key R&D Program of China (2017YFA0604604).

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

**Table 1. The ratio of intercellular $CO_2$ concentration ($C_i$) to atmospheric $CO_2$ concentration ($C_a$) for different C3 PFTs, mean and coefficient of variation (CV) across these PFTs of $C_i/C_a$ in C-only, C-N, C-N-P simulations of CABLE under RCP8.5 scenario. Values for shaded leaves are in brackets. Abbreviations are the same as Figure 1.**

| PFT | $C_i/C_a$ (C-only) sunlit(shaded) | $C_i/C_a$ (C-N) sunlit(shaded) | $C_i/C_a$ (C-N-P) sunlit(shaded) |
|---|---|---|---|
| ENF | 0.69(0.74) | 0.66(0.74) | 0.66(0.79) |
| EBF | 0.70(0.76) | 0.65(0.78) | 0.65(0.78) |
| DNF | 0.64(0.68) | 0.61(0.67) | 0.61(0.67) |
| DBF | 0.67(0.73) | 0.63(0.73) | 0.64(0.73) |
| SHB | 0.70(0.73) | 0.65(0.73) | 0.65(0.73) |
| C3GRAS | 0.69(0.73) | 0.63(0.73) | 0.63(0.73) |
| TUN | 0.68(0.71) | 0.63(0.71) | 0.63(0.71) |
| Mean | 0.68(0.73) | 0.64(0.73) | 0.64(0.73) |
| CV | 0.03(0.03) | 0.03(0.05) | 0.03(0.06) |

**Table 2. Coefficients of variation of $\mathcal{L}$, $\beta_p$, $\beta_{\text{GPP}}$, $\beta_{\text{NPP}}$ and $\beta_{cpool}$ across different geographical locations within each C₃ PFT at the year of 2023 in CABLE-C only simulation. The two numbers in the same unit are for sunlit leaves and shaded leaves respectively. Values for shaded leaves are in brackets. Abbreviations are the same as Figure 1.**

| PFT | CV($\mathcal{L}$) sunlit(shaded) | CV($\beta_p$) sunlit(shaded) | CV($\beta_{\text{GPP}}$) | CV($\beta_{\text{NPP}}$) | CV($\beta_{cpool}$) |
|---|---|---|---|---|---|
| ENF | 0.27(0.30) | 0.41(0.42) | 1.77 | 2.68 | 1.40 |
| EBF | 0.26(0.29) | 0.24(0.28) | 0.55 | 0.54 | 0.60 |
| DNF | 0.26(0.28) | 0.25(0.28) | 1.19 | 1.20 | 0.30 |
| DBF | 0.39(0.38) | 0.42(0.37) | 1.29 | 1.42 | 0.85 |
| SHB | 0.33(0.32) | 0.30(0.49) | 1.24 | 1.23 | 1.12 |
| C3GRAS | 0.38(0.34) | 0.35(0.34) | 1.12 | 1.10 | 0.98 |
| TUN | 0.35(0.34) | 0.36(0.37) | 1.86 | 1.85 | 1.92 |

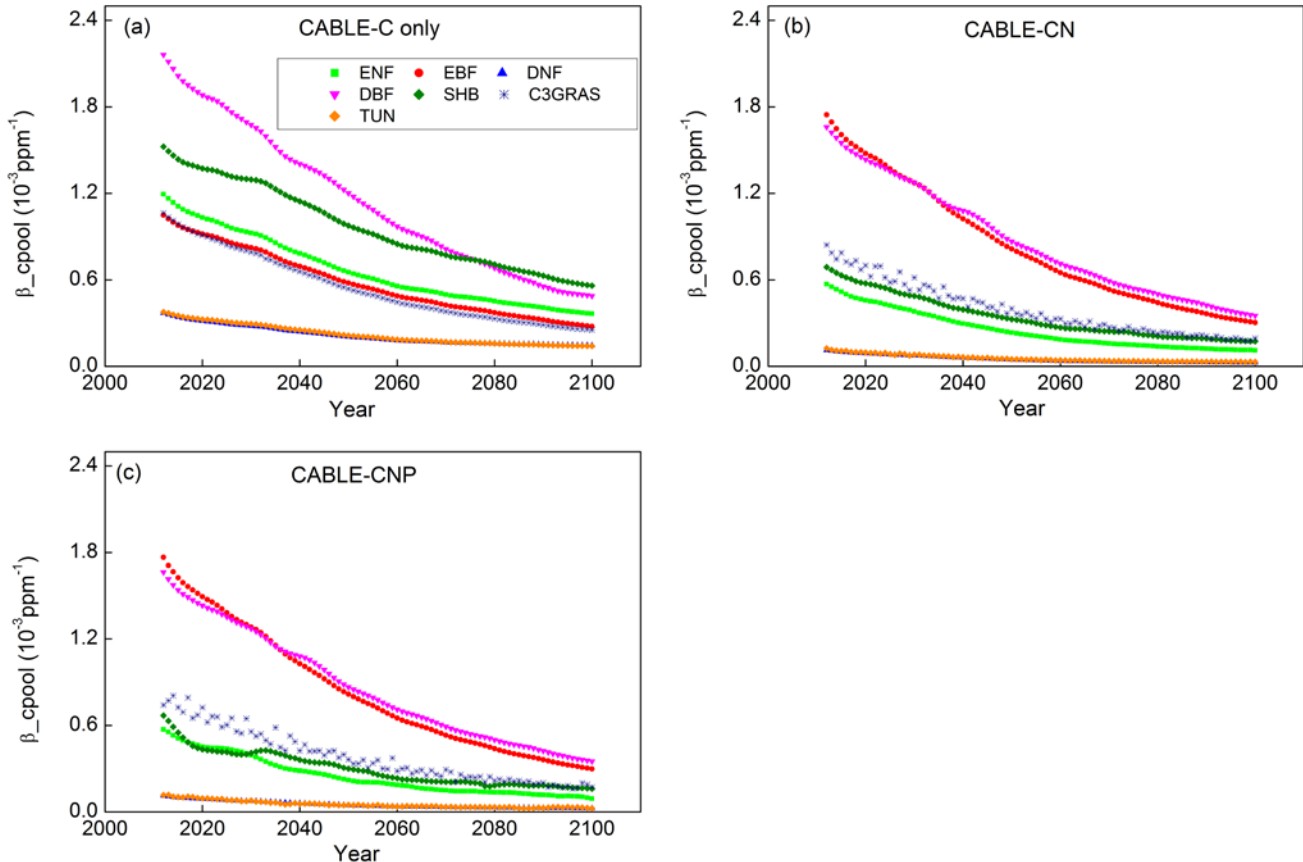

**Figure 1. Temporal trends of $\beta_{cpool}$ from 2011 to 2100 for C₃ PFTs from CABLE-C only (a), CABLE-CN (b), and**
**CABLE-CNP (c) simulations. $\beta_{cpool}$ values for different C₃ PFTs all decline with time from 2011 to 2100 under**
**RCP8.5 scenario, but the magnitudes of $\beta_{cpool}$ differ across them in all simulations. In C-N and C-N-P simulations,**
**magnitudes of $\beta_{cpool}$ are reduced compared with those in C-only simulation for all C₃ PFTs except evergreen**
**broadleaf forest. ENF, Evergreen Needleleaf Forest (light green squares); EBF, Evergreen Broadleaf Forest (red**
**circles); DNF, Deciduous Needleleaf Forest (dark blue triangles); DBF, Deciduous Broadleaf Forest (pink triangles);**
**SHB, Shrub (dark green diamonds); C3GRAS, C₃ grass (dark blue stars); TUN, tundra (orange diamonds).**

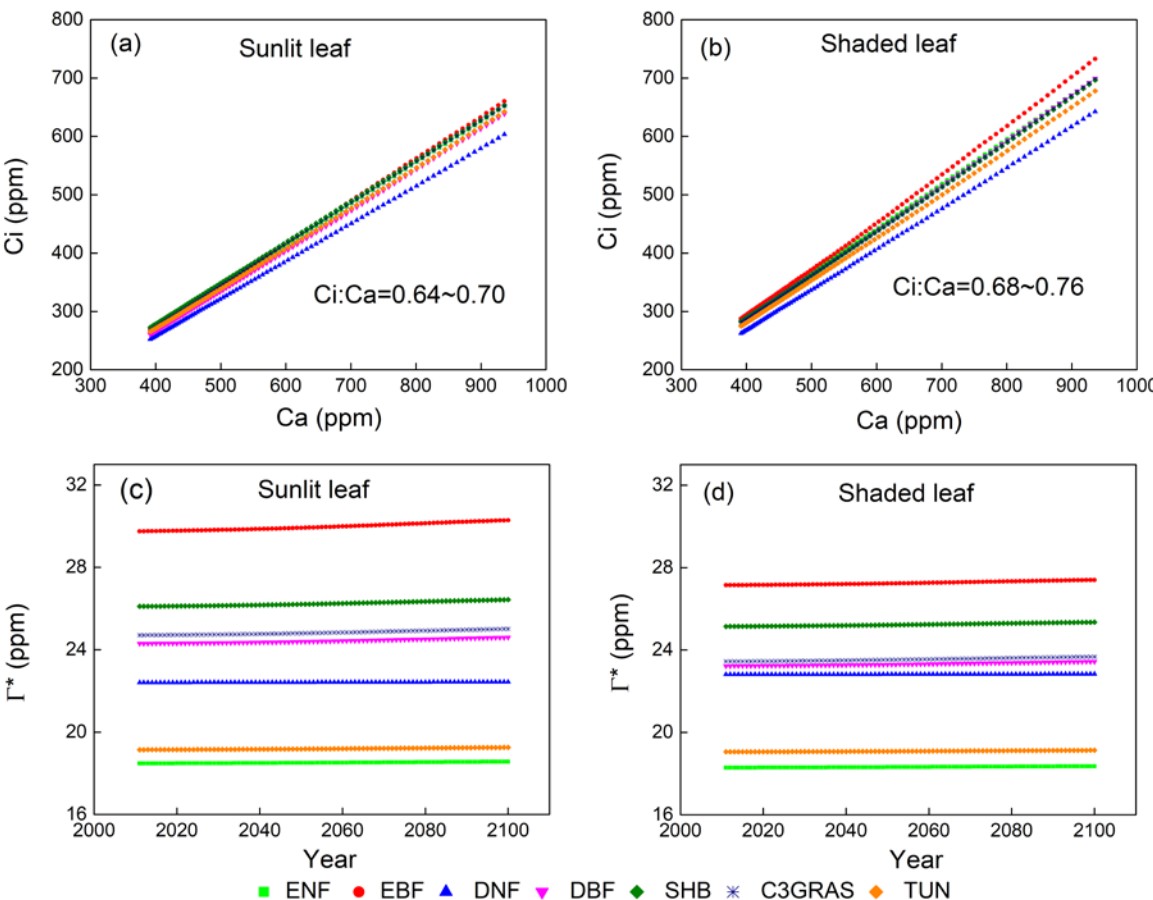

**Figure 2. Responses of yearly intercellular CO₂ concentration ($C_i$) to eCO₂ of a single sunlit leaf (a) and shaded leaf (b) for C₃ PFTs from CABLE-C only simulation. Temporal trends of CO₂ compensation point in the absence of day respiration ($\Gamma_*$) for sunlit leaf (c) and shaded leaf (d) from 2011 to 2100 from CABLE-C only simulation. The ratio of $C_i$ to $C_a$ ($C_i/C_a$) is approximately constant with eCO₂ for each PFT and varies little across PFTs. $\Gamma_*$ values vary across different PFTs, but do not change over time for each PFT. Abbreviations and symbols are the same as Figure 1.**

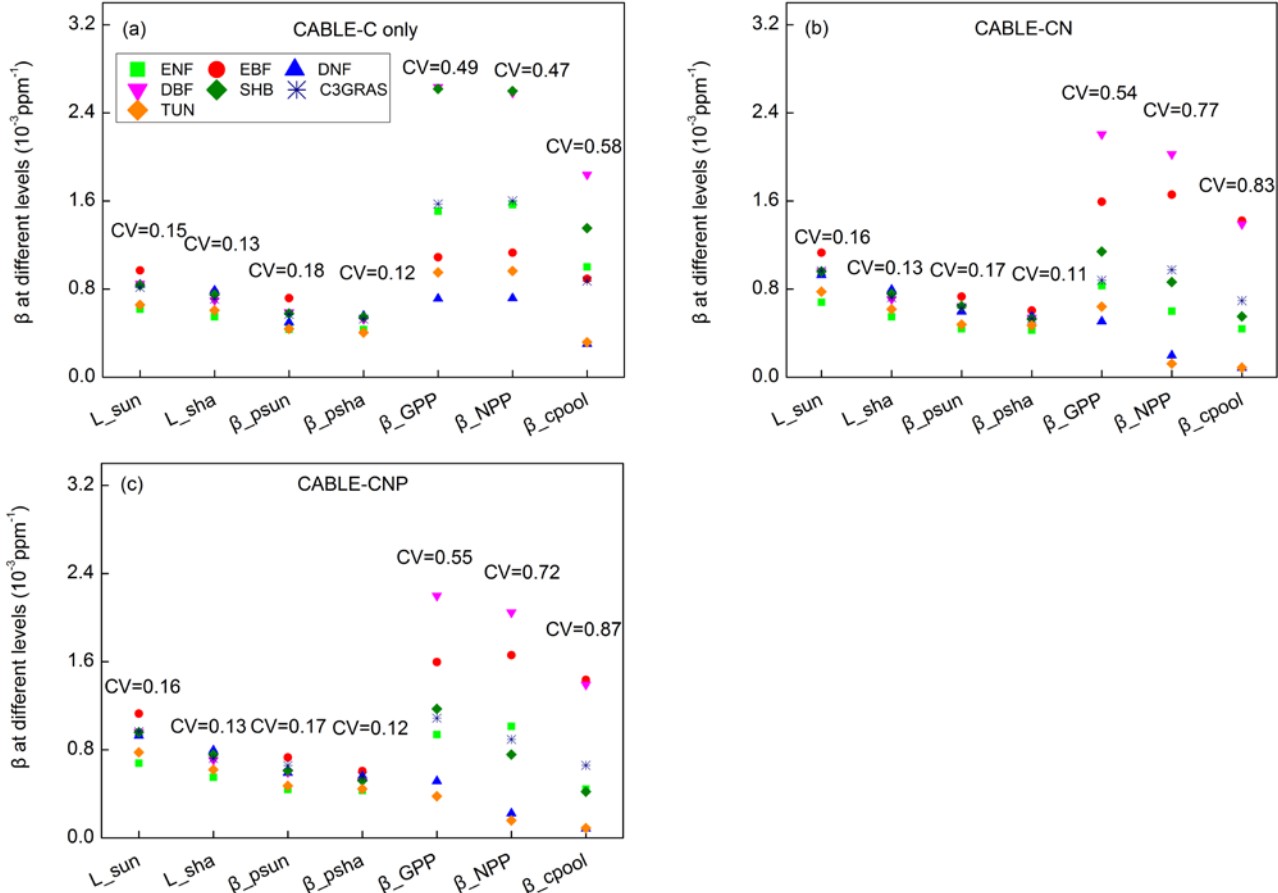

**Figure 3. Biome-level $\beta$ values at different levels at the year 2023 from CABLE-C only (a), CABLE-CN (b), and CABLE-CNP (c) simulations. CV means coefficient of variation of biome-level $\beta$ across C$_3$ PFTs. $\beta$ values at biochemical ($\mathcal{L}_{sun}$ and $\mathcal{L}_{sha}$ for sunlit and shaded leaves) and leaf levels ($\beta_{p_{sun}}$ and $\beta_{p_{sun}}$) are very similar across PFTs, but greatly diverge at canopy level ($\beta_{GPP}$), and ecosystem levels ($\beta_{NPP}$ and $\beta_{cpool}$) in all simulations. Unlike in C-only simulation, $\beta_{NPP}$ diverges more than $\beta_{GPP}$ across different PFTs in nutrient-coupled simulations. Abbreviations and symbols are the same as Figure 1.**

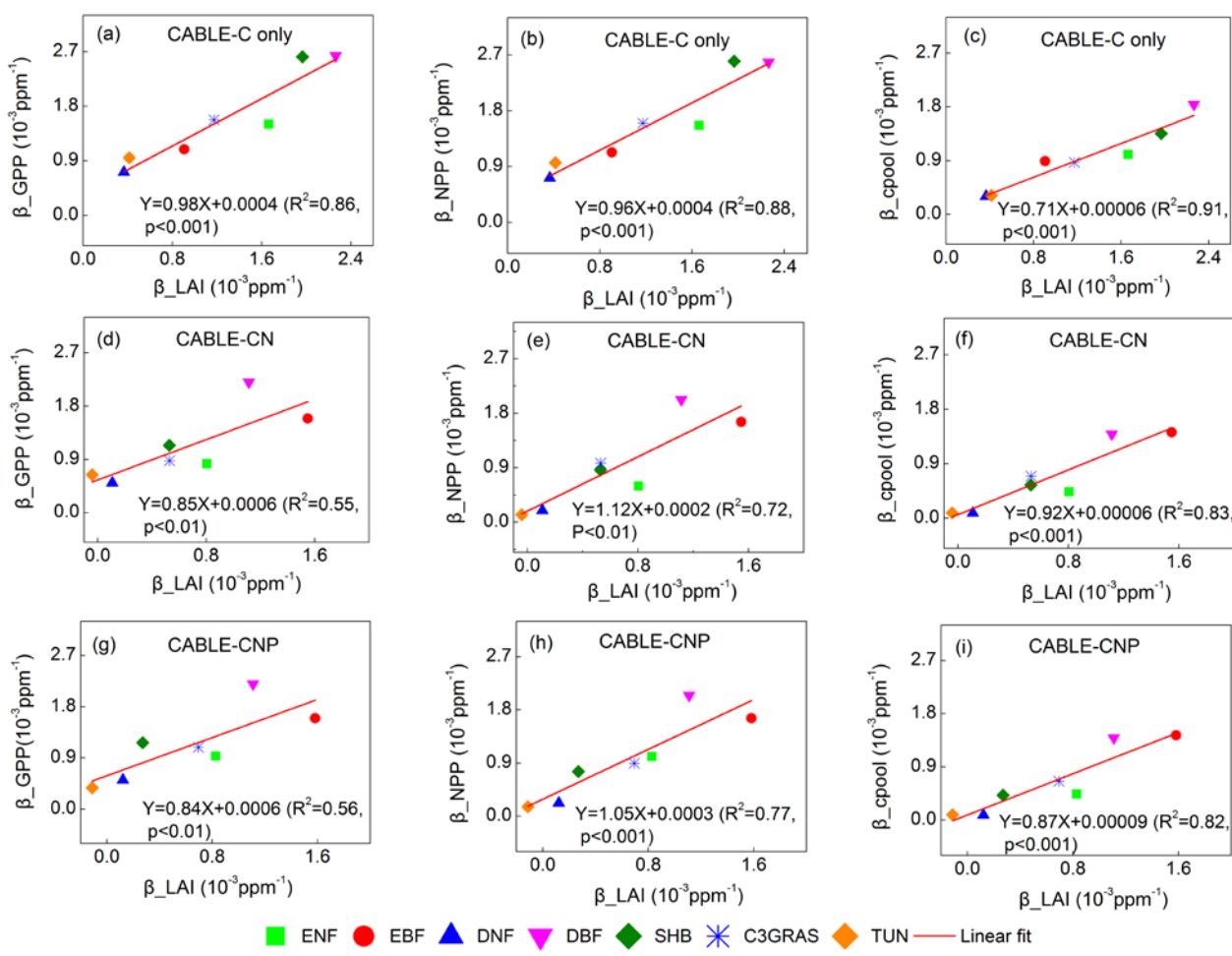

780

**Figure 4.** Correlations between $\beta_{GPP}$ and $\beta_{LAI}$, $\beta_{NPP}$ and $\beta_{LAI}$, $\beta_{cpool}$ and $\beta_{LAI}$ at the year 2023 across C₃ PFTs from CABLE C-only (a)~(c), CABLE-CN (d)~(f) and CABLE-CNP (g)~(i) simulations. $\beta_{GPP}$, $\beta_{NPP}$ and $\beta_{cpool}$ all have significant linear correlations with $\beta_{LAI}$ in all simulations. Abbreviations and symbols are the same as Figure 1.

785

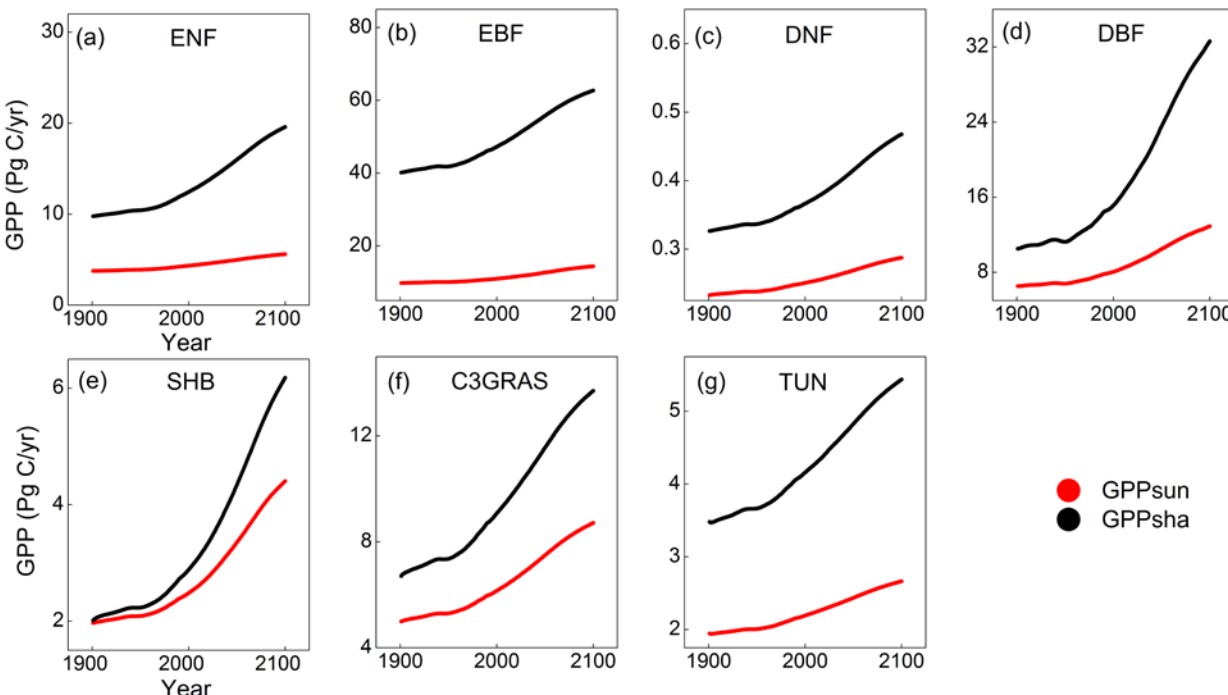

**Figure 5.** Temporal trends of $GPP_{sun}$ (red points) and $GPP_{sha}$ (black points) for $C_3$ PFTs from 1901 to 2100 from CABLE C-only simulation. $GPP_{sha}$ is higher than $GPP_{sun}$ for all PFTs. With significant increase of $CO_2$ concentration from 2011, $GPP_{sha}$ responds more drastically than $GPP_{sun}$.

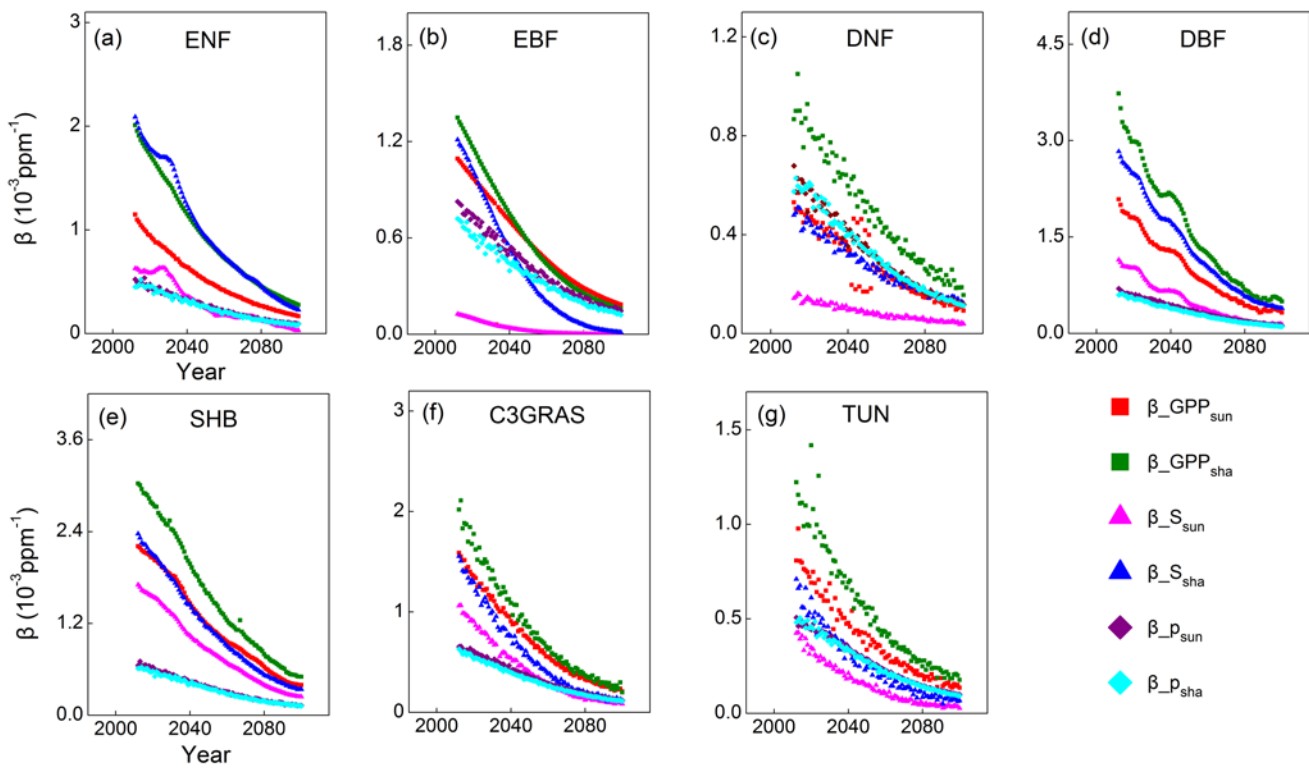

**Figure 6. Temporal trends of** $\beta_{\text{GPP}_{sun}}$ **(sensitivity of sunlit leaf GPP; red squares),** $\beta_{\text{GPP}_{sha}}$ **(sensitivity of shaded leaf GPP; green squares),** $\beta_{S_{sun}}$ **(sensitivity of scaling fatcor for sunlit leaf; pink triangles),** $\beta_{S_{sha}}$ **(sensitivity of scaling fatcor for shaded leaf; dark blue triangles),** $\beta_{p_{sun}}$ **(photosynthetic response for sunlit leaf; purple diamonds) and** $\beta_{p_{sha}}$ **(photosynthetic response for shaded leaf; sky blue diamonds) for C₃ PFTs from CABLE C-only simulation. The sensitivities of** $\text{GPP}_{sun}$ **and** $\text{GPP}_{sha}$ **tend to approach zero through time because the decomposing factors** $\beta_{p_{sun}}$, $\beta_{p_{sha}}$, $\beta_{S_{sun}}$ **and** $\beta_{S_{sha}}$ **all decline with time.** $\beta_{S_{sha}}$ **determines the magnitudes and trends of** $\beta_{\text{GPP}_{sha}}$ **for almost all PFTs.**

805