# Peer review of "Leaf Area Index identified as a major source of variability in modelled $CO_2$ fertilization"

_Biogeosciences, 2018_

## Referee Comment (RC1) · M. G. De Kauwe (Referee) · 16 May 2018

Li et al. use the CABLE model to explore the role of LAI in variability in the CO2 fertil-isation response. The analysis has some interesting aspects which I'm sure will be of interest to the modelling community, in particular I thought fig 5 was interesting. However, I think the manuscript could be carefully revised for greater impact and insight. I have a number of specific points below but also 4 key issues with the analysis as presented:

1. I don't understand the logic of using a model which simulates N and P cycles and then switching this functionality off to understand the CO2 fertilisation response? In my eyes, this is one of the great strengths of this model. So to not compare C against N

and P, or C against N, is a missed opportunity. Whilst I'm realistic enough to envisage the authors won't rethink this strategy, I do feel this requires some further justification.

2. It is stated that CABLE is largely RuBP-limited (line 179) and this point is given no further analysis. This is interesting and it isn't clear why this would be the case? Do the authors envisage that this is also true of other models? I would suggest it isn't but would be keen to read the authors thoughts on this. Surely this shapes the analysis (responsiveness to CO2)? So it warrants more than a single sentence that simply says "not shown" ...

3. The paper is about CABLE but surely the aim is to make the result general (otherwise the title would have the word CABLE...)? However, I wonder if I was developing JULES or CLM, (etc) what my take home messages would be? The authors urge other modelling groups to repeat their analysis, but could they also make suggestions as to the implications for other modelling groups? How do these results help us to understand model responses to CO2? The CMIP5 concentration-carbon feedback factor?

I didn't take much in the way of insight from the current section on this topic, i.e. section 4.3. For example, the authors assert that "It can be inferred that normalized leaf-level ðİŽ¡ values would diverge little across different land surface models as long as they use ...". Is that true? If the models had different levels of water stress (which they almost always do) they would get very different values of Ci even with the same model assumptions. As the authors also show, leaf temperature affects gamma_star, so I see no reason to assume that models would predict similar leaf temperatures. Leaf temperature itself is dependent on a whole range of assumptions. I've never seen any evidence that models with different architectures, with different assumptions about leaf-to-boundary conductance, etc, would predict similar leaf temperatures. If the authors disagree they should support these assumptions. The authors cite the Hasegawa et al study as an example of a consistent result of their conclusion. But wouldn't a number of the other model CO2 papers that point to marked divergence argue otherwise. My sense is their conclusion here is too simplistic.

The authors argue for the importance of LAI but don't really consider the role of allocation or turnover in great detail. Surely this is the key reason different models arrive at different LAI values? Even if you ignore changes in allocation/turnover due to CO2, this impacts on the scaling terms that the authors focus on.

4. The results are considered on a PFT level, but presumably they vary in interesting ways within a PFT (i.e. in space). Would this be worth showing or exploring further?

Specific comments =================

- Line 43: Could you explain the CO2 fertilising effect further? The text as written expects the casual reader has significant background knowledge for the second sentence of your manuscript.

- Line 48: 4 or 4.5? What does that mean, do you mean 4 to 4.5? How can it be OR?

- Line 49: the reference to the Smith et al. paper ignores a technical comment on this paper: De Kauwe et al. (2016). Satellite based estimates underestimate the effect of CO2 fertilization on net primary productivity. Nature Climate Change, 6, 892-893. This is important as the authors are using this study to leverage their question. See also point on line 340.

- Line 51: it isn't "reality" - the satellite estimates are also model estimates.

- Line 54: "increasing temperature in models" why is temperature being introduced as a factor here? Isn't the focus solely on the CO2 fertilisation effect rather than the than carbon-climate feedback factor? There are further studies cited in this paragraph which should be removed if the focus of this paper does not consider the carbon-climate feedback factor.

- Line 67: Despite models using apparently similar photosynthesis models, Rogers et al. (A roadmap for improving the representation of photosynthesis in Earth system models. New Phytologist, 213, 22-42.) showed some important differences. It would be worthwhile highlighting this study in the context of the section of the text.

- Line 72: what does carbon storage have to do with this sentence?

- Line 76/7: seems a narrow characterisation of the literature, the De Kauwe et al. 2014 study that the authors cite, explored these issues in depth.

- Line 81: Why would a high "basic" (delete basic) NPP necessarily lead to tropical regions having the highest stimulation by CO2? Wouldn't the opposite be expected? These regions have a high LAI and so would predominantly be light-limited and so have a more limited capacity to respond to CO2? Either way, the authors need to expand on this assertion.

- Line 89: Improved on what?

- Line 124: The assumption that Jmax25 = 2 x Vcmax25. Did the authors consider varying this assumption? Other models would make quite different assumptions about this ratio.

- Line 155: is there a citation, web link for "Community Climate System Model (CCSM) simulations"

- Line 168: the definition of S (line 171) needs to be moved up to this line.

- Line 215: just to clarify when the authors say total carbon storage - do they mean the soils too? Or just the plant? Or just the foliage pool? The equation isn't very clear. This also makes Fig 1 hard for me to interpret as I'm unclear what is being shown, I'm going to assume it is total plant carbon...

- Fig 1. Does it make sense to normalise these PFT lines? The authors say they decline but the magnitudes differ, the point is that the initial starting points are different too. This makes it hard for the eye to gauge.

- As a general comment the results need work, particularly in terms of transition text. For example 3.1 talks about the temporal trend in Bcpool and then switches immediately to the Ci/Ca ratio in 3.2? It is hard to follow the logic of the transition, is there is

meant to be any connection for the reader?

- What is the point of Fig. 2? It isn't clear what this figure has to do with the story of the paper?

- The text around line 261 which refers to Fig 4 could do with further explanation. I personally don't find this particularly surprising, but the reader isn't offered much as the way of explanation. Presumably the change in slope as you move from B_GPP to B_NPP relates to respiration assumptions and then to B_cpool, allocation/turnover assumptions? I think the authors could go further in assisting the reader with interpretation. As currently written, the text simply highlights that the slope changes.

- I think figure 5 is very interesting.

- Line 290: I think this discussion of Fig S5 is interesting but I'm not sure I follow the interpretation? The LAI is the emergent outcome of the model assumptions - 1 leaf, 2leaf, multi-layer. Of course this assumption will lead to differences? But why you do the analysis on the leaf-level? Surely you're interested in the emergent outcome - the LAI. Most likely I simply misunderstood this point but I think it could also be explained further as it seems like an important point the authors are making.

- Line 295: I don't fully follow that interpretation? Your differences in Ci/Ca were small across PFTs? And the differences in leaf temp would be expected between PFTs? Certainly, fig 2 doesn't show any within PFT variation.

- Line 362: This is an assumption of the model and might not necessarily be true!

Technical corrections =====================

- Abstract: "vegetation types is 0.15-0.13", presumably you meant 0.13 to 0.15? Also, why don't the other variables (e.g. BetaGPP) have ranges too?

- First line of the introduction, makes no sense. You can't start a sentence with Terrestrial carbon sink and then a comma.

- Line 45: In Coupled -> In the Coupled

- Line 138: In CABLE model -> in the cable model

---

## Referee Comment (RC2) · Anonymous Referee #2 · 29 May 2018

Review: Leaf Area Index identified as a major source of variability in modelled CO2 fertilization

Authors: Li, Qianyu, et al.

**Synopsis:**

In this paper, the authors run CABLE for seven C3 vegetation types, without nutrient cycling, and calculate CO2 fertilization for the RCP 8.5 scenario. CCSM simulations from 1901 to (the paper says 1910; I assume they mean 2010) holding carbon-climate feedbacks constant (driving the model with the averaged meteorology-I'm guessing average annual cycle, although the authors do not say) and feeding CABLE increasing CO2 concentration from the CCSM RCP 8.5 results.

They find that CO2 fertilization differs between PFTs, and decreases with time during the period 2011-2100. Fertilization is relatively constant both between PFTs and when the calculation is made on a per-unit leaf level, and shows much larger diversity both across PFTs and when the CO2 fertilization is calculated on a unit-leaf vs. integrated canopy basis. The authors close with the claim that simulated LAI is critical to the calculation of CO2 fertilization in climate simulations.

**Review:**

I have 2 major problems with this paper. Either one by itself, I believe, is fatal, but taken together I cannot make any recommendation for this paper other than rejection.

**Problem #1:** There is a rich body of literature from the FACE experiments that claims, pretty much unequivocally, that nutrient cycling and/or limitation becomes more and more important to CO2 fertilization as CO2 concentrations rise. Yet, in this experiment CABLE is run with nutrient cycling turned off!

Coskun et al. (2016) and references therein has a nice summary of both Free-Air CO2 Enrichment (FACE) as well as Open-Top Chamber (OTC) experiments. Smith et al. (2015) discusses the divergence between multiple models and a satellite-derived product that underscores the importance of the interaction between nutrient cycling and CO2 fertilization. Many of these studies focus on N limitation, although some research has indicated that P limitation is a factor as well (e.g. Hasegawa et al., 2016). These, and other studies, all conclude that understanding of CO2 fertilization requires taking nutrients into account.

Schimel et al (2014) demonstrate that, depending on the model, inversions can disagree on the location of the dominant location of the terrestrial sink (tropics or northern hemisphere). Quantification of the fertilization sink in a single model may bring information to bear on this uncertainty, as CO2 fertilization is really the only way to get a sink in the tropics, while other carbon-climate feedbacks (season lengthening, woody encroachment, forest regrowth) may also be possible at higher latitude. Studies that quantify CO2 fertilization may be able to shed

light on this discrepancy, although the interaction between higher levels of atmospheric CO2 and nutrients must be considered.

I have to confess that I was very surprised when I read that the authors ran the version of CABLE without nutrient cycling included. I am not a FACE 'expert', but even I am aware of the amount of research that has concluded that nutrient cycling is critical to understanding ecosystem-level response to higher atmospheric CO2. I found it very suspicious that nutrients were excluded from the study. Why, when there is this large body of work demonstrating the nutrient cycling is critical to understanding CO2 enrichment, would nutrients be turned off in the model? The authors claim that nutrients were turned off for 'simplicity', but the obvious answer, and one that I suspect to be the truth, is that the authors did run CABLE with nutrient cycling, and model pathology and/or unrealistic results ensued.

It may have been possible to evaluate a nutrient run, even if the results were unrealistic, and evaluate how atmospheric CO2 levels and nutrients interact in CABLE. The results may have provided an opportunity to evaluate or comment on the divergence of models in their predictions of atmospheric CO2 levels and source/sink strength (e.g. Friedlingstein et al., 2006, 2014). By not including the critical nutrient interaction, I'm not sure that the results presented here give the reader any insight into how ecosystems might realistically respond to increasing future CO2 levels in the atmosphere.

**Problem #2:** Without carbon-climate feedbacks and nutrient cycles, I don't think a model actually has to be run to determine CO2 fertilization. You can probably perform the calculation directly from the equations in the code. Between models there will be some differences:

- Is the model an enzyme-kinetic model (Farquhar et al., 1980; Michaelis-Menten kinetics), or light-response (e.g. VPRM, Mahadevan et al., 2008)?
- how is stomatal conductance calculated? Does it use Ball-Berry, with a dependence on relative humidity, or Leuning, which uses VPD? How is transpiration coupled to photosynthesis?
- What are the parameter values for Vcmax for a given PFT?
- What determines phenology? Is allocation static, or, if it is dynamic, how does it change during the year and in response to what?

I believe it would be possible to determine the constraints on CO2 fertilization for a suite of models without actually running any of them.

It is axiomatic that leaf-to-canopy scaling (LAI) is critical to total CO2 fertilization amount. Every model that I am aware of calculates biophysics on a per-unit-are basis and then scales to the canopy level either by summing over sunlit/shaded leaves (and PFTs) or integrating from leaf to canopy scale along the lines of Sellers (1985, 1992)(OK, a gap model like ED2 may be a little different). Canopies with an LAI close to 1 (think of grasslands) will not see much difference from unit- to canopy-scale, more dense canopies (like forests) will.

If there is a large divergence between models in LAI (and GPP) for a given PFT, or if there is a large trend in one model's LAI for a given PFT during a climate run, then these might be valid topics of analysis. Finding that LAI is critical to canopy-level CO2 fertilization (without nutrients being considered) does not really bring anything new to the field.

Sunlit and shaded leaf partitioning is fairly well-constrained and sunlit LAI can never get much above 1 to 1.5 or so even under the most direct-sun conditions. Solar angle and leaf angle distribution make it possible to exceed an LAI value of one. I know that CLM has had issues with shade leaf LAI becoming excessively large. The authors do not discuss total LAI in CABLE during their fertilization runs, and this makes me suspicious-if their shade-leaf LAI is becoming unrealistically large, that might be a reason why fertilization strength decreases with time; increase in the amount of sunlit leaf may result in large change in GPP, but once sunlit LAI is filled, any additional canopy growth will be as shade LAI, and GPP increase will be attenuated.

I just don't think there's anything new here. Without nutrient cycling the CO2 fertilization results don't have much meaningful application, and the fact that leaf-to-canopy scaling is important has been known for a long time.

**Specific comments:**
- English prose and grammar, while readable, need attention. There are multiple places, too many to list, where errors exist.
- There is no explanation for what eCO2 is (elevated CO2). Don't assume all your readers know the definition.
- There is no definition of 'gamma' either.
- In many of the equations the equals sign is obscured. More effective spacing will make these equations easier to read.

**References**

Coksun, D., D.T. Britto, H.J. Kronzucker, 2016: Nutrient constraints on terrestrial carbon fixation: The role of nitrogen. Journal of Plant Physiology, 203, 95-109, http://dx.doi.org/10.1016/j.jplph.2016.05.016.

Farquhar, G.D., S. von Caemmerer and J.A. Berry,1980. A biochemical model of photosynthetic $CO_2$ assimilation in leaves of C3 species. Planta, 149, 78-90.

Friedlingstein, P., et al., 2006: Climate-Carbon cycle feedback analysis results from C4MIP Model Intercomparison. Journal of Climate, 19, 3337-3353.

Friedlingstein P., et al., 2014: Uncertainties in CMIP5 climate predictions due to carbon cycle feedbacks. Journal of Climate, 27, 511-526, doi:10.1175/JCLI-D-12-00579.1.

Hasegawa, S., C.A. MacDonald, S.A. Power, 2016: Elevated carbon dioxide increases soil nitrogen and phosphorus availability in a phosphorus-limited Eucalyptus woodland. Global Change Biology, 22, 1628-1643, doi:10.1111/gcb.13147.

Mahadevan, P., et al., 2008: A satellite-based biosphere parameterization for net ecosystem CO2 exchange: Vegetation Photosynthesis and Respiration Model (VPRM). Global Biogeochemical Cycles, 22, GB2005, doi:10.1029/2006GB002735.

Schimel, D., B.B. Stephens, J.B. Fisher, 2014: Effect of increasing CO2 on the terrestrial carbon cycle. Proceedings of the National Academy of Sciences, www.pnas.org/cgi/doi/10.1073/pnas.1407302112.

Sellers, P.J., 1985: Canopy Reflectance, Photosynthesis and Transpiration. International Journal of Remote Sensing, 6(8), 1335-1372.

Sellers, P.J., J.A. Berry, G.J. Collatz, C.B. Field and F.G. Hall, 1992: Canopy Reflectance, Photosynthesis, and Transpiration. III. A Reanalysis Using Improved Leaf Models and a New Canopy Integration Scheme. Remote Sensing of Environment, 42, 187-216.

Smith, W.K., et al., 2015: Large divergence of satellite and Earth system model estimates of global terrestrial CO2 fertilization. Nature Climate Change, 6, 306-312, doi:10.1038/NCLIMATE2879.

---

## Author Comment (AC1) · 21 Aug 2018

Dear editor,

We appreciate the insightful comments on our manuscript by all reviewers, and will make substantial revisions to improve this manuscript. We hope that our responses below address the concerns raised by reviewers. The major proposed changes are: We will include results from carbon-nitrogen (C-N) coupled and carbon-nitrogen-phosphorus (C-N-P) coupled simulations of CABLE to study how the CO2 fertilization effects ($\beta$ factor) at different levels will change with nutrient limitations for different plant functional types. Our results from C-N and C-N-P coupled simulations support our pervious conclusion with C only simulations. We will clarify the motivation and contribution

of our study as the reviewers suggested in our revised manuscript. We will expand the scope of our work to attract more readers and carefully correct language in our revised manuscript.

Reviewer 1: Li et al. use the CABLE model to explore the role of LAI in variability in the CO2 fertilisation response. The analysis has some interesting aspects which I'm sure will be of interest to the modelling community, in particular I thought fig 5 was interesting. However, I think the manuscript could be carefully revised for greater impact and insight. I have a number of specific points below but also 4 key issues with the analysis as presented:

Response: We thank the reviewer for the positive comments. His suggestions are very important for improving our manuscript.

Reviewer 1: 1. I don't understand the logic of using a model which simulates N and P cycles and then switching this functionality off to understand the CO2 fertilisation response? In my eyes, this is one of the great strengths of this model. So to not compare C against N and P, or C against N, is a missed opportunity. Whilst I'm realistic enough to envisage the authors won't rethink this strategy, I do feel this requires some further justification.

Response: We appreciate the reviewer's critical comments. This concern has been raised by two reviewers. The reason why we didn't originally include nitrogen and phosphorus cycles in our previous study is that we tried to find the most important factor causing the variations of $\beta$ within and across different vegetation types with minimal confounding effects of other processes.

However, we totally agree with the reviewers that carbon-nutrient interactions should be considered when studying $\beta$ effects. The respective effects of N and P can be calculated through the difference in the carbon uptake between C-N and C-only or C-N-P and C-N coupled simulations. Wang et al. (2012) and Zhang et al. (2013) provided details explaining how nutrient limitations are incorporated into carbon cycle in CASA-CNP

module in the CABLE model. In brief, NPP is calculated as: NPP=(GPP(L,V_cmax (N_l),J_max (N_l))-$\sum \_i R\_mi(N\_i) - R\_g(N\_l/P\_l)) * x\_npup$(1)

Where L represents leaf area index, V_cmax and J_max are maximum carboxylation rate and maximum rate of electron transport of the top leaves, respectively, both are linearly dependent on leaf N (g N m-2) according to the relationships developed by Kattge et al. (2009) for different plant functional types. R_mi is maintenance respiration rates of plant tissue (i=leaf, wood and root) and contingent on nitrogen amount in each part of plant. R_g is growth respiration, which is described as a function of leaf nitrogen to phosphorus ratio. x_npup is the nutrient uptake limiting factor. x_npup will become less than 1 when the available nutrients (N or P) amount is less than the minimal amount of nutrient required by plants for a given NPP (Wang et al., 2010). Heterotrophic respiration (Rh) is limited by the mineral N pool required for microbial soil C decomposition (Wang et al., 2010). Net ecosystem productivity (NEP = GPP – Ra – Rh) is the amount of C that is either sequestered or lost from ecosystems, and is controlled by N and P availability via abovementioned C-N-P interactions.

Since effects of N and P on terrestrial carbon under CO2 fertilization in different regions in the CABLE model have been evaluated in Zhang et al. (2011), we will not elaborate on this point. Instead, we focus on the variations of $\beta$ values at different scales across C3 vegetation types. The new results are plotted in Fig. 1b and Fig. 1c (in this response letter) to show $\beta$ values at different levels for different vegetation types with carbon-nutrient interactions in this response letter. Our new results indicate that variations of $\beta$ factors at leaf level in C-N and C-N-P coupled simulations are as small as that in C-only simulation, because the normalization will eliminate the influence of nitrogen-related V_cmax and J_max in calculating leaf-level response (Luo et al., 1996), and estimates of intercellular CO2 concentration (C_i) with nutrient limitations are comparable to those without nutrient limitations. $\beta$ factors at canopy level ($\beta$_GPP) in C-N and C-N-P coupled simulations greatly diverge across vegetation types, which is similar to that with C-only simulation. However, unlike in C-only simulation, $\beta$_NPP

values are reduced for most C3 vegetation types and diverge more compared with $\beta\_GPP$ values in nutrient-coupled simulations, because the different nutrient-limiting effects on autotrophic respiration and plant growth as shown in Eq. (1) introduce additional variation across different vegetation types. Coefficients of variation of $\beta\_cpool$ in nutrient-coupled simulations exceed 0.8, larger than that in C-only simulation. It is noteworthy that in the current version of CABLE, P limitation is quite weak under present condition (Zhang et al., 2011). Therefore, the results of CABLE-CN are quite similar to those of CABLE-CNP.

We then found the linear relationships between $\beta\_GPP$ and $\beta\_LAI$, $\beta\_NPP$ and $\beta\_LAI$, $\beta\_cpool$ and $\beta\_LAI$ in C-only simulation (Fig. 2a∼2c in this response letter) still hold in the C-N and C-N-P coupled simulations (Fig. 2d∼2i in this response letter). From these results, our previous conclusion that LAI identified as a major source of variability in modelled CO2 fertilization is still valid under nutrient-limiting situations. We will add the above results into our revised manuscript.

Reviewer 1: It is stated that CABLE is largely RuBP-limited (line 179) and this point is given no further analysis. This is interesting and it isn't clear why this would be the case? Do the authors envisage that this is also true of other models? I would suggest it isn't but would be keen to read the authors thoughts on this. Surely this shapes the analysis (responsiveness to CO2)? So it warrants more than a single sentence that simply says "not shown" ...

Response: We thank the reviewer for this comment. It is an important prerequisite in our study. We agree it should be clarified in our manuscript. The formulation of leaf-level $\beta$ factors depends on the intercellular CO2 concentration (Farquhar et al., 1980). Generally, photosynthesis rate is RuBP-regeneration limited (limited by light) when CO2 concentration exceeds a certain level. And we coded a variable indicating which process (Rubisco activity, RuBP regeneration or sink) limits photosynthesis rate at each running step in the original CABLE code. Then we outputted this variable. We found photosynthesis rates are almost all limited by RuBP-regeneration process

globally since 2011 when CO2 concentration is 391 ppm. Then leaf-level biochemical $\beta$ factor can be expressed as an equation of intercellular CO2 concentration and CO2 compensation point. We didn't show the results because of the large volume of data (56560 model grids $\times$ 8760 hours in a year in total). We will clarify this in our revised manuscript.

Moreover, theoretical analysis by Luo and Mooney (1996) showed that leaf-level $\beta$ values are similar for either Rubisco- or RuBP-limited photosynthesis. We will also add this point in the revised manuscript.

Reviewer 1: The paper is about CABLE but surely the aim is to make the result general (otherwise the title would have the word CABLE...)? However, I wonder if I was developing JULES or CLM, (etc) what my take home messages would be? The authors urge other modelling groups to repeat their analysis, but could they also make suggestions as to the implications for other modelling groups? How do these results help us to understand model responses to CO2? The CMIP5 concentration-carbon feedback factor?

Response: We thank the reviewer for the suggestion to highlight the take-home messages more clearly. In the introduction and discussion part, we will clarify that it is the large uncertainty of concentration-carbon feedbacks produced by CMIP5 models that motivates our work. To understand the source of uncertainty, Koven et al. (2015) found the large uncertainty of equilibrium terrestrial carbon change in response to elevated CO2 (eCO2) across the CMIP5 models is mainly caused by the variation of change in productivity, rather than by the variation of change in turnover time. Although the authors suggested that it probably results from the unrealistic representations of allocation and mortality processes in the current generation of models, in-depth understanding of what causes the divergent $\beta\_NPP$ across models will still be conducive to narrow the large uncertainty of $\beta$ factors at ecosystem scale. Our study tried to understand $\beta$ factors from a more mechanistic way than previous studies by analyzing CO2 fertilization from leaf biochemistry to ecosystem levels in a land surface model. Although our

analysis was conducted with CABLE, we believe the results will be applicable to other models because most land surface models employ Farquhar photosynthesis model to represent leaf biochemical response to eCO2. The biochemical properties of Farquhar photosynthesis model have determined that the basic responses of C3 plants to eCO2 under a certain CO2 concentration are almost constant (Luo et al., 1996; Luo & Mooney, 1996). However, the leaf-canopy scaling methods, allocation schemes, vegetation dynamics and soil modules among models are divergent (Arora et al., 2013; He et al., 2016). Our analysis shed new lights on mechanisms underlying model-model differences in estimated $\beta$ factors and offers new diagnostics to be added in the next intermodal comparison project to help disaggregate the uncertainty of $\beta\_NPP$. We will add the above discussions in the revised manuscript.

Reviewer 1: I didn't take much in the way of insight from the current section on this topic, i.e. section 4.3. For example, the authors assert that "It can be inferred that normalized leaf-level ÃřËŹI ËŹ ZÂą values would diverge little across different land surface models as long as they use ...". Is that true? If the models had different levels of water stress (which they almost always do) they would get very different values of Ci even with the same model assumptions. As the authors also show, leaf temperature affects gamma_star, so I see no reason to assume that models would predict similar leaf temperatures. Leaf temperature itself is dependent on a whole range of assumptions. I've never seen any evidence that models with different architectures, with different assumptions about leaf –to-boundary conductance, etc, would predict similar leaf temperatures. If the authors disagree they should support these assumptions. The authors cite the Hasegawa et al study as an example of a consistent result of their conclusion. But wouldn't a number of the other model CO2 paperls that point to marked divergence argue otherwise. My sense is their conclusion here is too simplistic.

Response: Thank the reviewer for pointing out this issue. We agree with the reviewer that different models have diverse levels of water stress on photosynthesis (De Kauwe et al., 2017). Water stress is applied to regulate stomatal conductance in many models

(Rogers et al., 2017; Wu et al., 2018). For example, the CABLE model represents water stress by an empirical relationship based on soil texture and limits the slope of the coupled relationship between photosynthesis rate and stomatal conductance (Eq. S11). The influence of water stress is reflected by intercellular $CO_2$ concentration ($C_i$). Our results show modeled ratio of $C_i$ to atmospheric $CO_2$ concentration ($C_a$) is relatively constant for each PFT with $eCO_2$ and varies little among PFTs. This modeling result is consistent with the concept of homeostatic regulations through photosynthetic rate and stomatal conductance (Pearcy & Ehleringer, 1984; Evans & Farquhar, 1991). Wong et al. (1979) showed plant stomata could maintain a constant $C_i/C_a$ across wide range of environmental conditions, including water stress condition. Different models might have similar $C_i$ for a given $C_a$ but this assumption deserves further test. Moreover, Luo and Mooney (1996) found that changing $C_i/C_a$ ratio from 0.6 to 0.8 caused less than 15% variation in sensitivity of leaf photosynthesis to a unit of increase in $C_a$, which will not affect our conclusion about LAI as a major source of uncertainty. We will add the above discussions into our revised manuscript.

It's also true that different model might simulate different leaf temperatures as the reviewer pointed out. Sensitivity analysis in previous study has shown that a $\pm 5$℃ of leaf temperature changes caused approximately $\pm 7$ ppm changes in $\Gamma\_^*$, leading to coefficient of variation (CV) of 0.12 to leaf-level $\beta$ (Luo & Mooney, 1996). The overall variation of leaf-level $\beta$ caused by variation in leaf temperature is still quite small compared with that of $\beta\_GPP$.

Based on our literature review, only few studies like Hasegawa et al. (2017) have explored why different models simulated diverse responses of plant productivity to $eCO_2$. We will greatly appreciate it if the reviewer can show us some related references.

Reviewer 1: The authors argue for the importance of LAI but don't really consider the role of allocation or turnover in great detail. Surely this is the key reason different models arrive at different LAI values? Even if you ignore changes in allocation/turnover due to $CO_2$, this impacts on the scaling terms that the authors focus on.

Response: The reviewer made a great point. Changes in LAI are related to changes in allocation/turnover under eCO2. The response of allocation to eCO2 will influence $\beta$ in two ways. The first way is through altering the portion of carbon allocated to leaf, then changing LAI, which we have discussed in Discussion 4.2 (Line 348-351). The second way is by changing the allocation pattern to plant organs with different lifespan, thereby altering carbon turnover time in plants and soil. It has been briefly discussed through the difference between $\beta$_NPP and $\beta$_cpool (Line 363-373). We will discuss more about the first way in the revised manuscript: "Second, diverse allocation schemes will influence the responses of LAI for different plants. And, results from two FACE (Duke Forest and Oak Ridge) experiments indicate that the carbon allocated to leaves is decreased and more carbon is allocated to woods or roots at higher CO2 concentration (De Kauwe et al., 2014). Unfortunately, CABLE has fixed allocation coefficients and likely overestimates LAI response, leading to overestimated responses of GPP, NPP and total carbon storage".

Reviewer 1: 4. The results are considered on a PFT level, but presumably they vary in interesting ways within a PFT (i.e. in space). Would this be worth showing or exploring further?

Response: We have analyzed within-PFT variations of $\beta$ at different levels in Table 1, Results 3.3, and Fig. S1-S3 in the previously submitted manuscript.

Reviewer 1: Specific comments ================== - Line 43: Could you explain the CO2 fertilising effect further? The text as written expects the casual reader has significant background knowledge for the second sentence of your manuscript.

Response: Agree. We will add the following sentences in the first paragraph:"Persistent increase of atmospheric CO2 concentration will stimulate plant growth and ecosystem carbon storage, forming a negative feedback to CO2 concentration (Long et al., 2004; Friedlingstein et al., 2006). This concentration-carbon feedback ($\beta$), also called CO2 fertilizing effect, has been identified as a major uncertainty in modeling terrestrial

carbon-cycle response to historical climate change (Huntzinger et al., 2017)".

Reviewer 1: - Line 48: 4 or 4.5? What does that mean, do you mean 4 to 4.5? How can it be OR?

Response: Sorry for the ambiguity. Actually, the contribution of $\beta$ is 4 times larger than that of carbon-climate feedback factor $\gamma$ in Gregory et al. (2009) and Bonan and Levis (2010), but is 4.5 times larger in Arora et al. (2013). We will change this sentence to "Some studies pointed out that the contribution of $\beta$ is 4 to 4.5 times larger, and more uncertain, than carbon-climate feedback factor ($\gamma$) (Gregory et al., 2009; Bonan & Levis, 2010; Arora et al., 2013)".

Reviewer 1: - Line 49: the reference to the Smith et al. paper ignores a technical comment on this paper: De Kauwe et al. (2016). Satellite based estimates underestimate the effect of CO2 fertilization on net primary productivity. Nature Climate Change, 6, 892-893. This is important as the authors are using this study to leverage their question. See also point on line 340.

Response: We thank the reviewer for pointing to the related comment paper by De Kauwe et al. (2016). It is indeed an important reference to supplement the point we were trying to make. We have modified the last sentence in the first paragraph to "Though satellite products they used may underestimate the effect of CO2 fertilization on net primary productivity (De Kauwe et al., 2016), the large disparity between models and FACE experiments gives us little confidence in making policies to combat global warming".

Reviewer 1: - Line 51: it isn't "reality" - the satellite estimates are also model estimates.

Response: Agree. See the response above.

Reviewer 1: - Line 54: "increasing temperature in models" why is temperature being introduced as a factor here? Isn't the focus solely on the CO2 fertilisation effect rather than the than carbon-climate feedback factor? There are further studies cited in this

paragraph which should be removed if the focus of this paper does not consider the carbon-climate feedback factor.

Response: Agree. We will remove the $\gamma$-related part in the revised manuscript.

Reviewer 1: - Line 67: Despite models using apparently similar photosynthesis models, Rogers et al. (A roadmap for improving the representation of photosynthesis in Earth system models. New Phytologist, 213, 22-42.) showed some important differences. It would be worthwhile highlighting this study in the context of the section of the text.

Response: We thank the reviewer for sharing us this important reference. We will adjust the sentence to a more accurate one: "The leaf-level $CO_2$ fertilization for C3 plants is generally well characterized with models from Farquhar et al. (1980), and the basic biochemical mechanisms have been adopted by most land surface models although some models implement variants of Farquhar et al. (1980) (Rogers et al., 2017)". We will discuss more about how those different implementations influence photosynthetic response in the Discussion: "Some models use variants of Farquhar photosynthesis model such as co-limitation approach described by Collatz et al. (1991). The absolute values of photosynthetic response to eCO2 in these models are diverse mainly due to model divergence in inflection point from Rubisco- to RuBP- limited processes (Rogers et al., 2017). However, the relative photosynthetic responses will converge to a small range because the normalized photosynthetic response to eCO2 only depends on estimates of intercellular $CO_2$ concentration ($C\_i$), Michaelis-Menten constants ($K\_c$, $K\_o$) and $CO_2$ compensation point in the absence of day respiration ($\Gamma\_*$), and relative leaf-level responses are similar for either Rubisco- or RuBP-limited photosynthesis (Luo et al., 1996; Luo & Mooney, 1996)".

Reviewer 1: - Line 72: what does carbon storage have to do with this sentence?

Response: Thanks for pointing out what we have missed. Besides NPP, allocation and carbon turnover process can influence carbon storage. We will change this sentence to "However, the $CO_2$ fertilization effects are considerably more variable at canopy- and

**BGD**

ecosystem-level than at the leaf-level, because a cascade of uncertain factors, such as soil moisture feedback (Fatichi et al., 2016), nutrient limitation (Zaehle et al., 2014), allocation (De Kauwe et al., 2014), and carbon turnover process (Friend et al., 2014) influence the responses of GPP, NPP and carbon storage".

Reviewer 1: - Line 76/7: seems a narrow characterisation of the literature, the De Kauwe et al. 2014 study that the authors cite, explored these issues in depth.

Response: We will add related references as the reviewer suggested: "Models generally predict that LAI dynamics will respond to eCO2 positively due to enhanced NPP and leaf biomass (De Kauwe et al., 2014). Zhu et al. (2016) has attributed global increases in satellite LAI primarily to increased CO2 concentrations. But how the increasing LAI in turn feeds back to ecosystem carbon uptake as a result of more light interception has not been discussed in previous research".

Reviewer 1: - Line 81: Why would a high "basic" (delete basic) NPP necessarily lead to tropical regions having the highest stimulation by CO2? Wouldn't the opposite be expected? These regions have a high LAI and so would predominantly be light-limited and so have a more limited capacity to respond to CO2? Either way, the authors need to expand on this assertion.

Response: We agree this sentence is not very clear. We are going to change the first sentence in this paragraph into "The largest absolute CO2 fertilization effect has been found in tropical area where already has the highest initial NPP (Joos et al., 2001; Peng et al., 2014). But with gradual eCO2, relative response in tropical area might not be very high owing to canopy closure (Norby et al., 2005)".

Reviewer 1: - Line 89: Improved on what?

Response: We will change this sentence to: "CABLE (version 2.0) is the Australian community land surface model (Kowalczyk et al., 2006) and incorporates CASA-CNP to simulate global carbon, nitrogen and phosphorus cycles (Wang et al., 2010; Wang

et al., 2011)".

Reviewer 1: - Line 124: The assumption that Jmax25 = 2 x Vcmax25. Did the authors consider varying this assumption? Other models would make quite different assumptions about this ratio.

Response: It's true that the ratio of the maximum electron transport rate ($J_{(max,25)}$) to maximum photosynthetic capacity ($V_{(cmax,25)}$) are different in models (Rogers et al., 2017). But difference of this ratio will not change the conclusion because $\beta$ factors in our study are normalized values, irrespective of $J_{(max,25)}$ or $V_{(cmax,25)}$. In terms of the variation of this ratio due to eCO2, we have discussed the downregulation of $J_{(max,25)}$ and $V_{(cmax,25)}$ in the manuscript Line 308-313.

Reviewer 1: - Line 155: is there a citation, web link for "Community Climate System Model (CCSM) simulations"

Response: We will add a citation "Hurrell, J. W., Holland, M. M., Gent, P. R., Ghan, S., Kay, J. E., Kushner, P. J., ... & Lipscomb, W. H. 2013. The community earth system model: a framework for collaborative research. Bulletin of the American Meteorological Society, 94 1339-1360".

Reviewer 1: - Line 168: the definition of S (line 171) needs to be moved up to this line.

Response: Agree.

Reviewer 1: - Line 215: just to clarify when the authors say total carbon storage - do they mean the soils too? Or just the plant? Or just the foliage pool? The equation isn't very clear. This also makes Fig 1 hard for me to interpret as I'm unclear what is being shown, I'm going to assume it is total plant carbon...

Response: Total carbon storage is the sum of plant, litter and soil carbon pools. We'll make it clearer in the revised manuscript.

Reviewer 1: - Fig 1. Does it make sense to normalise these PFT lines? The authors

say they decline but the magnitudes differ, the point is that the initial starting points are different too. This makes it hard for the eye to gauge.

Response: Indeed, the CO2 fertilization effects at different levels in our manuscript are all normalized values. See Eq.19, 20, 23, 24, 27,28.

Reviewer 1: - As a general comment the results need work, particularly in terms of transition text. For example 3.1 talks about the temporal trend in Bcpool and then switches immediately to the Ci/Ca ratio in 3.2? It is hard to follow the logic of the transition, is there is meant to be any connection for the reader?

Response: Section 3.1 is about $\beta$ factors at ecosystem level, showing $\beta$ factors are diverging for different PFTs through time. It stimulates our following study that calculating $\beta$ values from leaf biochemical level to canopy level in order to identify the key processes. We will add one transition sentence at the beginning of 3.2: "To reveal which processes cause the large disparity of $\beta$ factors across vegetation types as shown in Fig. 1, we first compared biochemical parameters: intercellular CO2 concentration and CO2 compensation point, which are critical parameters for leaf-level biochemical response".

Reviewer 1: - What is the point of Fig. 2? It isn't clear what this figure has to do with the story of the paper?

Response: Please see the above response.

Reviewer 1: - The text around line 261 which refers to Fig 4 could do with further explanation. I personally don't find this particularly surprising, but the reader isn't offered much as the way of explanation. Presumably the change in slope as you move from B_GPP to B_NPP relates to respiration assumptions and then to B_cpool, allocation/turnover assumptions? I think the authors could go further in assisting the reader with interpretation. As currently written, the text simply highlights that the slope changes.

Response: After thinking carefully about this concern, we agree that the slopes of the three fitting lines are not making much sense so we will remove this sentence in the revised manuscript.

Reviewer 1: - I think figure 5 is very interesting.

Response: Thank the reviewer for the positive comment.

Reviewer 1: - Line 290: I think this discussion of Fig S5 is interesting but I'm not sure I follow the interpretation? The LAI is the emergent outcome of the model assumptions - 1 leaf, 2leaf, multi-layer. Of course this assumption will lead to differences? But why you do the analysis on the leaf-level? Surely you're interested in the emergent outcome – the LAI. Most likely I simply misunderstood this point but I think it could also be explained further as it seems like an important point the authors are making.

Response: What we would like to discuss here is that CMIP5 model outputs have limited information for identifying mechanisms for model uncertainty since there are no leaf-level process outputs. We will reorganize the first paragraph in Section 4.1: "By contrast, CMIP5 model outputs have limited information in identifying mechanisms for model uncertainty since there are no leaf-level process outputs. In Hajima et al. (2014), they used GPP divided by LAI as a proxy to represent leaf-level photosynthesis for CMIP5 models. In our study, we also compared the sensitivities of GPP/LAI to eCO2 with our calculation of leaf-level $\beta$ values which are derived from $C_i$ and $\Gamma\_*$ for different vegetation types. Results from former calculation are greatly underestimated for trees and slightly overestimated for C3 grass and tundra (Fig. S5). The divergence of sensitivities of GPP/LAI across vegetation types is larger compared with that of our mechanistic calculation of leaf-level $\beta$. The bias is not only derived from the complex canopy structure used by each model (two-leaf or multiple-layer), but also from the nonlinear effect of LAI on GPP. Thus, the relatively large divergence of the sensitivities of GPP/LAI to eCO2 in Hajima et al. (2014) may not indicate diverse leaf-level photosynthesis responses among CMIP5 models. This comparison confirms the urgent

need to include leaf-level diagnostics in the next intermodal comparison project".

Reviewer 1: - Line 295: I don't fully follow that interpretation? Your differences in Ci/Ca were small across PFTs? And the differences in leaf temp would be expected between PFTs? Certainly, fig 2 doesn't show any within PFT variation.

Response: Yes, we think the reviewer's understanding is correct. We would like to express that the leaf-level $\beta_i$ computed in our study can be mechanistically traced back to intercellular CO2 concentration and leaf temperature. Since Fig. 2 shows the results across different PFTs, we'll change this sentence to: "Another advantage of our calculation of leaf-level $\beta_i$ is that the reason for the divergence of leaf-level $\beta_i$ across vegetation types can be traced back to differences in $c_i/c_a$ and leaf temperature as shown in Fig. 2".

Reviewer 1: - Line 362: This is an assumption of the model and might not necessarily be true!

Response: Agree. We will add the following sentences in the manuscript: "FACE experimental results indicate that CUE values under eCO2 are not changed in N-limited Duke site (Hamilton et al., 2002; Schäfer et al., 2003), increase in fertile POPFACE site (Gielen et al., 2005) or decrease in fertile ORNL site (DeLucia et al., 2005). Thus, representations of nutrient limitations on GPP and autotrophic respiration in land surface models should be carefully calibrated with experimental data (DeLucia et al., 2007)".

Reviewer 1: Technical corrections =====================

- Abstract: "vegetation types is 0.15-0.13", presumably you meant 0.13 to 0.15? Also, why don't the other variables (e.g. BetaGPP) have ranges too?

Response: Yes, we meant 0.13 and 0.15 for shaded leaf and sunlit leaf, respectively. At canopy level, we did not differentiate sunlit leaves and shaded leaves, so there is only one value for $\beta$_GPP.

Reviewer 1: - First line of the introduction, makes no sense. You can't start a sentence

with Terrestrial carbon sink and then a comma.

Response: Agree. We will change the first sentence to: "Terrestrial ecosystems take up roughly 30% of anthropogenic CO2 emissions, and is of great uncertainty and vulnerable to global climate change (Cox et al., 2000; Le Quéré et al., 2017)".

Reviewer 1: - Line 45: In Coupled -> In the Coupled

Response: Agree.

Reviewer 1: - Line 138: In CABLE model -> in the cable model

Response: Agree.

References

Arora, V. K., Boer, G. J., Friedlingstein, P., Eby, M., Jones, C. D., Christian, J. R., ... & Hajima, T. (2013). Carbon–concentration and carbon–climate feedbacks in CMIP5 Earth system models. Journal of Climate, 26(15), 5289-5314.

De Kauwe, M. G., Medlyn, B. E., Walker, A. P., Zaehle, S., Asao, S., Guenet, B., ... & Lu, X. (2017). Challenging terrestrial biosphere models with data from the long‐term multifactor Prairie Heating and CO2 Enrichment experiment. Global change biology, 23(9), 3623-3645.

DeLucia, E. H., Moore, D. J., & Norby, R. J. (2005). Contrasting responses of forest ecosystems to rising atmospheric CO2: implications for the global C cycle. Global Biogeochemical Cycles, 19(3).

DeLucia, E. V. A. N., Drake, J. E., Thomas, R. B., & GONZALEZ‐MELER, M. I. Q. U. E. L. (2007). Forest carbon use efficiency: is respiration a constant fraction of gross primary production? Global Change Biology, 13(6), 1157-1167.

Evans, J.R., Farquhar, G. D. (1991). Modeling canopy photosynthesis from the biochemistry of the C3 chloroplast. Modeling crop photosynthesis—from biochemistry

to canopy, (modelingcroppho), 1-15.

Fatichi, S., Leuzinger, S., Paschalis, A., Langley, J. A., Barraclough, A. D., & Hovenden, M. J. (2016). Partitioning direct and indirect effects reveals the response of water-limited ecosystems to elevated CO2. Proceedings of the National Academy of Sciences, 113(45), 12757-12762.

Friedlingstein, P., Cox, P., Betts, R., Bopp, L., von Bloh, W., Brovkin, V., ... & Bala, G. (2006). Climate–carbon cycle feedback analysis: results from the C4MIP model intercomparison. Journal of climate, 19(14), 3337-3353.

Gielen BC, Calfapietra C, Lukac M, Wittig VE, DeAngelis P, Janssens IA, Moscatelli MC, Grego S, Cotrufo MF, Godbold DL, Hoosbeek MR, Long SP, Miglietta F, Polle A, Bernacchi CJ, Davey PA, Ceulemans R, Scarascia-Mugnozza GE (2005). Net carbon storage in a poplar plantation (POPFACE) after three years of free-air CO2 enrichment. Tree Physiology, 25, 1399-1408.

Hamilton, J. G., DeLucia, E. H., George, K., Naidu, S. L., Finzi, A. C., & Schlesinger, W. H. (2002). Forest carbon balance under elevated CO 2. Oecologia, 131(2), 250-260.

He, Y., Trumbore, S. E., Torn, M. S., Harden, J. W., Vaughn, L. J., Allison, S. D., & Randerson, J. T. (2016). Radiocarbon constraints imply reduced carbon uptake by soils during the 21st century. Science, 353(6306), 1419-1424.

Huntzinger, D. N., Michalak, A. M., Schwalm, C., Ciais, P., King, A. W., Fang, Y., ... & Hayes, D. (2017). Uncertainty in the response of terrestrial carbon sink to environmental drivers undermines carbon-climate feedback predictions. Scientific Reports, 7(1), 4765.

Kattge, J., Knorr, W., Raddatz, T., & Wirth, C. (2009). Quantifying photosynthetic capacity and its relationship to leaf nitrogen content for global‐scale terrestrial biosphere models. Global Change Biology, 15(4), 976-991.

Koven, C. D., Chambers, J. Q., Georgiou, K., Knox, R., Negron-Juarez, R., Riley, W.

J., ... & Jones, C. D. (2015). Controls on terrestrial carbon feedbacks by productivity versus turnover in the CMIP5 Earth System Models.

Long, S. P., Ainsworth, E. A., Leakey, A. D., Nösberger, J., & Ort, D. R. (2006). Food for thought: lower-than-expected crop yield stimulation with rising CO2 concentrations. Science, 312(5782), 1918-1921.

Luo, Y., Sims, D. A., Thomas, R. B., Tissue, D. T., & Ball, J. T. (1996). Sensitivity of leaf photosynthesis to CO2 concentration is an invariant function for C3 plants: A test with experimental data and global applications. Global Biogeochemical Cycles, 10(2), 209-222.

Luo, Y., & Mooney, H. A. (1996). Stimulation of global photosynthetic carbon influx by an increase in atmospheric carbon dioxide concentration. In Carbon dioxide and terrestrial ecosystems (pp. 381-397).

Norby, R. J., DeLucia, E. H., Gielen, B., Calfapietra, C., Giardina, C. P., King, J. S., ... & De Angelis, P. (2005). Forest response to elevated CO2 is conserved across a broad range of productivity. Proceedings of the National Academy of Sciences, 102(50), 18052-18056.

Pearcy, R. W., & Ehleringer, J. (1984). Comparative ecophysiology of C3 and C4 plants. Plant, Cell & Environment, 7(1), 1-13.

Schäfer, K. V., Oren, R., Ellsworth, D. S., Lai, C. T., Herrick, J. D., Finzi, A. C., ... & Katul, G. G. (2003). Exposure to an enriched CO2 atmosphere alters carbon assimilation and allocation in a pine forest ecosystem. Global Change Biology, 9(10), 1378-1400.

Wang, Y. P., Law, R. M., & Pak, B. (2010). A global model of carbon, nitrogen and phosphorus cycles for the terrestrial biosphere. Biogeosciences, 7(7), 2261-2282.

Wang, Y. P., Lu, X. J., Wright, I. J., Dai, Y. J., Rayner, P. J., & Reich, P. B. (2012). Correlations among leaf traits provide a significant constraint on the estimate of global gross primary production. Geophysical Research Letters, 39(19).

Wong, S. C., Cowan, I. R., & Farquhar, G. D. (1979). Stomatal conductance correlates with photosynthetic capacity. Nature, 282(5737), 424.

Wu, D., Ciais, P., Viovy, N., & Vicca, S. (2018). Asymmetric responses of primary productivity to altered precipitation simulated by ecosystem models across three long-term grassland sites. Biogeosciences, 15, 3421-3437.

Zhang, Q., Wang, Y. P., Pitman, A. J., & Dai, Y. J. (2011). Limitations of nitrogen and phosphorous on the terrestrial carbon uptake in the 20th century. Geophysical Research Letters, 38(22).

Zhang, Q., Pitman, A. J., Wang, Y. P., Dai, Y. J., & Lawrence, P. J. (2013). The impact of nitrogen and phosphorous limitation on the estimated terrestrial carbon balance and warming of land use change over the last 156 yr. Earth System Dynamics, 4(2), 333-345.

Zhu, Z., Piao, S., Myneni, R. B., Huang, M., Zeng, Z., Canadell, J. G., ... & Cao, C. (2016). Greening of the Earth and its drivers. Nature Climate Change, 6(8), 791-795.
* * *
[Figure]

[Figure]

**Fig. 1.** $\beta$ values at different levels for various C3 plants at the year 2023 from CABLE-C only(a), CABLE-CN (b) and CABLE-CNP (c) simulations.

[Figure]

**Fig. 2.** Correlations between $\beta\_GPP$ and $\beta\_LAI$, $\beta\_NPP$ and $\beta\_LAI$, $\beta\_cpool$ and $\beta\_LAI$ from CABLE C-only (a)∼(c), CABLE-CN (d)∼(f) and CABLE-CNP (g)∼(i) at the year 2023 across C3 plants.

[Figure]

---

## Author Comment (AC2) · 21 Aug 2018

Dear editor,

We appreciate the insightful comments on our manuscript by all reviewers, and will make substantial revisions to improve this manuscript. We hope that our responses below address the concerns raised by reviewers. The major proposed changes are: We will include results from carbon-nitrogen (C-N) coupled and carbon-nitrogen-phosphorus (C-N-P) coupled simulations of CABLE to study how the CO2 fertilization effects ($\beta$ factor) at different levels will change with nutrient limitations for different plant functional types. Our results from C-N and C-N-P coupled simulations support our pervious conclusion with C only simulations. We will clarify the motivation and contribution

of our study as the reviewers suggested in our revised manuscript. We will expand the scope of our work to attract more readers and carefully correct language in our revised manuscript.

Reviewer 2: Synopsis: In this paper, the authors run CABLE for seven C3 vegetation types, without nutrient cycling, and calculate CO2 fertilization for the RCP 8.5 scenario. CCSM simulations from 1901 to (the paper says 1910; I assume they mean 2010) holding carbon-climate feedbacks constant (driving the model with the averaged meteorology-I'm guessing average annual cycle, although the authors do not say) and feeding CABLE increasing CO2 concentration from the CCSM RCP 8.5 results.

They find that CO2 fertilization differs between PFTs, and decreases with time during the period 2011-2100. Fertilization is relatively constant both between PFTs and when the calculation is made on a per-unit leaf level, and shows much larger diversity both across PFTs and when the CO2 fertilization is calculated on a unit-leaf vs. integrated canopy basis. The authors close with the claim that simulated LAI is critical to the calculation of CO2 fertilization in climate simulations.

Response: We thank the reviewer for the time she or he spent on reviewing our manuscript. The above paragraphs are a good summary of what we did for this study. While most of the summary is accurate, we would like to clarify here that CABLE model has been run from 1901 to 2100. Before that, CABLE was spun up by using meteorological forcing from 1901 to 1910 repetitively until a steady state was achieved. And we indeed used the average annual cycle of meteorological forcing data to fix carbon-climate feedbacks. We will clarify these points in our revised manuscript.

Reviewer 2: Review: I have 2 major problems with this paper. Either one by itself, I believe, is fatal, but taken together I cannot make any recommendation for this paper other than rejection.

Response: We are sorry that this reviewer did not, unfortunately, find our study scientifically meritorious, largely due to the fact that our research objective was not well

understood by the reviewer. We hope our responses to her or his comments could help the reviewer to re-evaluate our manuscript.

Reviewer 2: Problem #1: There is a rich body of literature from the FACE experiments that claims, pretty much unequivocally, that nutrient cycling and/or limitation becomes more and more important to $CO_2$ fertilization as $CO_2$ concentrations rise. Yet, in this experiment CABLE is run with nutrient cycling turned off!

Response: We feel sorry that our research objective was not clearly conveyed to the reviewer. Our study was to examine how variability, as measured by coefficient of variation (CV), in the $CO_2$ fertilization effect (i.e., CV of $\beta$ factor) changes from leaf to canopy GPP, ecosystem NPP and total carbon storage levels. Our study was not to quantify the $CO_2$ fertilization effect itself.

We agree with the reviewer that nutrient limitations are universally observed in experiments. Nutrient cycling influences the $CO_2$ fertilization effect. But running CABLE with coupled carbon, nitrogen and phosphorus cycles does not change the conclusion about CV of the $CO_2$ fertilization effects, which was previously reached by running carbon-only version of CABLE. Thus, we hope this reviewer will re-evaluate our manuscript, particularly with new simulations results from running the coupled CABLE-CN and CABLE-CNP models as presented below.

Reviewer 2: Coskun et al. (2016) and references therein has a nice summary of both Free-Air $CO_2$ Enrichment (FACE) as well as Open-Top Chamber (OTC) experiments. Smith et al. (2015) discusses the divergence between multiple models and a satellite-derived product that underscores the importance of the interaction between nutrient cycling and $CO_2$ fertilization. Many of these studies focus on N limitation, although some research has indicated that P limitation is a factor as well (e.g. Hasegawa et al., 2016). These, and other studies, all conclude that understanding of $CO_2$ fertilization requires taking nutrients into account.

Response: We thank the reviewer for showing us these important references. The

reviewer cited a paper of Smith et al. (2015) and believed the overestimation of the $CO_2$ fertilization in those CMIP5 models is mainly caused by lack of nutrient limitations. But a related comment De Kauwe et al. (2016) suggests that it still too premature to reach this conclusion because the nitrogen-incorporated model CESM1-BGC did not work well in simulating $CO_2$ uptake and response of NPP to elevated $CO_2$ ($eCO_2$) against FACE experimental results.

Again, our study was not to quantify the $CO_2$ fertilization effect itself but to understand what caused changes in CV of $\beta$ factor. Running CABLE without or with nutrient limitation reached a similar conclusion as shown below.

Reviewer 2: I have to confess that I was very surprised when I read that the authors ran the version of CABLE without nutrient cycling included. I am not a FACE 'expert', but even I am aware of the amount of research that has concluded that nutrient cycling is critical to understanding ecosystem-level response to higher atmospheric $CO_2$. I found it very suspicious that nutrients were excluded from the study. Why, when there is this large body of work demonstrating the nutrient cycling is critical to understanding $CO_2$ enrichment, would nutrients be turned off in the model? The authors claim that nutrients were turned off for 'simplicity', but the obvious answer, and one that I suspect to be the truth, is that the authors did run CABLE with nutrient cycling, and model pathology and/or unrealistic results ensued.

Response: We thank the reviewer for the critical comments and his/her insistence on the necessity of nutrient-coupled simulations. We absolutely agree with the reviewer that the $CO_2$ fertilization effect (or $\beta$ factor) could be more realistically represented with nutrient limitations considered. However, we feel sorry that the reviewer might not understand the purpose of our study properly. This study was designed to diagnose important model processes that cause divergence in $\beta$ factor. The large uncertainty of $\beta$ values remains a big challenge in CMIP5 models, many of which don't have carbon-nitrogen interactions. For the sake of diagnosis, we turned off the nutrient interactions to identify the most critical carbon-cycle processes in the CABLE model,

which has not been attempted in previous studies. We found $\beta$ factors at canopy and ecosystem levels in C-only simulation diverge in a way that is largely attributable to variations in LAI responses within and across C3 vegetation types in the CABLE model. Our new results from CABLE-CN (carbon-nitrogen) and CABLE-CNP (carbon-nitrogen-phosphorus) simulations suggest nutrient effects add more variations to $\beta$ values at ecosystem level compared with C-only simulation (Fig. 1 in this response letter). However, the results from the CABLE-CN and CABLE-CNP simulations add more layers of complexity to understand the primary mechanisms underlying the divergence of $\beta$ factors at different levels and in different ecosystems albert the conclusion is similar with that reached from running carbon-only CABLE.

Besides, we feel the reviewer's conjecture: "and one that I suspect to be the truth, is that the authors did run CABLE with nutrient cycling, and model pathology and/or unrealistic results ensued" is too speculative, largely due to her/his incomplete understanding of our study. Per the suggestions from the two reviewers, we tested whether the patterns and mechanisms for the variability of $\beta$ factors for C-only simulation still hold for nutrient-coupled simulations. We will add results from C-N and C-N-P coupled simulations of CABLE in the revised manuscript.

Until now, our new results from the C-N and C-N-P coupled simulations support our previous conclusions that at leaf-level $\beta$ factors do not vary much for different vegetation types (Fig. 1 in this response letter). But at canopy and ecosystem levels, $\beta$ factors diverge because the responses of LAI and nutrient limitations differentiate among vegetation types (Fig. 2 in this response letter). Please see more details about the mechanism of nutrient limitations in the CABLE model and our responses to the comments by Dr. De Kauwe.

Reviewer 2: It may have been possible to evaluate a nutrient run, even if the results were unrealistic, and evaluate how atmospheric CO2 levels and nutrients interact in CABLE. The results may have provided an opportunity to evaluate or comment on the divergence of models in their predictions of atmospheric CO2 levels and source/sink

strength (e.g. Friedlingstein et al., 2006, 2014). By not including the critical nutrient interaction, I'm not sure that the results presented here give the reader any insight into how ecosystems might realistically respond to increasing future CO2 levels in the atmosphere.

Response: We agree with the reviewer that CO2 and nutrient interactions could cause the divergence of models. Our new results with CABLE-CN and CABLE-CNP show that CV of $\beta$ is much higher than that with CABLE-C only for NPP and total carbon storage (Fig. 1 in this response letter). However, the objective of our study is not to evaluate nutrient effects on carbon cycle under CO2 fertilization. As we have stated before, our study is to identify mechanisms underlying expanding CV from biochemical and leaf levels to canopy GPP, ecosystem NPP and carbon pool. All the three versions of CABLE point to the same mechanism, which is LAI as the major source of variability in modelled CO2 fertilization.

Reviewer 2: Problem #2: Without carbon-climate feedbacks and nutrient cycles, I don't think a model actually has to be run to determine CO2 fertilization. You can probably perform the calculation directly from the equations in the code. Between models there will be some differences: • Is the model an enzyme-kinetic model (Farquhar et al., 1980; Michaelis-Menten kinetics), or light-response (e.g. VPRM, Mahadevan et al., 2008)? • how is stomatal conductance calculated? Does it use Ball-Berry, with a dependence on relative humidity, or Leuning, which uses VPD? How is transpiration coupled to photosynthesis? • What are the parameter values for Vcmax for a given PFT? • What determines phenology? Is allocation static, or, if it is dynamic, how does it change during the year and in response to what? I believe it would be possible to determine the constraints on CO2 fertilization for a suite of models without actually running any of them.

Response: We agree with the reviewer that these assumptions and processes are key to modelling terrestrial carbon-cycle responses to eCO2. The reviewer is very knowl-edgeable to identify those key ecosystem carbon-cycle processes. In this comment

alone, the reviewer mentioned more than 10 processes that influence photosynthesis. We were very curious how the reviewer could "perform the calculation directly from the equations in the code" to evaluate all those 10 processes and to gain a mechanistic understanding of what causes the change of $\beta$ values. Obviously, we did not understand how to perform the calculation of $\beta$ factors directly from the model equations as the reviewer suggested. The reviewer was apparently very confident for doing so. Even if we could calculate based on several equations, the results might not truly reflect model mechanisms for variabilities of the CO2 fertilization effects within and across vegetation types. Because carbon-cycle processes are tightly coupled with radiation transfer, energy balance, nutrient interactions and water cycles in a land surface model. For example, leaf temperature and intercellular CO2 concentration are two important variables for leaf-level $\beta$ values, which are collectively controlled by air temperature, radiation transfer and humidity. We were not sure if the reviewer meant to construct a simplified model or emulator to mimic the complex land surface models, it is worthy trying but we were not confident that the simplified approach can reveal model mechanisms.

Nevertheless, we ran a well-evaluated land surface model and outputted process-level variables such as intercellular CO2 concentration, LAI, GPP, NPP, and ecosystem carbon storage for all land cells, as many analyses have done based on C4MIP and CMIP5. Combining previous theoretical analysis, we have shown that CV of $\beta$ is small for biochemical and leaf-level photosynthesis but large for canopy GPP, ecosystem NPP and carbon pools.

Reviewer 2: It is axiomatic that leaf-to-canopy scaling (LAI) is critical to total CO2 fertilization amount. Every model that I am aware of calculates biophysics on a per-unit-are basis and then scales to the canopy level either by summing over sunlit/shaded leaves (and PFTs) or integrating from leaf to canopy scale along the lines of Sellers (1985, 1992)(OK, a gap model like ED2 may be a little different). Canopies with an LAI close to 1 (think of grasslands) will not see much difference from unit- to canopy-scale,

more dense canopies (like forests) will.

Response: We are happy that the reviewer agrees with us that LAI is critical for plant productivity. Many models exhibit increasing LAI trends under CO2 fertilization (Zhu et al., 2016). However, to what extent the increasing LAI feeds back to ecosystem response to eCO2 is not clear. Our study for the first time calculated $\beta$ factors from leaf biochemical level to ecosystem level, and found the LAI response to eCO2 is the dominating factor for variabilities of the CO2 fertilization effects at canopy and ecosystem levels within and across C3 vegetation types, namely the global CO2 fertilization effects are very sensitive to the LAI responses. The value of our study is that it can urge modelling groups to improve the representation of LAI in land surface models, for example by calibrating allocation coefficients and specific leaf area (SLA) based on FACE experimental results (De Kauwe et al., 2014), so as to realistically predict concentration–carbon feedback.

Reviewer 2: If there is a large divergence between models in LAI (and GPP) for a given PFT, or if there is a large trend in one model's LAI for a given PFT during a climate run, then these might be valid topics of analysis. Finding that LAI is critical to canopy-level CO2 fertilization (without nutrients being considered) does not really bring anything new to the field.

Response: Our results may not be much new for this reviewer but the key message from our study is still crucial for the community to improve land modeling. Actually, in our manuscript we have cited a paper showing CMIP5 models have simulated diverse GPP and LAI values for different regions during 1985-2006. And both GPP and LAI have been overestimated for most CMIP5 models according to observations (Anav et al., 2013) (Line 329-330). Satellite and modelled LAI both have experienced significant increasing trends during historical period as reported by Zhu et al. (2016). However, how the uncertainty and increasing trend of LAI contribute to modelled plant productivity and ecosystem carbon storage have not been discussed in previous research. Our study fills this gap and indicates the CO2 fertilization effects are very sensitive to LAI

responses. We agree with the reviewer that the CO2 fertilization effects on GPP and LAI will be more realistically presented with nutrient limitations. Our new results show GPP and LAI still positively respond to eCO2 under nutrient limitations but with reduced magnitudes. The responses of LAI still dominate the change of ecosystem responses across vegetation types in C-N and C-N-P coupled simulations (Fig. 2 in this response letter). The merit of our study is that we systematically diagnose model processes and find LAI is the most important factor in modelling the CO2 fertilization effects, to which modelers should pay greater attentions and efforts in the future research.

Reviewer 2: Sunlit and shaded leaf partitioning is fairly well-constrained and sunlit LAI can never get much above 1 to 1.5 or so even under the most direct-sun conditions. Solar angle and leaf angle distribution make it possible to exceed an LAI value of one. I know that CLM has had issues with shade leaf LAI becoming excessively large. The authors do not discuss total LAI in CABLE during their fertilization runs, and this makes me suspicious-if their shade-leaf LAI is becoming unrealistically large, that might be a reason why fertilization strength decreases with time; increase in the amount of sunlit leaf may result in large change in GPP, but once sunlit LAI is filled, any additional canopy growth will be as shade LAI, and GPP increase will be attenuated.

Response: We appreciate the reviewer for the insightful comments. Actually, we did analyze total LAI change in the submitted supplementary material Fig. S6. LAI value of evergreen broadleaf forest increases with time but gradually saturates at the prescribed maximum value. LAI values of other plant types also increase but are far below the prescribed maximum values at 2100. To address the reviewer's concern about the magnitudes and changes of sunlit and shaded leaf LAI (we called the scaling factors in our manuscript according to the standard definition in the CABLE model), we plotted temporal trends of the scaling factors for sunlit leaf and shaded leaves in CABLE-C only simulation (Fig. 3 in this response letter). Results show that the magnitudes of the scaling factors for shaded leaves are greatly larger than those for sunlit leaves for all C3 plants. This is because in models it is usually defined that portion of sunlit leaves

decreases exponentially with increasing LAI (f_sun= exp (-k_b LAI)) (Dai et al., 2004). The scaling factors for sunlit leaves are below 1 as the reviewer stated. And the scaling factors for sunlit leaves of evergreen broadleaf forest, evergreen needleleaf forest and deciduous broadleaf forest gradually saturate with eCO2.

The increasing portion of shaded leaves will lead to the attenuation of GPP increase as the reviewer mentioned. And we believe that saturation of GPP is jointly controlled by biochemical enzyme kinetics and canopy closure. In our submitted manuscript, we have stated that the saturation of GPP in response to eCO2 is not only regulated by the leaf-level response, but also by the response of the scaling factors to eCO2 (Fig. 6 in submitted manuscript; Line 324-325). The mechanisms for leaf-level saturation have been discussed in detail in Luo et al. (1996) and Luo and Mooney (1996).

Reviewer 2: I just don't think there's anything new here. Without nutrient cycling the CO2 fertilization results don't have much meaningful application, and the fact that leaf-to-canopy scaling is important has been known for a long time.

Response: The reviewer's assertion about the scientific contributions of our study is partly due to her or his incomplete understanding of our research objective. We have run CABLE with coupled carbon-nitrogen-phosphorus cycles. Our original conclusion still stands. Although leaf-to-canopy scaling has been known for a long time, no study has done before as we did in this study to evaluate variation of $\beta$ factors from biochemical and leaf levels to canopy, ecosystem scales. The leaf-to-canopy scaling is a basis of our study but the conclusion of our study goes far beyond it.

Here, we will strengthen our contributions through the following ways:

Analyzing the CO2 fertilization effects at different levels with C-N and C-N-P interactions for different C3 vegetation types in the CABLE model to evaluate whether our conclusions are still valid under nutrient limitations.

In the introduction and discussion part, we will clarify that it is the large uncertainty of

concentration-carbon feedbacks produced by CMIP5 models that motivates our work. Our mechanistic study, for the first time, shows that $\beta$ factors vary at different hierarchical levels across C-fluxes and stocks, and across PFTs in a way that is largely attributable to variations in LAI dynamics across PFTs at canopy and ecosystem levels. Our study can be useful by urging different modeling groups to quantify the CO2 fertilization effects at different levels as we did and output leaf-level diagnostics for the next CMIP.

Our finding about the dominant role of LAI can stimulate modelling groups to focus more on uncertainty arising from processes related to LAI, and use FACE experiments to narrow the uncertainty of land model predictions.

Reviewer 2: Specific comments: • English prose and grammar, while readable, need attention. There are multiple places, too many to list, where errors exist.

Response: We will carefully revise the manuscript and improve the language in the revised version.

Reviewer 2: • There is no explanation for what eCO2 is (elevated CO2). Don't assume all your readers know the definition.

Response: We will add the following sentences in the first paragraph:"Persistent increase of atmospheric CO2 concentration will stimulate plant growth and ecosystem carbon storage, forming a negative feedback to CO2 concentration (Long et al., 2004; Friedlingstein et al., 2006). This concentration-carbon feedback ($\beta$), also called CO2 fertilizing effect, has been identified as a major uncertainty in modeling terrestrial carbon-cycle response to historical climate change (Huntzinger et al., 2017)".

Reviewer 2: • There is no definition of 'gamma' either.

Response: According to another reviewer's comments, we will remove $\gamma$-related contents in the revised manuscript.

Reviewer 2: • In many of the equations the equals sign is obscured. More effective

spacing will make these equations easier to read

Response: Agree.

References

Arora VK, Boer GJ, Eby M et al. 2013. Carbon–Concentration and Carbon–Climate Feedbacks in CMIP5 Earth System Models. dx.doi.org, 26 5289–314.

Anav, A., Friedlingstein, P., Kidston, M., Bopp, L., Ciais, P., Cox, P., ... & Zhu, Z. (2013). Evaluating the land and ocean components of the global carbon cycle in the CMIP5 Earth System Models. Journal of Climate, 26(18), 6801-6843.

Dai, Y., Dickinson, R. E., & Wang, Y. P. (2004). A two-big-leaf model for canopy temperature, photosynthesis, and stomatal conductance. Journal of Climate, 17(12), 2281-2299.

De Kauwe, M. G., Medlyn, B. E., Zaehle, S., Walker, A. P., Dietze, M. C., Wang, Y. P., ... & Wårlind, D. (2014). Where does the carbon go? A model–data intercomparison of vegetation carbon allocation and turnover processes at two temperate forest free‐air CO2 enrichment sites. New Phytologist, 203(3), 883-899.

De Kauwe, M. G., Keenan, T. F., Medlyn, B. E., Prentice, I. C., & Terrer, C. (2016). Satellite based estimates underestimate the effect of CO 2 fertilization on net primary productivity. Nature Climate Change, 6(10), 892.

Friedlingstein P, Cox P, Betts R et al. 2006. Climate–Carbon Cycle Feedback Analysis: Results from the C4MIP Model Intercomparison. Journal of Climate, 19 3337–53.

Huntzinger, D. N., Michalak, A. M., Schwalm, C., Ciais, P., King, A. W., Fang, Y., ... & Hayes, D. (2017). Uncertainty in the response of terrestrial carbon sink to environmental drivers undermines carbon-climate feedback predictions. Scientific Reports, 7(1), 4765.

Long, S. P., Ainsworth, E. A., Rogers, A., & Ort, D. R. 2004. Rising atmospheric carbon

dioxide: plants FACE the Future*. Annu. Rev. Plant Biol., 55 591-628.

Luo, Y., Sims, D. A., Thomas, R. B., Tissue, D. T., & Ball, J. T. 1996. Sensitivity of leaf photosynthesis to CO2 concentration is an invariant function for C3 plants: A test with experimental data and global applications. Global Biochemical Cycles, 10 209-222.

Luo, Y. and H.A. Mooney. 1996. Stimulation of global photosynthetic carbon influx by an increase in atmospheric carbon dioxide concentration. (In G.W. Koch and H.A. Mooney (eds.) Carbon Dioxide and Terrestrial Ecosystems. Academic Press, San Diego) pp. 381-97.

Smith, W. K., Reed, S. C., Cleveland, C. C., Ballantyne, A. P., Anderegg, W. R., Wieder, W. R., ... & Running, S. W. 2016. Large divergence of satellite and Earth system model estimates of global terrestrial CO2 fertilization. Nature Climate Change, 6 306-10.

Wang, Y. P., Law, R. M., & Pak, B. (2010). A global model of carbon, nitrogen and phosphorus cycles for the terrestrial biosphere. Biogeosciences, 7(7), 2261-2282.

Zhu, Z., Piao, S., Myneni, R. B., Huang, M., Zeng, Z., Canadell, J. G., ... & Cao, C. (2016). Greening of the Earth and its drivers. Nature climate change, 6(8), 791-795.

[Figure]

**Fig. 1.** $\beta$ values at different levels for various C3 plants at the year 2023 from CABLE-C only(a), CABLE-CN (b) and CABLE-CNP (c) simulations.

**Fig. 2.** Correlations between $\beta$\_GPP and $\beta$\_LAI, $\beta$\_NPP and $\beta$\_LAI, $\beta$\_cpool and $\beta$\_LAI from CABLE C-only (a)$\sim$(c), CABLE-CN (d)$\sim$(f) and CABLE-CNP (g)$\sim$(i) at the year 2023 across C3 plants.

[Figure]

**Fig. 3.** Temporal trends of the leaf-to-canopy scaling factors for sunlit leaves and shaded leaves of different C3 vegetation types from the CABLE-C only model from 2011 to 2100.

---

## Author Response (AR1)

**The point-by-point response to the reviews**

Response to reviews of manuscript "Leaf Area Index identified as a major source of variability in modelled $CO_2$ fertilization" (bg-2018-213).

Dear Editor,

We appreciate the insightful comments on our manuscript by all reviewers, and accordingly made substantial revision to improve this manuscript. We hope that our point-by-point responses below satisfactorily address the concerns raised by reviewers. The major proposed changes we made include:

(1) We included results from carbon-nitrogen (C-N) coupled and carbon-nitrogen-phosphorus (C-N-P) coupled simulations of CABLE to study how the $CO_2$ fertilization effects ($\beta$) at different levels change with nutrient limitations for different plant functional types. Our results from C-N and C-N-P coupled simulations support our original conclusion reached from C only simulations.

(2) We clarified the motivation and contribution of our study as the reviewers suggested.

(3) We carefully corrected language errors in our revised manuscript.

The original reviewers' comments are italicized and our response to the reviewers' comments follow.

**Response to Associate Editor's comments**

*Associate Editor: The revisions proposed in the response letter go a long way to address the critiques raised by reviewers and a duly revised manuscript is correspondingly welcomed. Adding the nutrient dynamics to the simulation system, and comparing those results to the original C-dynamics-only simulations, will add a good deal of value to the paper. The paper's contributions will also be deepened by including text from the response letters in the paper's results and discussion sections, where appropriate.*

Response: We deeply appreciate you for the encouragement and giving us an opportunity to revise our manuscript. We have included text from the response letters in the paper's results and discussion sections. Please see more details in our response comments to the reviewers.

*The work has the potential to make a valuable contribution, providing useful insights into the model mechanisms that govern terrestrial ecosystem responses to elevated CO2. However, as noted when handling the BG_2018_153 version of this manuscript, care must be taken to ensure that the paper's major interpretations and conclusions are framed to accurately match the methods and findings. My understanding of what this study mainly shows is that beta factors vary (a) across hierarchical levels of C-fluxes and stocks, and (b) across PFTs in a way that is largely attributable to variations in LAI dynamics across PFTs. This finding is framed as if it sheds light on across model spread in beta but that is not correct. The methods and results do not pinpoint LAI as a leading source of across-model spread or uncertainty in the land carbon cycle response to elevated CO2. The authors attempt to make a conceptual and motivational link between the across-model spread in beta factors shown in other studies and the findings presented in this paper however this is misleading. Consider this - it is possible that all of models in some MIP would show the same kind of across-PFT variation in beta, and that those models still having large across-model disagreement in the overall beta factors for different ecosystems and for the global biosphere. Some of the response comments still seem to misunderstand this point. Again, the paper has a contribution to make, particularly with the revisions in response to reviews, but I would reiterate that the introductory framing, interpretations and conclusions may still need to modified to match its methods and findings if the paper is to be ultimately accepted for publication.*

Response: Thanks a lot for your comments and suggestions. We agree with you that previous frame of the manuscript is a little confusing. We particularly focused on this issue in the revised manuscript. We have made corresponding adjustments in the Abstract (Line 20-21), Introduction (Line 95-100), Discussion (411-431) and Conclusion (Line 443-444) to make them more compatible with our methods and results.

**Response to reviewer#1**

*Reviewer 1: Li et al. use the CABLE model to explore the role of LAI in variability in the CO2 fertilisation response. The analysis has some interesting aspects which I'm sure will be of interest to the modelling community, in particular I thought fig 5 was interesting. However, I think the manuscript could be carefully revised for greater impact and insight. I have a number of specific points below but also 4 key issues with the analysis as presented:*

Response: We thank the reviewer for the positive comments. His suggestions are very important for improving our manuscript.

*Reviewer 1: 1. I don't understand the logic of using a model which simulates N and P cycles and*

*then switching this functionality off to understand the CO2 fertilisation response? In my eyes, this is one*

*of the great strengths of this model. So to not compare C against N*

*and P, or C against N, is a missed opportunity. Whilst I'm realistic enough to envisage*

*the authors won't rethink this strategy, I do feel this requires some further justification.*

Response: We appreciate the reviewer's critical comments. This concern has been raised by two reviewers. The reason why we didn't originally include nitrogen and phosphorus cycles in our previous study is that we tried to find the most important factor causing the variations of $\beta$ within and across different plant functional types (PFTs) with minimal confounding effects of other processes.

However, we totally agree with the reviewers that carbon-nutrient interactions should be considered when studying the $CO_2$ fertilization effects. C-N and C-N-P simulations were useful to explore how nutrients affect the patterns of and mechanisms underlying the variability of the $CO_2$ fertilization effects. We have conducted C-N and C-N-P coupled simulations of CABLE and added related descriptions and

analyses in the revised manuscript. Results show that although $\beta$ values at ecosystem levels are more variable with nutrient effects, LAI responses are still linearly correlated well with $\beta_{\text{GPP}}$, $\beta_{\text{NPP}}$ and $\beta_{cpool}$ across different C₃ PFTs in nutrient-coupled simulations as in C-only simulation, confirming the dominant role of LAI in regulating carbon cycle response under $CO_2$ fertilization. The related major changes in the revised manuscript are:

(1) In the Abstract part, we clarified our simulation designs with nutrient cycles (Line 21-23).

(2) In the Introduction part, we reviewed the effects of nutrient limitations on $CO_2$ fertilization effects in Line 85-93. We proposed the scientific questions related to carbon-nutrient interactions in Line 98-99.

(3) In the Materials and Methods part, we introduced how nutrient limitations were incorporated into carbon cycle in the CABLE model in Line 129-138. We clarified our experimental design and calculation in Line 140-154 and 212-215.

(4) In the Results part, we presented temporal trends of $\beta$ at ecosystem level for different vegetation types in C-N and C-N-P simulations in Line 223-228, Fig. 1b and Fig. 1c. We showed variations of intercellular $CO_2$ concentration and $CO_2$ compensation point under nutrient limitations in Line 236-244, Fig. S2 and Fig. S3. We compared $\beta$ values at different hierarchical levels in nutrient-coupled simulations in Line 261-267, Fig. 3b and Fig. 3c. Correlations between $\beta_{\text{GPP}}$ and $\beta_{\text{LAI}}$, $\beta_{\text{NPP}}$ and $\beta_{\text{LAI}}$, $\beta_{cpool}$ and $\beta_{\text{LAI}}$ in nutrient-coupled simulations were discussed in Line 277-280 and Fig. 4.

(5) In the Discussion part, we discussed about why magnitudes of biochemical and leaf-level $\beta$ with nutrient-limitations are similar to those without nutrient limitations in Line 314-315. We discussed the nutrient effects on the magnitudes and variations of $\beta_{\text{NPP}}$ in Line 381-391. We also discussed about the variability of nutrient-limited $\beta_{cpool}$ in Line 407-409.

(6) In the Conclusion part, we clarified our simulation designs and results with nutrient cycles (Line 444-446).

*Reviewer 1: It is stated that CABLE is largely RuBP-limited (line 179) and this point is given no further analysis. This is interesting and it isn't clear why this would be the case? Do the authors envisage that*

*this is also true of other models? I would suggest it isn't but would be keen to read the authors thoughts on this. Surely this shapes the analysis (responsiveness to CO2)? So it warrants more than a single sentence that simply says "not shown" ...*

Response: We thank the reviewer for this comment. It is an important prerequisite in our study. We agree it should be clarified in our manuscript. The formulation of leaf-level $\beta$ depends on the intercellular $CO_2$ concentration (Farquhar et al., 1980). Generally, photosynthesis rate is RuBP-regeneration limited (limited by light) when $CO_2$ concentration exceeds a certain level. And we coded a variable indicating which process (Rubisco activity, RuBP regeneration or sink) limits photosynthesis rate at each running step in the original CABLE code. Then we outputted this variable. We found photosynthesis rates are almost all limited by RuBP-regeneration process globally since 2011 when $CO_2$ concentration is 391 ppm. Then leaf-level biochemical $\beta$ can be expressed as an equation of intercellular $CO_2$ concentration and $CO_2$ compensation point. We didn't show the results because of the large volume of data (56560 model grids $\times$ 8760 hours in a year in total). Moreover, theoretical analysis by Luo and Mooney (1996) showed that leaf-level $\beta$ values are similar for either Rubisco- or RuBP-limited photosynthesis. We have added these points in the revised manuscript (Line 174-177).

*Reviewer 1: The paper is about CABLE but surely the aim is to make the result general (otherwise the title would have the word CABLE...)? However, I wonder if I was developing JULES or CLM, (etc) what my take home messages would be? The authors urge other modelling groups to repeat their analysis, but could they also make suggestions as to the implications for other modelling groups? How do these results help us to understand model responses to CO2? The CMIP5 concentration-carbon feedback factor?*

Response: We thank the reviewer for the suggestion to highlight the take-home messages more clearly. We believe our conclusions about the across-PFT variation of $\beta$ and the dominant role of LAI for the variability of $\beta$ in CABLE is generally applicable to other models. According to the comments from

the Associate Editor, in order to match our methods and findings, we will not extend the implication of our study to the uncertainty of CMIP5 concentration-carbon feedback factor. But we have discussed about the implication for across-PFT variation of $\beta$ in other models in the revised manuscript (Line 411-431).

*Reviewer 1: I didn't take much in the way of insight from the current section on this topic, i.e. section 4.3. For example, the authors assert that "It can be inferred that normalized leaf-level ð'I ˙ $Z_i$ values would diverge little across different land surface models as long as they use ...". Is that true? If the models had different levels of water stress (which they almost always do) they would get very different values of Ci even with the same model assumptions. As the authors also show, leaf temperature affects gamma_star, so I see no reason to assume that models would predict similar leaf temperatures. Leaf temperature itself is dependent on a whole range of assumptions. I've never seen any evidence that models with different architectures, with different assumptions about leaf –to-boundary conductance, etc, would predict similar leaf temperatures. If the authors disagree they should support these assumptions. The authors cite the Hasegawa et al study as an example of a consistent result of their conclusion. But wouldn't a number of the other model CO2 paperls that point to marked divergence argue otherwise. My sense is their conclusion here is too simplistic.*

Response: Thank the reviewer for pointing out this issue. We agree with the reviewer that different models have diverse levels of water stress on photosynthesis (De Kauwe et al., 2017). Water stress is applied to regulate stomatal conductance in many models (Rogers et al., 2017; Wu et al., 2018). For example, the CABLE model represents water stress by an empirical relationship based on soil texture and limits the slope of the coupled relationship between photosynthesis rate and stomatal conductance (Eq. (S11)). The influence of water stress is reflected by intercellular $CO_2$ concentration ($C_i$). Our results show modelled ratio of $C_i$ to atmospheric $CO_2$ concentration ($C_a$) is relatively constant for each PFT with $eCO_2$ and varies little among PFTs (Table 1). This modelling result is consistent with the concept of homeostatic regulations through photosynthetic rate and stomatal conductance (Pearcy and Ehleringer,

1984; Evans and Farquhar, 1991). Wong et al. (1979) showed plant stomata could maintain a constant $C_i/C_a$ across wide range of environmental conditions, including water stress condition. Therefore different vegetation types might have similar $C_i$ for a given $C_a$ in other models. Moreover, Luo and Mooney (1996) found that changing $C_i/C_a$ ratio from 0.6 to 0.8 caused less than variation of 0.08 in sensitivity of leaf photosynthesis to a unit of increase in $C_a$. We have added the above discussions into our revised manuscript (Line 419-429).

It's also true that different model might simulate different leaf temperatures as the reviewer pointed out. Sensitivity analysis in a previous study has shown that a ±5°C of leaf temperature changes caused approximately ±7 ppm changes in $\Gamma_*$, leading to variation of 0.12 to leaf-level $\beta$ (Luo and Mooney, 1996). The overall variation of leaf-level $\beta$ caused by variation in leaf temperature is still quite small compared with that of $\beta_{GPP}$. We have added the above discussions into our revised manuscript (Line 309-313).

Based on our literature review, only few studies like Hasegawa et al. (2017) have explored why different models simulated diverse responses of plant productivity to eCO₂. We will greatly appreciate it if the reviewer can show us some related references.

*Reviewer 1: The authors argue for the importance of LAI but don't really consider the role of allocation or turnover in great detail. Surely this is the key reason different models arrive at different LAI values? Even if you ignore changes in allocation/turnover due to CO2, this impacts on the scaling terms that the authors focus on.*

Response: The reviewer made a great point. Changes in LAI are related to changes in allocation/turnover under eCO₂. The response of allocation to eCO₂ will influence $\beta$ in two ways. The first way is through altering the portion of carbon allocated to leaf, then changing LAI. We have discussed more about this point in the revised manuscript: "Second, diverse allocation schemes influence the responses of LAI for

different PFTs. And, results from two FACE (Duke Forest and Oak Ridge) experiments indicate that the carbon allocated to leaves is decreased and more carbon is allocated to woods or roots at higher $CO_2$ concentration (De Kauwe et al., 2014). Unfortunately, CABLE has fixed allocation coefficients and likely overestimates LAI response, leading to overestimated responses of GPP, NPP and total carbon storage" (Line 364-368).

The second way is by changing the allocation pattern to plant organs with different lifespan, thereby altering carbon turnover time in plants and soil. It has been discussed in the revised version Line 397-401:"In this study and many other models, allocation coefficients are fixed over time (Walker et al., 2014). But allocation pattern to plant organs with different lifespan has been reported to change in response to $eCO_2$ in experiments, thereby altering carbon residence time in plants and soil (De Kauwe et al., 2014). Therefore, the fixed allocation scheme we adopted in this study might lead to some biases in simulating the response of carbon residence time to $eCO_2$".

*Reviewer 1: 4. The results are considered on a PFT level, but presumably they vary in interesting ways within a PFT (i.e. in space). Would this be worth showing or exploring further?*

Response: Actually in the previously submitted manuscript, we have analyzed within-PFT variations of $\beta$ at different levels in Table 1, Results 3.3, and Fig. S1-S3. In the revised manuscript, the related parts are in Line 215-217, 269-275, 280-282, 361-362, Table 2 and Fig. S5-S7.

*Reviewer 1:*

*Specific comments ==================*

*- Line 43: Could you explain the CO2 fertilising effect further? The text as written expects the casual reader has significant background knowledge for the second sentence of your manuscript.*

Response: Agree. We added the following sentences in the first paragraph: "Persistent increase of

atmospheric $CO_2$ concentration will stimulate plant growth and ecosystem carbon storage, forming a negative feedback to $CO_2$ concentration (Long et al., 2004; Friedlingstein et al., 2006; Canadell et al., 2007). This concentration-carbon feedback ($\beta$), also called the $CO_2$ fertilization effect, has been identified as a major uncertainty in modelling terrestrial carbon-cycle response to historical climate change (Huntzinger et al., 2017)" (Line 41-45).

*Reviewer 1: - Line 48: 4 or 4.5? What does that mean, do you mean 4 to 4.5? How can it be OR?*

Response: Sorry for the ambiguity. Actually, the contribution of $\beta$ is 4 times larger than that of carbon-climate feedback ($\gamma$) in Gregory et al. (2009) and Bonan and Levis (2010), but is 4.5 times larger in Arora et al. (2013). We have changed this sentence to "Some studies pointed out that the contribution of $\beta$ is 4 to 4.5 times larger, and more uncertain, than climate-climate feedback ($\gamma$) (Gregory et al., 2009; Bonan and Levis, 2010; Arora et al., 2013)" (Line 47-49).

*Reviewer 1: - Line 49: the reference to the Smith et al. paper ignores a technical comment on this paper: De Kauwe et al. (2016). Satellite based estimates underestimate the effect of CO2 fertilization on net primary productivity. Nature Climate Change, 6, 892-893. This is important as the authors are using this study to leverage their question. See also point on line 340.*

Response: We thank the reviewer for pointing to the related comment paper by De Kauwe et al. (2016). It is indeed an important reference to supplement the point we were trying to make. We have modified the last sentence in the first paragraph to "Though satellite products they used may underestimate the effect of $CO_2$ fertilization on net primary productivity (De Kauwe et al., 2016), the large disparity between models and FACE experiments gives us little confidence in making policies to combat global warming" (Line 51-53).

*Reviewer 1: - Line 51: it isn't "reality" - the satellite estimates are also model estimates.*

Response: Agree. See the response above.

*Reviewer 1: - Line 54: "increasing temperature in models" why is temperature being introduced as a factor here? Isn't the focus solely on the CO2 fertilisation effect rather than the than carbon-climate feedback factor? There are further studies cited in this paragraph which should be removed if the focus of this paper does not consider the carbon-climate feedback factor.*

Response: Agree. We have removed the $\gamma$-related part in the revised manuscript.

*Reviewer 1: - Line 67: Despite models using apparently similar photosynthesis models, Rogers et al. (A roadmap for improving the representation of photosynthesis in Earth system models. New Phytologist, 213, 22-42.) showed some important differences. It would be worthwhile highlighting this study in the context of the section of the text.*

Response: We thank the reviewer for sharing us this important reference. We have adjusted the sentence to a more accurate one: "The leaf-level $CO_2$ fertilization for $C_3$ plants is generally well characterized with models from Farquhar et al. (1980), and the basic biochemical mechanisms have been adopted by most land surface models although some models implement variants of Farquhar et al. (1980) (Rogers et al., 2017)" (Line 59-61). We also discussed about how those different implementations influence photosynthetic response in the Discussion: "Some models use variants of Farquhar photosynthesis model such as co-limitation approach described by Collatz et al. (1991). Inflection point from Rubisco- to RuBP- limited processes is an important control of the absolute values of photosynthetic response to $eCO_2$ (Rogers et al., 2017). However, the relative photosynthetic responses for different ecosystems will converge to a small range because the normalized photosynthetic response to $eCO_2$ only depends on estimates of intercellular $CO_2$ concentration ($C_i$), Michaelis-Menten constants ($K_c$, $K_o$) and $CO_2$ compensation point ($\Gamma_*$), and relative leaf-level responses are similar for either Rubisco- or RuBP-limited

photosynthesis (Luo et al., 1996; Luo and Mooney, 1996)" (Line 413-419).

*Reviewer 1: - Line 72: what does carbon storage have to do with this sentence?*

Response: Thanks for pointing out what we have missed. Besides NPP, allocation and carbon turnover process can influence carbon storage. We have changed this sentence to "However, the $CO_2$ fertilization effects are considerably more variable at canopy- and ecosystem-level than at the leaf-level, because a cascade of uncertain factors, such as soil moisture feedback (Fatichi et al., 2016), canopy scaling (Rogers et al., 2017), nutrient limitation (Zaehle et al., 2014), allocation (De Kauwe et al., 2014), and carbon turnover process (Friend et al., 2014) influence the responses of GPP, NPP and carbon storage" (Line 64-67).

*Reviewer 1: - Line 76/7: seems a narrow characterisation of the literature, the De Kauwe et al. 2014 study that the authors cite, explored these issues in depth.*

Response: We have added related references as the reviewer suggested: "LAI plays a key role in scaling leaf-level biogeophysical and biogeochemical processes to global scale responses in ecosystem models, and the representation of LAI in models causes large uncertainty (Ewert, 2004; Hasegawa et al., 2017). Models generally predict that LAI dynamics will respond to $eCO_2$ positively due to enhanced NPP and leaf biomass (De Kauwe et al., 2014). But how the increasing LAI in turn feeds back to ecosystem carbon uptake as a result of more light interception has not been discussed in previous research" (Line 73-78).

*Reviewer 1: - Line 81: Why would a high "basic" (delete basic) NPP necessarily lead to tropical regions having the highest stimulation by CO2? Wouldn't the opposite be expected? These regions have a high LAI and so would predominantly be light-limited and so have a more limited capacity to respond to CO2? Either way, the authors need to expand on this assertion.*

Response: We agree this sentence is not very clear. This sentence in this paragraph has now been changed into "The strongest absolute $CO_2$ fertilization effect has been found in tropical and temperate forests where the larger biomass presents than other regions. In comparison, the weakest response to $eCO_2$ occurs in boreal forests (Joos et al., 2001; Peng et al., 2014). But with gradual $eCO_2$, relative response in tropical forests might not be very high owing to light limitation caused by canopy closure (Norby et al., 2005)" (Line 81-85).

*Reviewer 1: - Line 89: Improved on what?*

Response: We will change this sentence to: "CABLE (version 2.0) is the Australian community land surface model (Kowalczyk et al., 2006) and incorporates CASA-CNP to simulate global carbon (C), nitrogen (N) and phosphorus (P) cycles (Wang et al., 2010; Wang et al., 2011)" (Line 103-104).

*Reviewer 1: - Line 124: The assumption that Jmax25 = 2 x Vcmax25. Did the authors consider varying this assumption? Other models would make quite different assumptions about this ratio.*

Response: It's true that the ratio of the maximum electron transport rate ($J_{max,25}$) to maximum photosynthetic capacity ($V_{cmax,25}$) are different in models (Rogers et al., 2017). But difference of this ratio will not change the conclusion because $\beta$ values in our study are normalized values, irrespective of $J_{max,25}$ or $V_{cmax,25}$.

*Reviewer 1: - Line 155: is there a citation, web link for "Community Climate System Model (CCSM) simulations"*

Response: We have added a citation "Hurrell, J. W., Holland, M., Gent, P., Ghan, S., Kay, J. E., Kushner, P., Lamarque, J.-F., Large, W., Lawrence, D., Lindsay, K., Lipscomb, W. H., Long, M. C., Mahowald, N., Marsh, D. R., Neale, R. B., Rasch, P., Vavrus, S., Vertenstein, M., Bader, D., Collins, W. D., Hack,

J. J., Kiehl, J., and Marshall, S.: The community earth system model: a framework for collaborative research, Bull. Am. Meteorol. Soc., 94, 1339–1360, 2013." (Line 145, 571-574).

*Reviewer 1: - Line 168: the definition of S (line 171) needs to be moved up to this line.*

Response: Agree. Please see related changes in Line 191-194.

*Reviewer 1: - Line 215: just to clarify when the authors say total carbon storage - do they mean the soils too? Or just the plant? Or just the foliage pool? The equation isn't very clear. This also makes Fig 1 hard for me to interpret as I'm unclear what is being shown, I'm going to assume it is total plant carbon...*

Response: Total carbon storage is the sum of plant, litter and soil carbon pools. We have made it clear in the revised manuscript (Line 158-159).

*Reviewer 1: - Fig 1. Does it make sense to normalise these PFT lines? The authors say they decline but the magnitudes differ, the point is that the initial starting points are different too. This makes it hard for the eye to gauge.*

Response: Indeed, the $CO_2$ fertilization effects at different levels in our manuscript are all normalized values. See Eq. (3) in Line 163.

*Reviewer 1: - As a general comment the results need work, particularly in terms of transition text. For example 3.1 talks about the temporal trend in Bcpool and then switches immediately to the Ci/Ca ratio in 3.2? It is hard to follow the logic of the transition, is there is meant to be any connection for the reader?*

Response: Section 3.1 is about $\beta$ at ecosystem level, showing that $\beta$ values differ among different PFTs and decrease over time. It stimulates the following study that calculating $\beta$ values from leaf

biochemical level to ecosystem level in order to identify the key processes for the divergent $\beta$ at ecosystem level. We have added one transition sentence at the beginning of Section 3.2: "To reveal which processes cause the large disparity of $\beta$ across PFTs as shown in Fig. 1, we first compared intercellular $CO_2$ concentration $(C_i)$ and $CO_2$ compensation point in the absence of day respiration $(\Gamma_*)$, which are critical parameters for leaf-level biochemical response" (Line 230-232).

*Reviewer 1: - What is the point of Fig. 2? It isn't clear what this figure has to do with the story of the paper?*

Response: Please see the above response.

*Reviewer 1: - The text around line 261 which refers to Fig 4 could do with further explanation. I personally don't find this particularly surprising, but the reader isn't offered much as the way of explanation. Presumably the change in slope as you move from B_GPP to B_NPP relates to respiration assumptions and then to B_cpool, allocation/turnover assumptions? I think the authors could go further in assisting the reader with interpretation. As currently written, the text simply highlights that the slope changes.*

Response: After thinking carefully about this concern, we agree that the slopes of the three fitting lines are not making much sense so we have removed this sentence in the revised manuscript.

*Reviewer 1: - I think figure 5 is very interesting.*

Response: Thank the reviewer for the positive comment.

*Reviewer 1: - Line 290: I think this discussion of Fig S5 is interesting but I'm not sure I follow the interpretation? The LAI is the emergent outcome of the model assumptions - 1 leaf, 2leaf, multi-layer.*

*Of course this assumption will lead to differences? But why you do the analysis on the leaf-level? Surely you're interested in the emergent outcome – the LAI. Most likely I simply misunderstood this point but I think it could also be explained further as it seems like an important point the authors are making.*

Response: The discussion of Fig. S5 (Fig. S8 in the revised manuscript) is primarily triggered by the comparison between our results and Hajima et al. (2014). We believe that leaf-level photosynthesis cannot be simplified as GPP/LAI for CMIP5 models as Hajima et al. (2014) did since CMIP5 models use different canopy structure such as big-leaf, two-leaf or multiple-layer. Most previous studies focused on variation in $\beta$ for the land carbon storage, the standard definition of $\beta$ as in Friedlingstein et al. (2006). But diagnosis of leaf-level response has not been attempted by modelling groups before. And CMIP5 model outputs have limited information for identifying mechanisms for model uncertainty since there are no leaf-level process outputs. So we did the analysis on the leaf-level processes. We have reorganized this part to make our manuscript more concise: "To identify the source of uncertainty of $\beta$ in CMIP5 models, Hajima et al. (2014) decomposed $\beta$ into several carbon cycle components. They used GPP divided by LAI (GPP/LAI) as a proxy to represent leaf-level photosynthesis for CMIP5 models, since there are no leaf-level process outputs of these models. They found the sensitivities of GPP/LAI to $eCO_2$ diverged a lot among models. This calculation is likely debatable for ignoring different canopy structure used by each CMIP5 model such as big-leaf, two-leaf or multiple-layer. Our results just show that the sensitivities of GPP/LAI are different from our mechanistic calculation of leaf-level $\beta$ for different PFTs (Fig. S8)" (Line 317-322).

*Reviewer 1: - Line 295: I don't fully follow that interpretation? Your differences in Ci/Ca were small across PFTs? And the differences in leaf temp would be expected between PFTs? Certainly, fig 2 doesn't show any within PFT variation.*

Response: We would like to express that the leaf-level $\beta$ computed in our study can be mechanistically traced back to intercellular $CO_2$ concentration and leaf temperature. Since Fig. 2 shows the results across

different PFTs, we have changed this sentence to: "Another advantage of our calculation of leaf-level $\beta$ is that the reason for the divergence of leaf-level $\beta$ across PFTs can be traced back to the difference from $C_i$ and leaf temperature as shown in Fig. 2" (Line 322-324).

*Reviewer 1: - Line 362: This is an assumption of the model and might not necessarily be true!*

Response: Agree. We have added the following sentences in the manuscript: "However, FACE experimental results indicate that CUE values under $eCO_2$ are not changed in N-limited Duke site (Hamilton et al., 2002; Schäfer et al., 2003), increase in fertile POPFACE site (Gielen et al., 2005) or decrease in fertile ORNL site (DeLucia et al., 2005). Thus, representations of nutrient effects on GPP and autotrophic respiration in land surface models should be carefully calibrated with experimental data (DeLucia et al., 2007)" (Line 382-386).

*Reviewer 1:*

*Technical corrections =====================*

*- Abstract: "vegetation types is 0.15-0.13", presumably you meant 0.13 to 0.15? Also,*

*why don't the other variables (e.g. BetaGPP) have ranges too?*

Response: Yes, we meant 0.13 and 0.15 for shaded leaf and sunlit leaf, respectively. We did not show the coefficients of variation in Abstract in the revised manuscript. In this sentence, we did not differentiate sunlit leaves and shaded leaves for canopy GPP, so there is only one value for $\beta_{\mathrm{GPP}}$.

*Reviewer 1: - First line of the introduction, makes no sense. You can't start a sentence with Terrestrial carbon sink and then a comma.*

Response: Agree. We have changed the first sentence to: "Terrestrial ecosystems take up roughly 30%

of anthropogenic $CO_2$ emissions, and is of great uncertainty and vulnerable to global climate change (Cox et al., 2000; Le Quéré et al., 2018)" (Line 40-41).

*Reviewer 1: - Line 45: In Coupled -> In the Coupled*

Response: Agree (Line 45).

*Reviewer 1: - Line 138: In CABLE model -> in the cable model*

Response: Agree (Line 112).

**Response to reviewer#2**

*Reviewer 2:*

*Synopsis:*

*In this paper, the authors run CABLE for seven C3 vegetation types, without nutrient cycling, and calculate CO2 fertilization for the RCP 8.5 scenario. CCSM simulations from 1901 to (the paper says 1910; I assume they mean 2010) holding carbon-climate feedbacks constant (driving the model with the averaged meteorology-I'm guessing average annual cycle, although the authors do not say) and feeding CABLE increasing CO2 concentration from the CCSM RCP 8.5 results.*

*They find that CO2 fertilization differs between PFTs, and decreases with time during the period 2011-2100. Fertilization is relatively constant both between PFTs and when the calculation is made on a per-unit leaf level, and shows much larger diversity both across PFTs and when the CO2 fertilization is calculated on a unit-leaf vs. integrated canopy basis. The authors close with the claim that simulated LAI is critical to the calculation of CO2 fertilization in climate simulations.*

Response: We thank the reviewer for the time she or he spent on reviewing our manuscript. The above paragraphs are a good summary of what we did for this study. While most of the summary is accurate, we would like to clarify here that CABLE model has been run from 1901 to 2100. Before that, CABLE was spun up by using meteorological forcing from 1901 to 1910 repetitively until a steady state was achieved. And we indeed used the average annual cycle of meteorological forcing data to fix carbon-climate feedbacks. We have clarified these points in our revised manuscript (Line 140, 144-146).

*Reviewer 2:*

*Review:*

*I have 2 major problems with this paper. Either one by itself, I believe, is fatal, but taken*

*together I cannot make any recommendation for this paper other than rejection.*

Response: We have made significant changes to the both the text and simulations (by including nutrient-enabled CABLE simulations). We hope that major criticisms have been addressed in our revised manuscript.

*Reviewer 2:*

*Problem #1: There is a rich body of literature from the FACE experiments that claims, pretty much unequivocally, that nutrient cycling and/or limitation becomes more and more important to CO2 fertilization as CO2 concentrations rise. Yet, in this experiment CABLE is run with nutrient cycling turned off!*

Response: We may not have made our research objective quite clear. In the revised manuscript, we now stated that our study was to examine how variability, as measured by coefficient of variation (CV) within and across different plant functional types (PFTs), in the $CO_2$ fertilization effect (i.e., CV of $\beta$) changes from leaf to canopy GPP, ecosystem NPP and total carbon storage levels. Our study was not intended to quantify the $CO_2$ fertilization effect itself. We have clarified our goal in the revised manuscript Line 95-

100.

We agree with the reviewer that nutrient limitations are universally observed in experiments. Nutrient cycling influences the $CO_2$ fertilization effect. When comparing the carbon-only simulations with nutrient (N, or NP) enabled simulations using CABLE, we find that the previous conclusion about CV of the $CO_2$ fertilization effects was not significantly changed. Thus, we hope this reviewer will re-evaluate our manuscript, particularly in light of significant work in including the results from additional new CABLE nutrient-enabled simulations.

*Reviewer 2: Coskun et al. (2016) and references therein has a nice summary of both Free-Air CO2 Enrichment (FACE) as well as Open-Top Chamber (OTC) experiments. Smith et al. (2015) discusses the divergence between multiple models and a satellite-derived product that underscores the importance of the interaction between nutrient cycling and CO2 fertilization. Many of these studies focus on N limitation, although some research has indicated that P limitation is a factor as well (e.g. Hasegawa et al., 2016). These, and other studies, all conclude that understanding of CO2 fertilization requires taking nutrients into account.*

Response: We thank the reviewer for showing us these important references. We have cited related references in our revised manuscript (Line 49-51, 85-86). Again, our study was not to quantify the $CO_2$ fertilization effect itself but to understand what caused changes in CV of $\beta$. Running CABLE without or with nutrient limitation reached a similar conclusion as shown below.

*Reviewer 2: I have to confess that I was very surprised when I read that the authors ran the version of CABLE without nutrient cycling included. I am not a FACE 'expert', but even I am aware of the amount of research that has concluded that nutrient cycling is critical to understanding ecosystem-level response to higher atmospheric CO2. I found it very suspicious that nutrients were excluded from the study. Why, when there is this large body of work demonstrating the nutrient cycling is critical to understanding CO2*

*enrichment, would nutrients be turned off in the model? The authors claim that nutrients were turned off for 'simplicity', but the obvious answer, and one that I suspect to be the truth, is that the authors did run CABLE with nutrient cycling, and model pathology and/or unrealistic results ensued.*

Response: We thank the reviewer for the critical comments and his/her insistence on the necessity of nutrient-coupled simulations. We absolutely agree with the reviewer that the $CO_2$ fertilization effect (or $\beta$) could be more realistically represented with nutrient limitations considered. The reason why we didn't originally include nitrogen and phosphorus cycles in our previous study is that we tried to find the most important factor causing the variations of $\beta$ within and across different vegetation types with minimal confounding effects of other processes. Per the suggestions from the two reviewers, we tested whether the patterns of and mechanisms underlying the variability of $\beta$ for C-only simulation still hold for nutrient-coupled simulations. We have added results and analyses from C-N and C-N-P coupled simulations of CABLE in the revised manuscript. The related major changes in the revised manuscript are:

(1) In the Abstract part, we clarified our simulation designs with nitrogen cycles (Line 21-23).

(2) In the Introduction part, we reviewed the effects of nutrient limitations on $CO_2$ fertilization effects in Line 85-93. We proposed the scientific questions related to carbon-nutrient interactions in Line 98-99.

(3) In the Materials and Methods part, we introduced how nutrient limitations were incorporated into carbon cycle in the CABLE model in Line 129-138. We clarified our experimental design and calculation in Line 140-154 and 212-215.

(4) In the Results part, we presented temporal trends of $\beta$ at ecosystem level for different vegetation types in C-N and C-N-P simulations in Line 223-228, Fig. 1b and Fig. 1c. We showed variations of intercellular $CO_2$ concentration and $CO_2$ compensation point under nutrient limitations in Line 236-244, Fig. S2 and Fig. S3. We compared $\beta$ values at different hierarchical levels in nutrient-coupled simulations in Line 261-267, Fig. 3b and Fig. 3c. Correlations between $\beta_{GPP}$ and $\beta_{LAI}$, $\beta_{NPP}$ and $\beta_{LAI}$, $\beta_{cpool}$ and $\beta_{LAI}$ in nutrient-coupled simulations were discussed in Line 277-280 and Fig. 4.

(5) In the Discussion part, we discussed about why magnitudes of biochemical and leaf-level $\beta$ with nutrient-limitations are similar to those without nutrient limitations in Line 314-315. We discussed the nutrient effects on the magnitudes and variations of $\beta_{\text{NPP}}$ in Line 381-391. We also discussed about the variability of nutrient-limited $\beta_{cpool}$ in Line 407-409.

(6) In the Conclusion part, we clarified our simulation design and results with nutrient cycles (Line 444-446).

We found $\beta$ values at canopy and ecosystem levels in C-N and C-N-P simulations diverge in a way that is largely attributable to variations in LAI responses across $C_3$ vegetation types, as in C-only simulation. It should be noted that nutrient effects add more variations to $\beta$ values at ecosystem level compared with C-only simulation (Fig. 3 in the revised manuscript). The CABLE-CN and CABLE-CNP simulations add more layers of complexity to understand the primary mechanisms underlying the divergence of $\beta$ at different levels and in different ecosystems although the conclusion is similar with that reached from running carbon-only CABLE. This finding proves that our previous design that turning off the nutrient cycles in model simulation to identify the most critical carbon cycle processes is reasonable. But we agree with the reviewer that adding nutrient cycle will further strengthen our conclusions.

*Reviewer 2: It may have been possible to evaluate a nutrient run, even if the results were unrealistic, and evaluate how atmospheric CO2 levels and nutrients interact in CABLE. The results may have provided an opportunity to evaluate or comment on the divergence of models in their predictions of atmospheric CO2 levels and source/sink strength (e.g. Friedlingstein et al., 2006, 2014). By not including the critical nutrient interaction, I'm not sure that the results presented here give the reader any insight into how ecosystems might realistically respond to increasing future CO2 levels in the atmosphere.*

Response: We agree with the reviewer that $CO_2$ and nutrient interactions could cause the divergence of models. Our new results with CABLE-CN and CABLE-CNP show that CV of $\beta$ is much higher than

that with CABLE-C only for NPP and total carbon storage (Fig. 3 in the revised manuscript). However, the objective of our study is not to evaluate nutrient effects on carbon cycle under $CO_2$ fertilization. As we have stated before, our study is to identify mechanisms underlying expanding CV from biochemical and leaf levels to canopy GPP, ecosystem NPP and carbon pool. All the three versions of CABLE point to the same mechanism, which is LAI as the major source of variability in modelled $CO_2$ fertilization.

*Reviewer 2:*

*Problem #2: Without carbon-climate feedbacks and nutrient cycles, I don't think a model actually has to be run to determine CO2 fertilization. You can probably perform the calculation directly from the equations in the code. Between models there will be some differences:*

*• Is the model an enzyme-kinetic model (Farquhar et al., 1980; Michaelis-Menten*

*kinetics), or light-response (e.g. VPRM, Mahadevan et al., 2008)?*

*• how is stomatal conductance calculated? Does it use Ball-Berry, with a dependence on relative humidity, or Leuning, which uses VPD? How is transpiration coupled to*

*photosynthesis?*

*• What are the parameter values for Vcmax for a given PFT?*

*• What determines phenology? Is allocation static, or, if it is dynamic, how does it change during the year and in response to what?*

*I believe it would be possible to determine the constraints on CO2 fertilization for a suite of models without actually running any of them.*

Response: We agree with the reviewer that these assumptions and processes are key to modelling terrestrial carbon-cycle responses to $eCO_2$. The reviewer is very knowledgeable to identify those key ecosystem carbon-cycle processes. In this comment alone, the reviewer mentioned more than 10 processes that influence photosynthesis. We were very curious how the reviewer could "perform the calculation directly from the equations in the code" to evaluate all those 10 processes and to gain a mechanistic understanding of what causes the change of $\beta$ values. Obviously, we did not understand

how to perform the calculation of $\beta$ directly from the model equations as the reviewer suggested. Even if we could calculate based on several equations, the results might not truly reflect model mechanisms for variabilities of the $CO_2$ fertilization effects within and across vegetation types. Because carbon-cycle processes are tightly coupled with radiation transfer, energy balance, nutrient interactions and water cycles in a land surface model. For example, leaf temperature and intercellular $CO_2$ concentration are two important variables for leaf-level $\beta$, which are collectively controlled by air temperature, radiation transfer and humidity. We were not sure if the reviewer meant to construct a simplified model or emulator to mimic the complex land surface models, it is worthy trying but we were not confident that the simplified approach could reveal model mechanisms.

Nevertheless, we ran a well-evaluated land surface model and outputted process-level variables such as intercellular $CO_2$ concentration, LAI, GPP, NPP, and ecosystem carbon storage for all land cells, as many analyses have done based on C4MIP and CMIP5. Combining previous theoretical analysis, we have shown that CV of $\beta$ is small for biochemical and leaf-level photosynthesis but large for canopy GPP, ecosystem NPP and carbon pools.

*Reviewer 2: It is axiomatic that leaf-to-canopy scaling (LAI) is critical to total $CO_2$ fertilization amount. Every model that I am aware of calculates biophysics on a per-unit-are basis and then scales to the canopy level either by summing over sunlit/shaded leaves (and PFTs) or integrating from leaf to canopy scale along the lines of Sellers (1985, 1992)(OK, a gap model like ED2 may be a little different). Canopies with an LAI close to 1 (think of grasslands) will not see much difference from unit- to canopy-scale, more dense canopies (like forests) will.*

Response: We are happy that the reviewer agrees with us that LAI is critical for plant productivity. Many models exhibit increasing LAI trends under $CO_2$ fertilization (Zhu et al., 2016). However, to what extent the increasing LAI feeds back to ecosystem response to $eCO_2$ is not clear. Our study for the first time calculated $\beta$ from leaf biochemical level to ecosystem level, and found the LAI response to $eCO_2$ is the

dominating factor for variabilities of the $CO_2$ fertilization effects at canopy and ecosystem levels within and across $C_3$ vegetation types, namely the global $CO_2$ fertilization effects are very sensitive to the LAI responses. The value of our study is that it can urge modelling groups to improve the representation of LAI in land surface models, for example by calibrating allocation coefficients and specific leaf area (SLA) based on FACE experimental results (De Kauwe et al., 2014), so as to realistically predict concentration–carbon feedback.

*Reviewer 2: If there is a large divergence between models in LAI (and GPP) for a given PFT, or if there is a large trend in one model's LAI for a given PFT during a climate run, then these might be valid topics of analysis. Finding that LAI is critical to canopy-level CO2 fertilization (without nutrients being considered) does not really bring anything new to the field.*

Response: Our results may not be much new for this reviewer but the key message from our study is still crucial for the community to improve land modelling. Actually, in our previous manuscript we have cited a paper showing CMIP5 models have simulated diverse GPP and LAI values during 1985-2006. And both GPP and LAI have been overestimated for most CMIP5 models according to observations (Anav et al., 2013). Satellite and modelled LAI both have experienced significant increasing trends during historical period as reported by Zhu et al. (2016). However, how the uncertainty and increasing trend of LAI contribute to modelled plant productivity and ecosystem carbon storage have not been discussed in previous research. Our study fills this gap and indicates the $CO_2$ fertilization effects are very sensitive to LAI responses. The merit of our study is that we systematically diagnose model processes and find LAI is the most important factor in modelling the $CO_2$ fertilization effects, to which modelers should pay greater attentions and efforts in the future research.

As inspired by the reviewer, we added a paragraph reviewing the latest reports about trends of LAI in the Introduction part in the revised manuscript (Line 71-79). We also discussed the uncertainty of modelled LAI in the Discussion part (Line 338-348).

*Reviewer 2: Sunlit and shaded leaf partitioning is fairly well-constrained and sunlit LAI can never get much above 1 to 1.5 or so even under the most direct-sun conditions. Solar angle and leaf angle distribution make it possible to exceed an LAI value of one. I know that CLM has had issues with shade leaf LAI becoming excessively large. The authors do not discuss total LAI in CABLE during their fertilization runs, and this makes me suspicious-if their shade-leaf LAI is becoming unrealistically large, that might be a reason why fertilization strength decreases with time; increase in the amount of sunlit leaf may result in large change in GPP, but once sunlit LAI is filled, any additional canopy growth will be as shade LAI, and GPP increase will be attenuated.*

Response: We appreciate the reviewer for the insightful comments. Actually, we did analyze total LAI change in the previous supplementary material Fig. S6 (Fig. S1 in the revised version). LAI value of evergreen broadleaf forest increases with time but gradually saturates at the prescribed maximum value. LAI values of other plant types also increase but are far below the prescribed maximum values at 2100. To address the reviewer's concern about the magnitudes and changes of sunlit and shaded leaf LAI (we called the scaling factors in our manuscript according to the standard definition in the CABLE model), we plotted temporal trends of the scaling factors for sunlit leaves and shaded leaves in CABLE-C only simulation (Fig. S9 in the revised version). Results show that the magnitudes of the scaling factors for shaded leaves are greatly larger than those for sunlit leaves for all $C_3$ plants. This is because in models it is usually defined that portion of sunlit leaves decreases exponentially with increasing LAI ($f_{sun}$= exp (-$k_b$LAI)) (Dai et al., 2004). The scaling factors for sunlit leaves are below 1 as the reviewer stated. And the scaling factors for sunlit leaves of evergreen broadleaf forest, evergreen needleleaf forest and deciduous broadleaf forest gradually saturate with eCO$_2$. We discussed temporal changes of scaling factors for sunlit and shaded leaves in revised manuscript Line 330-332: "This is because the portion of shaded leaves increase exponentially with increasing LAI (Fig. S9), leading to a rapid change of shaded leaf GPP. While for sunlit leaves, GPP shows a saturating response because of the decreasing portion of sunlit leaves with increasing LAI (Dai et al., 2004)".

The increasing portion of shaded leaves will lead to the attenuation of GPP increase as the reviewer mentioned. And we believe that saturation of GPP is jointly controlled by biochemical enzyme kinetics and canopy closure. The mechanisms for leaf-level saturation have been discussed in detail in Luo et al. (1996) and Luo and Mooney (1996).

*Reviewer 2: I just don't think there's anything new here. Without nutrient cycling the CO2 fertilization results don't have much meaningful application, and the fact that leaf-to-canopy scaling is important has been known for a long time.*

Response: We are sorry that the scientific value of our study has not been fully recognized by the reviewer. We have run CABLE with coupled carbon-nitrogen-phosphorus cycles as suggested. Our original conclusion still stands. Although leaf-to-canopy scaling has been known for a long time, no study has done before as we did in this study to evaluate variation of $\beta$ from biochemical and leaf levels to canopy and ecosystem scales. The leaf-to-canopy scaling is a basis of our study but the conclusion of our study goes far beyond it.

Here, we have strengthened our contributions through the following ways:

(1) Analyzing the $CO_2$ fertilization effects at different levels with C-N and C-N-P interactions for different $C_3$ plant functional types (PFTs) in the CABLE model to evaluate whether our conclusions are still valid under nutrient limitations.

(2) In the Introduction and Discussion part, we have clarified that our study was aimed to understand the variability of $CO_2$ fertilization effects from biochemical to ecosystem levels and the dominant factor (Line 95-100). Our mechanistic study, for the first time, shows that $\beta$ values vary at different hierarchical levels across C-fluxes and stocks, and across PFTs in a way that is largely attributable to variations in LAI dynamics at canopy and ecosystem levels. This finding is of significance in light of the uncertainty and increasing trends of modelled LAI reported by recent research. Our finding

can stimulate modelling groups to focus more on uncertainty arising from processes related to LAI, and use FACE experiments to narrow the uncertainty of land model predictions.

(3) We believe our conclusions about the across-PFT variation of $\beta$ and the dominant role of LAI for the variability of $\beta$ in CABLE is generally applicable to other models. Our analyses can inspire other modelling groups to explore mechanisms for the variability of $\beta$ from different hierarchical levels (Line 411-441 in the revised manuscript).

*Reviewer 2:*

*Specific comments:*

*• English prose and grammar, while readable, need attention. There are multiple places, too many to list, where errors exist.*

Response: We have carefully revised the manuscript and improved the language in the revised version.

*Reviewer 2: • There is no explanation for what eCO2 is (elevated CO2). Don't assume all your readers know the definition.*

Response: Agree. We defined it in "The response of ecosystem carbon cycle to elevated $CO_2$ ($eCO_2$) is primarily driven by stimulation of leaf-level carboxylation rate in plants (Polglase and Wang, 1992; Long et al., 2004; Heimann et al., 2008)" (Line 55-56).

*Reviewer 2: • There is no definition of 'gamma' either.*

Response: According to another reviewer's comments, we removed $\gamma$-related contents in the revised manuscript.

*Reviewer 2: • In many of the equations the equals sign is obscured. More effective spacing will make*

Response: Agree. To make our manuscript more clear and concise, we have moved the basic equations for photosynthesis and complex mathematical derivation to the supplementary materials in the revised manuscript. And we used the generalized equation of $\beta$ at the beginning of Section 2.3 (Line 163).

**Supplementary materials**

(1) Page 1 lines 2-46: we described photosynthesis module in the CABLE model.

(2) Page 3 lines 47-66: we described mathematic derivations of big-leaf β.

(3) Page 6: we presented LAI trends of C-N and C-N-P simulations in Fig. S1.

(4) Page 7: we presented responses of $C_i$ and $\Gamma_*$ for C-N simulation in Fig. S2.

(5) Page 8: we presented responses of $C_i$ and $\Gamma_*$ for C-N-P simulation in Fig. S3.

(6) Page 10-12: we presented the correlations of $\beta_{\text{GPP}}$ and $\beta_{\text{LAI}}$, $\beta_{\text{NPP}}$ and $\beta_{\text{LAI}}$, $\beta_{cpool}$ and $\beta_{\text{LAI}}$ at the year 2023.

(7) Page 13: we compared leaf-level $\beta$ calculated through biochemical parameters $C_i$ and $\Gamma_*$ for sunlit leaf ($\beta\_\text{psun}$) and shaded leaf ($\beta\_\text{psha}$) and sensitivities of GPP/LAI ($\beta\_\text{GPP/LAI}$) for different $C_3$ PFTs at the year 2023.

(8) Page 14: we added time trends of leaf-to-canopy scaling factor for sunlit leaves and shaded leaves (Fig. S9).

[revised manuscript text omitted]

230 variables were calculated as the normalized sensitivitysensitivities of those variables to eCO$_2$.

Equ. (4) and (5) can be simplifiedatmospheric $CO_2$ concentration ($C_a$) as: $\beta_V$:

$$A_c = v_{cmax,25} * f_{vcmax}\left(T_f\right) * \frac{c_i - \Gamma_*}{c_i + K_c(1 + C_o - K_O)} * S = a_c * S \tag{13}$$

$$A_q = j_{cmax,25} * f_{jcmax}\left(T_f\right) * \frac{c_i - \Gamma_*}{c_i + 2\Gamma_*} * S = a_q * S \tag{14}$$

$$\beta_V = \frac{1}{V} * \frac{dV}{dC_a} \tag{3}$$

235 Where $a_c$ and $a_q$ represent leaf-level Rubisco- and RuBP-limit photosynthesis rates respectively:

$$a_c = v_{cmax,25} * f_{vcmax}\left(T_f\right) * \frac{c_i - \Gamma_*}{c_i + K_c(1 + C_o - K_O)} \tag{15}$$

$$a_q = j_{cmax,25} * f_{jcmax}\left(T_f\right) * \frac{c_i - \Gamma_*}{c_i + 2\Gamma_*} \tag{16}$$

*S* indicates the scaling factor that scales fluxes at  in the  denominator represents average annual value of $S_{sun}$, $S_{sha}$, LAI, GPP, $GPP_{sun}$, $GPP_{sha}$, NPP and ecosystem carbon storage between two consecutive years. Subscripts:

$$S_{sun} = \frac{1-\exp[-LAI(k_n+k_b)]}{k_n+k_b} \tag{17}$$

For shaded leaves:

$$S_{sha} = \frac{1-\exp(-k_n LAI)}{k_n} - \frac{1-\exp[-LAI(k_n+k_b)]}{k_n+k_b} \tag{18}$$

where subscripts "*sun*" and "*sha*" denote the sunlit and shaded components

The  dV is the  difference of these variables between two consecutive years. $dC_a$ ÷

$$\beta_{p_{sun}} = \frac{1}{p_{sun}}*\frac{dp_{sun}}{dC_a} = \frac{1}{a_{qsun}}*\frac{da_{qsun}}{dc_{isun}}*\frac{dc_{isun}}{dC_a} = \mathcal{L}_{sun}*\frac{dc_{isun}}{dC_a} \tag{19}$$

$$\beta_{p_{sha}} = \frac{1}{p_{sha}}*\frac{dp_{sha}}{dC_a} = \frac{1}{a_{qsha}}*\frac{da_{qsha}}{dc_{isha}}*\frac{dc_{isha}}{dC_a} = \mathcal{L}_{sha}*\frac{dc_{isha}}{dC_a} \tag{20}$$

Where  is the  difference of corresponding $C_a$. The unit of $\beta_V$ is ppm[-1]. It should be noted that $\beta_V$ is the relative response, which is similar to the traditional definition of $\beta$ factor by Bacastow and Keeling (1973), but different from the carbon-concentration feedback parameter in Friedlingstein et al., (2006). The relative response facilitates the comparison among PFTs with different initial biomass and the comparison across carbon fluxes and storages with different units.

Leaf biochemical response ($\mathcal{L}$) was first proposed by Luo et al. (1996). $\mathcal{L}$ function is the normalized response of leaf photosynthesis rate to a small change in  intercellular CO₂ concentration ($C_i$) and has been suggested to be an invariant

function for C$_3$ plants grown in diverse environments. The rate of photosynthesis is typically RuBP-regeneration-limited under high $CO_2$ concentration. We found photosynthesis rates are almost all limited by RuBP-regeneration process globally under RCP8.5 scenario since 2011 when $CO_2$ concentration exceeds 390 ppm. Besides, theoretical analysis by Luo and Mooney (1996) showed that biochemical responses are similar for either Rubisco- or RuBP-limited photosynthesis. In this study, $\mathcal{L}$ can be used to indicate leaf biochemical response to eCO$_2$. For sunlit leaf and shaded leaf, $\mathcal{L}$ is formulations of $\mathcal{L}$ under RuBP-regeneration-limitation are defined as:

$$\mathcal{L}_{sun} = \frac{1}{a_{qsun}} * \frac{da_{qsun}}{dc_{isun}} = \frac{3 * \Gamma_{*sun}}{(c_{isun}+2*\Gamma_{*sun})(c_{isun}-\Gamma_{*sun})} \tag{21(4)}$$

$$\mathcal{L}_{sha} = \frac{1}{a_{qsha}} * \frac{da_{qsha}}{dc_{isha}} = \frac{3 * \Gamma_{*sha}}{(c_{isha}+2*\Gamma_{*sha})(c_{isha}-\Gamma_{*sha})} \tag{22(5)}$$

In this study, $\Gamma_{*sun}$ and $\Gamma_{*sha}$ are yearly average $CO_2$ compensation points in the absence of day respiration for sunlit leaf and shaded leaf, respectively. Intercellular CO$_2$ concentration ($c_i$) $c_i$ varies significantly at sub-daily, intra-annual and inter-annual basisbases. We're interested in how $c_i$ responds to eCO$_2$ on an inter-annual basis. So, we first outputted hourly $c_i$ then calculated yearly GPP-weighted average $c_i$ for sunlit leaf ($c_{isun}$) and shaded leaf ($c_{isha}$).

CanopyThen leaf-level $\beta_{GPP}\beta_p$ is defined as:

$$\beta_{GPP} = \frac{1}{GPP} * \frac{dGPP}{dC_a} \tag{23}$$

Where GPP is the average annual GPP between the two adjacent years. dGPPproduct of $\mathcal{L}$ and $dC_a$ are the differences of GPP and $C_a$ between two adjacent years respectively.

The sensitivity of yearly average LAI to CO$_2$ is defined as:

$$\beta_{LAI} = \frac{1}{LAI} * \frac{dLAI}{dC_a} \tag{24}$$

Where LAI and dLAI are similarly defined as those about GPP.

Canopy GPP is the sum of $\frac{dc_i}{dc_a}$. For sunlit leaf GPP (GPP$_{sun}$) and shaded leaf, the formulations are:

$$\beta_{p_{sun}} = \mathcal{L}_{sun} \ast \frac{\mathrm{d}C_{isun}}{\mathrm{d}C_a} \tag{6}$$

$$\beta_{p_{sha}} = \mathcal{L}_{sha} \ast \frac{\mathrm{d}C_{isha}}{\mathrm{d}C_a} \tag{7}$$

Leaf-to-canopy scaling factor ($S$) scales fluxes at the single top leaf of the canopy to whole canopy fluxes. The formulations of $S$ for sunlit leaves and shaded leaves are:

$$S_{sun} = \frac{1 - \exp[-\mathrm{LAI}(k_n + k_b)]}{k_n + k_b} \; \mathrm{GPP} \; (\mathrm{GPP}_{sha}). \tag{8}$$

$$S_{sha} = \frac{1 - \exp(-k_n \mathrm{LAI})}{k_n} - \frac{1 - \exp[-\mathrm{LAI}(k_n + k_b)]}{k_n + k_b} \tag{9}$$

Where $k_b$ is extinction coefficient of a canopy of black leaves for direct beam radiation. $k_n$ is an empirical parameter used to describe the vertical distribution of leaf nitrogen in the canopy (Kowalczyk et al., 2006). In our simulation, $k_n$ is uniformly assigned as 0.001 for different PFTs. The leaf-to-canopy scaling factor varies with time because $k_b$ is the function of sun angle, and LAI varies seasonally and inter-annually. The annual value of the leaf-to-canopy scaling factor is just calculated as the average from hourly leaf-to-canopy scaling factors in a year.

Big-leaf $\beta_{\mathrm{GPP}_{sun}}$ (or $\beta_{\mathrm{GPP}_{sha}}$) can be decomposed as the sum of normalized sensitivity of photosynthesis rate: $\beta_{p_{sun}}$ (or $\beta_{p_{sha}}$) and leaf-to-canopy scaling factor: $\beta_{S_{sun}}$ (or $\beta_{S_{sha}}$) as shown in Eq. (10) and Eq. (11). Detailed mathematical derivations are in supplementary Text S2.

$$\beta_{\mathrm{GPP}_{sun}} = \frac{1}{\mathrm{GPP}_{sun}} \ast \frac{\mathrm{dGPP}_{sun}}{\mathrm{d}C_a} = \frac{1}{p_{sun} \ast s_{sun}} \ast \frac{\mathrm{d}(p_{sun} \ast s_{sun})}{\mathrm{d}C_a} = \frac{1}{p_{sun}} \ast \frac{\mathrm{d}p_{sun}}{\mathrm{d}C_a} + \frac{1}{s_{sun}} \ast \frac{\mathrm{d}s_{sun}}{\mathrm{d}C_a} = \mathcal{L}_{sun} \ast \frac{\mathrm{d}C_{isun}}{\mathrm{d}C_a} + \frac{1}{s_{sun}} \ast \frac{\mathrm{d}s_{sun}}{\mathrm{d}C_a} = \beta_{p_{sun}} + \beta_{S_{sun}} \tag{25}$$

$$\beta_{\mathrm{GPP}_{sha}} = \frac{1}{\mathrm{GPP}_{sha}} \ast \frac{\mathrm{dGPP}_{sha}}{\mathrm{d}C_a} = \frac{1}{p_{sha} \ast s_{sha}} \ast \frac{\mathrm{d}(p_{sha} \ast s_{sha})}{\mathrm{d}C_a} = \frac{1}{p_{sha}} \ast \frac{\mathrm{d}p_{sha}}{\mathrm{d}C_a} + \frac{1}{s_{sha}} \ast \frac{\mathrm{d}s_{sha}}{\mathrm{d}C_a} = \mathcal{L}_{sha} \ast \frac{\mathrm{d}C_{isha}}{\mathrm{d}C_a} + \frac{1}{s_{sha}} \ast \frac{\mathrm{d}s_{sha}}{\mathrm{d}C_a} = \beta_{p_{sha}} + \beta_{S_{sha}} \tag{26}$$

[revised manuscript text omitted]

Hurrell, J. W., Holland, M. M., Gent, P. R., Ghan, S., Kay, J. E., Kushner, P. J., ... & Lipscomb, W. H. 2013. The community earth system model: a framework for collaborative research. *Bulletin of the American Meteorological Society*, **94** 1339-1360.

Iversen, C. M., Keller, J. K., Garten, C. T., & Norby, R. J. 2012. Soil carbon and nitrogen cycling and storage throughout the soil profile in a sweetgum plantation after 11 years of $CO_2$-enrichment. *Global Change Biology*, **18** 1684-97.

Jiang, C., Ryu, Y., Fang, H., Myneni, R., Claverie, M., & Zhu, Z. 2017. Inconsistencies of interannual variability and trends in long-term satellite leaf area index products. *Global change biology*, **23**, 4133-4146.

755 Joos, F., Prentice, I. C., Sitch, S., Meyer, R., Hooss, G., Plattner, G. K., ... & Hasselmann, K. 2001. Global warming feedbacks on terrestrial carbon uptake under the Intergovernmental Panel on Climate Change (IPCC) emission scenarios. *Global Biogeochemical Cycles*, **15** 891-907.

Kattge, J., Knorr, W., Raddatz, T., & Wirth, C. 2009. Quantifying photosynthetic capacity and its relationship to leaf nitrogen content for global-scale terrestrial biosphere models. *Global Change Biology*, **15** 976-991.

760 Kowalczyk, E. A., Wang, Y. P., Law, R. M., Davies, H. L., McGregor, J. L., & Abramowitz, G. 2006. The CSIRO Atmosphere Biosphere Land Exchange (CABLE) model for use in climate models and as an offline model. *CSIRO Marine and Atmospheric Research Paper*, **13** 42.

Lamarque, J. F., Bond, T. C., Eyring, V., Granier, C., Heil, A., Klimont, Z., ... & Schultz, M. G. 2010. Historical (1850–2000) gridded anthropogenic and biomass burning emissions of reactive gases and aerosols: methodology and application. *Atmospheric Chemistry and Physics*, **10** 7017-7039.

765 Lamarque, J. F., Kyle, G. P., Meinshausen, M., Riahi, K., Smith, S. J., van Vuuren, D. P., ... & Vitt, F. 2011. Global and regional evolution of short-lived radiatively-active gases and aerosols in the Representative Concentration Pathways. *Climatic change*, **109** 191.

Le Quéré, C.,  Andrew, R. M., Friedlingstein, P., Sitch, S., Pongratz, J.

770 ., Manning, A. C., ... & Boden, T. A. 2017. Global carbon budget  2017. *Earth System Science Data*,  *Discussions*, 1-79.

Leuning, R. 1990. Modelling stomatal behaviour and  photosynthesis of *Eucalyptus grandis*. *Functional Plant Biology*, **17** 159-75.

Leuzinger, S., & Körner, C. (2007). Water savings in mature deciduous forest trees under elevated CO2. *Global Change Biology, 13*(12), 2498-2508.

Leuzinger, S., Luo, Y., Beier, C., Dieleman, W., Vicca, S., & Körner, C. 2011. Do global change experiments overestimate impacts on terrestrial ecosystems? *Trends in ecology & evolution*, **26** 236-41.

Long, S. P., Ainsworth, E. A., Rogers, A., & Ort, D. R. 2004. Rising atmospheric carbon dioxide: plants FACE the Future*. *Annu. Rev. Plant Biol.,* **55** 591-628.

Long, S. P., Ainsworth, E. A., Leakey, A. D., Nösberger, J., & Ort, D. R. 2006. Food for thought: lower-than-expected crop yield stimulation with rising $CO_2$ concentrations. *Science*, **312** 1918-1921.

Luo, Y., Field, C. B., & Mooney, H. A. 1994. Predicting responses of photosynthesis and root fraction to elevated [CO2] a: interactions among carbon, nitrogen, and growth. *Plant, Cell & Environment*, **17**, 1195-1204.

Luo, Y., Sims, D. A., Thomas, R. B., Tissue, D. T., & Ball, J. T. 1996. Sensitivity of leaf photosynthesis to $CO_2$ concentration is an invariant function for C3 plants: A test with experimental data and global applications. *Global Biochemical Cycles*, **10** 209-222.

Luo, Y. and H.A. Mooney. 1996. Stimulation of global photosynthetic carbon influx by an increase in atmospheric carbon dioxide concentration. (In G.W. Koch and H.A. Mooney (eds.) *Carbon Dioxide and Terrestrial Ecosystems*. Academic Press, San Diego) pp. 381-97.

Luo, Y., Su, B. O., Currie, W. S., Dukes, J. S., Finzi, A., Hartwig, U., ... & Pataki, D. E. 2004. Progressive nitrogen limitation of ecosystem responses to rising atmospheric carbon dioxide. *AIBS Bulletin*, **54** 731-9.

Luo, Y., Ahlström, A., Allison, S. D., Batjes, N. H., Brovkin, V., Carvalhais, N., ... & Georgiou, K. 2016. Toward more realistic projections of soil carbon dynamics by Earth system models. *Global Biochemical Cycles*, **30** 40-56.

Matthews, MacFarling Meure, C., Etheridge, D., Trudinger, C., Steele, P., Langenfelds, R., Van Ommen, T., ... & Elkins, J. 2006. Law Dome CO2, CH4 and N2O ice core records extended to 2000 years BP. *Geophysical Research Letters*, **33**.

Mahowald, N., Jickells, T. D., Baker, A. R., Artaxo, P., Benitez‑Nelson, C. R., Bergametti, G., ... & Kubilay, N. 2008. Global distribution of atmospheric phosphorus sources, concentrations and deposition rates, and anthropogenic impacts. *Global biogeochemical cycles*, **22**.

Mystakidis, S., Seneviratne, S. I., Gruber, N., & Davin, E. L. 2017. Hydrological and biogeochemical constraints on terrestrial carbon cycle feedbacks. *Environmental Research Letters*, **12** 014009.

Neff, J. C., Townsend, A. R., Gleixner, G., Lehman, S. J., Turnbull, J., & Bowman, W. D. 2002. Variable effects of nitrogen additions on the stability and turnover of soil carbon. *Nature*, **419** 915.

Norby, R. J., Sholtis, J. D., Gunderson, C. A., & Jawdy, S. S. 2003. Leaf dynamics of a deciduous forest canopy: no response to elevated $CO_2$. *Oecologia*, **136** 574-584.

Norby, R. J., DeLucia, E. H., Gielen, B., Calfapietra, C., Giardina, C. P., King, J. S., ... & De Angelis, P. 2005. Forest response to elevated CO2 is conserved across a broad range of productivity. *Proceedings of the National Academy of Sciences*, **102** 18052-18056.

Pearcy, R. W., & Ehleringer, J. 1984. Comparative ecophysiology of C3 and C4 plants. *Plant, Cell & Environment*, **7** 1-13.

Peng, J., Dan, L., & Huang, M. 2014. Sensitivity of global and regional terrestrial carbon storage to the direct CO2 effect and climate change based on the CMIP5 model intercomparison. *PloS one*, **9** e95282.

Polglase, P. J., & Wang, Y. P. 1992. Potential CO2-enhanced carbon storage by the terrestrial biosphere. *Australian Journal of Botany*, **40** 641-656.

Qu, Y., & Zhuang, Q. 2018. Modeling leaf area index in North America using a process‑based terrestrial ecosystem model. *Ecosphere*, **9**.

Rogers, A., Medlyn, B. E., Dukes, J. S., Bonan, G., Caemmerer, S., Dietze, M. C., ... & Prentice, I. C. 2017. A roadmap for improving the representation of photosynthesis in Earth system models. *New Phytologist*, **213** 22-42.

Schäfer, K. V. R, Oren, R., Ellsworth, D. S., Lai, C. T., Herrick, J. D., Finzi, A. C., Richter, D. D., Katul, G. G.: Exposure to
an enriched $CO_2$ atmosphere alters carbon assimilation and allocation in a pine forest ecosystem, Glob. Chang Biol., 9,
1378–1400, 2003.

Smith, W. K., Reed, S. C., Cleveland, C. C., Ballantyne, A. P., Anderegg, W. R., Wieder, W. R., ... & Running, S. W. 2016.
Large divergence of satellite and Earth system model estimates of global terrestrial $CO_2$ fertilization. *Nature Climate
Change*, **6** 306-10.

Soolanayakanahally, R. Y., Guy, R. D., Silim, Sokolov, A. P., Kicklighter, D. W., Melillo, J. M., Felzer, B. S., Schlosser, C.
A., & Cronin, T. W. 2008. Consequences of considering carbon–nitrogen interactions on the feedbacks between climate
and the terrestrial carbon cycle. *Journal of Climate*, **21** 3776-3796.
S. N., Drewes, E. C., & Schroeder, W. R. 2009. Enhanced assimilation rate and water use efficiency with latitude through
increased photosynthetic capacity and internal conductance in balsam poplar (Populus balsamifera L.). *Plant, Cell &
Environment*, **32** 1821-1832.

[revised manuscript text omitted]

---

## Author Response (AR2)

**The point-by-point response to the reviews**

Response to reviews of manuscript "Leaf Area Index identified as a major source of variability in modelled $CO_2$ fertilization" (bg-2018-213). The original reviewers' comments are italicized and our response to the reviewers' comments follow.

**Response to Associate Editor's comments**

*Comments to the Author:*

*The revised version of this manuscript addresses many of the major concerns raised by reviewers. This assessment is supported by a follow-on review which also offers some suggestions for minor revisions. Before issuing a final acceptance I encourage the authors to consider incorporating these suggested revisions in a final version of the manuscript that I will review quickly once more.*

Response: Thank you for your consistent encouragement. We have incorporated suggestions from the reviewer in the final version of our manuscript. We also devoted a lot of time and efforts to improving the language and readability in this version. We hope this version will lead to a satisfactory acceptance.

*Reviewer 1: Overall, I think that the revisions the author's have carried out considerably improve the readability and framing for the reader, so I thank them for making those efforts. I think the addition of the CN and CNP is a nice contribution even if ultimately it does not change the interpretation; that message in itself is a useful one.*

*I have a few minor responses to their responses to my comments.*

*- In their revision the authors note: "$\beta$ values at ecosystem levels are more variable with nutrient effects, LAI responses are still linearly correlated well with $\beta GPP$, $\beta NPP$ and $\beta cpool$ across different C3 PFTs in nutrient-coupled simulations as in C-only simulation, confirming the dominant role of LAI in regulating carbon cycle response under CO2 fertilization"*

*Here, I think a valuable link with this conclusion would be to discuss current models simulate allocation to leaves (discussion perhaps?). For example, the EucFACE CO2 experiment shows no increased LAI in response to CO2 (Duursma et al. 2016), despite the roughly expected theoretical increase in leaf-level photosynthesis in response to CO2 (Ellsworth et al. 2017). This would question a linear correlation between βGPP and LAI, I think? It is very likely that we have more to learn as we now begin to think further about mature ecosystems.*

*Duursma, R. A., Gimeno, T. E., Boer, M. M., Crous, K. Y., Tjoelker, M. G. and Ellsworth, D. S. (2016), Canopy leaf area of a mature evergreen Eucalyptus woodland does not respond to elevated atmospheric [CO2] but tracks water availability. Glob Change Biol, 22: 1666-1676. doi:10.1111/gcb.13151*

*Ellsworth, D. S., Anderson, I. C., Crous, K. Y., Cooke, J., Drake, J. E., Gher-lenda, A. N., & Tjoelker, M. G. (2017). Elevated CO2 does not increase eucalypt forest productivity on a low-phosphorus soil. Nature Climate Change, 7, 279–282. https://doi.org/10.1038/nclimate3235*

Response: We greatly appreciate the reviewer for sharing these important references with us. We think these references actually support our point that the responses at canopy and ecosystem level are significantly dominated by the response of LAI, rather than by leaf photosynthetic response to $eCO_2$. We have included these references into our discussion: "Some studies reported that LAI dynamics did not significantly change in specific FACE experiments, such as in a closed-canopy deciduous broadleaf forest (ORNL FACE; Norby et al., 2003) and in a mature evergreen broadleaf forest (EucFACE; Duursma et al., 2016). The negligible change of LAI at the EucFACE probably leads to insignificant response of productivity at this site, even though leaf photosynthesis rate significantly increases under $eCO_2$ (Ellsworth et al., 2017)" (Line 344-348).

*Reviewer 1: - To my question about the CABLE simulations almost always being limited by RuBP-regeneration rate ... I agree that at elevated CO2 concentrations this would be true, but I disagree this should be true when the CO2 concentration is "391 ppm" as they stated. If one assumed a Ci/Ca of 0.7, then the Ci concentration would be ~270, which should make the model Rubisco limited (excluding the contribution of LAI). I do suggest they should check this point again. I guess my point is fundamentally about interpretation. A fraction of the readership will read their statement and begin to question whether there is an underlying issue with the model simulations. However, as the authors argue (citing Luo and Mooney), it probably does not matter, but I feel it would be useful to remove any doubt from the reader's*

*mind.*

Response: We thank the reviewer for the careful check of our interpretation. We agree that removing "391 ppm" will make this sentence more accurate. We have changed the sentence to "We found photosynthesis rates are increasingly limited by RuBP regeneration under RCP 8.5 scenario" (Line 175).

*Reviewer 1: - To the author's response about me asking how different levels of water-stress across models would affect their conclusions, they now state: "Our results show modelled ratio of Ci to atmospheric CO2 concentration (Ca) is relatively constant for each PFT with eCO2 and varies little among PFTs (Table 1)". Here I refer them to my original point ... this may very well be true for CABLE, but what about if a model had twice as much water stress as CABLE? Whilst it may not be true that water stress has a bit impact on CABLE's results, it may not be true to conclude this factor does not impact a broader CMIP5 model ensemble which was their original comparison point. It is simply not true to asset that: "Wong et al. (1979) showed plant stomata could maintain a constant Ci/Ca across wide range of environmental conditions, including water stress condition. Therefore different vegetation types might have similar Ci for a given Ca in other models". The Wong study is not a model result, it may very well be theoretically true but models are known to disagree markedly on the impact on water stress, so they cannot have the a "similar Ci for given Ca" across models. In Fig 7, in De Kauwe et al. 2017, Global Change Biology (2017), doi: 10.1111/gcb.13643, I showed the average water stress for a range of models during the growing season. In these simulations Ca was increased in exactly the same way across models, so these differences must have equated to differences in Ci. Finally, they argue that Luo and Mooney showed insensitivity to a change in Ci/Ca from 0.8 to 0.6. But that is essentially without water stress, some of the models in the figure I referred to must have been considerably lower than 0.6. Frankly, a better argument to make to me here - is that water stress comes and goes and that it can be ignored as a factor when looking across years!*

Response: We agree that Wong study is experimental results. We also greatly appreciate the reviewer for offering an alternative explanation that "water stress comes and goes and that it can be ignored as a factor when looking across years". We have carefully studied the paper by De Kauwe et al. (2017) and Fig. 7 in the paper. The figure plots summer water availability factors in term of limiting productivity as water content declines for 10 models. Water influences on productivity through not only $C_i/C_a$ ratio but also many other processes, such as canopy structure as explored in our study of this paper. It is hard to infer from that figure that $C_i/C_a$ "must be lower than 0.6".

We cited this paper in the revised manuscript:" Different models simulate diverse levels of water stress on productivity (De Kauwe et al., 2017)" (Line 424). We also added one sentence in our revised manuscript: "Land surface models might simulate relatively constant $C_i/C_a$ ratios under water stress as well since photosynthesis and stomatal conductance are theoretically depicted based on experimental results" (Line 431-433).

*Reviewer 1: - To their response on leaf temperature. I can't recall the details of the Luo and Mooney study, which they seem to know well. Whilst I do not take issue with their point on the impact of leaf temp on gamma star, I was actually thinking about the impact of the different resolved leaf temperatures on Ci, which is solved via the energy balance. However, this is most likely speculative (on my part) anyway, so can be ignored.*

Response: The reviewer raised a good question. Actually there is not a clear relationship between leaf temperature and intercellular $CO_2$ concentration for $C_3$ plants (Fig. 1), indicating other factors such as the stomatal conductance might play an important role in regulating $C_i$.

[Figure]

Fig. 1: Dependence of intercellular $CO_2$ concentration ($C_i$) on leaf temperature for sunlit leaf (a) and shaded leaf (b).

*Reviewer 1: - In their reframing of the text in reference to the Hajima et al. paper, I wonder if the author's could go further in their explanations. This strikes me as a very important and interesting point, but as written is a little too superficial. The author's are arguing that one gets a different interpretation of β if one calculates it at the leaf-level vs. as canopy/stand-level, GPP/LAI. They appear to attribute this to the fact that the authors have ignored differences in the scaling in how the canopy is treated. If I follow, these are the only factors, or were there more? If these are the only factors, if the calculation is formulated on*

*a big-leaf vs a two-leaf vs a multi-layer, which one best matches the GPP/LAI formulation that Hajima used? Or which PFTs does this issue matter most for? These strike me as points worth making, or perhaps I've oversimplified?*

Response: We thank the reviewer for the insightful comments. We believe that which canopy structure best fits GPP/LAI formulation that Hajima used is an interesting research question. Canopy structure is one of the important factors. We would like to expand on this point through explaining which PFTs this issue matters most for. We have reorganized this part into:" One possible issue of this calculation is that it ignores different canopy structure used by each CMIP5 model such as big-leaf, two-leaf or multiple-layer. Our results just show that the sensitivities of GPP/LAI are different from our mechanistic calculation of leaf-level $\beta$ for different PFTs in a two-leaf model. $\beta$ values estimated from GPP/LAI formulation are greatly underestimated for woody trees and slightly overestimated for $C_3$ grass and tundra, but best match for shrub if compared with our calculation (Fig. S8). Therefore, diagnostics such as $C_i$ and $\Gamma_*$ for leaf-level $\beta$ are more desirable for woody trees" (Line 319-324).

**The list of all relevant changes made in the manuscript**

Here are the relevant changes made in the manuscript.

(1) Page 8 line 158-159: we clarified the relationship between canopy GPP and big-leaf GPP.

(2) Page 8 line 175: we modified the sentence as suggested by the reviewer.

(3) Page 15 line 319-324: we added the discussion as suggested by the reviewer.

(4) Page 16 line 344-348: we discussed the references as suggested by the reviewer.

(5) Page 19 line 424: we cited the paper as mentioned by the reviewer.

(6) Page 20 line 431-433: we discussed the references as suggested by the reviewer.

[revised manuscript text omitted]